# No Representation, No Trust: Connecting Representation, Collapse, and Trust Issues in PPO

**Skander Moalla**[1][*]  **Andrea Miele**[1]  **Daniil Pyatko**[1]  **Razvan Pascanu**[2]  **Caglar Gulcehre**[1]

[1] CLAIRE, EPFL   [2] Google DeepMind

## Abstract

Reinforcement learning (RL) is inherently rife with non-stationarity since the states and rewards the agent observes during training depend on its changing policy. Therefore, networks in deep RL must be capable of adapting to new observations and fitting new targets. However, previous works have observed that networks trained under non-stationarity exhibit an inability to continue learning, termed loss of plasticity, and eventually a collapse in performance. For off-policy deep value-based RL methods, this phenomenon has been correlated with a decrease in representation rank and the ability to fit random targets, termed capacity loss. Although this correlation has generally been attributed to neural network learning under non-stationarity, the connection to representation dynamics has not been carefully studied in on-policy policy optimization methods. In this work, we empirically study representation dynamics in Proximal Policy Optimization (PPO) on the Atari and MuJoCo environments, revealing that PPO agents are also affected by feature rank deterioration and capacity loss. We show that this is aggravated by stronger non-stationarity, ultimately driving the actor's performance to collapse, regardless of the performance of the critic. We ask why the trust region, specific to methods like PPO, cannot alleviate or prevent the collapse and find a connection between representation collapse and the degradation of the trust region, one exacerbating the other. Finally, we present Proximal Feature Optimization (PFO), a novel auxiliary loss that, along with other interventions, shows that regularizing the representation dynamics mitigates the performance collapse of PPO agents. Code and run histories are available at https://github.com/CLAIRE-Labo/no-representation-no-trust.

## 1   Introduction

Reinforcement learning (RL) agents are inherently subject to non-stationarity as the states and rewards they observe change during learning. Therefore, neural networks in deep RL must be capable of adapting to new inputs and fitting new targets. However, previous works have observed that networks trained under non-stationarity exhibit an inability to continue learning, termed loss of plasticity, and a collapse in performance (Dohare et al., 2021; Abbas et al., 2023; Kumar et al., 2023; Dohare et al., 2023a,b). Kumar et al. (2021); Lyle et al. (2022) connect this phenomenon to representation dynamics and show that value networks in off-policy value-based RL algorithms exhibit a decrease in the rank of their representations, termed feature rank collapse, and a decrease in their ability to regress to arbitrary targets, called capacity loss. Although this deterioration in representation is more generally attributed to neural networks trained under non-stationarity (Lyle et al., 2023), the connection to representation dynamics has been overlooked in on-policy policy optimization methods. In particular, Proximal Policy Optimization (PPO) (Schulman et al., 2017), one of the most popular policy optimization methods, makes several minibatch updates over non-stationary data, unlike vanilla policy gradient methods, and optimizes a surrogate loss that depends on a moving old

---

[*]Correspondence to skander.moalla@epfl.ch.

policy. This raises the question of how much PPO agents are impacted by the same representation degradation attributed to non-stationarity. Dohare et al. (2021, 2023a,b) have shown that PPO agents lose plasticity throughout training but have only diagnosed it as a collapse in performance or as an Adam optimization issue. Igl et al. (2021) have shown that non-stationarity affects the generalization of PPO agents (learning speed when training episodes are very different otherwise performance at test time on novel episodes) but not necessarily training, and no connection was made with the feature rank and capacity measures used in the recent value-based works. One crucial outstanding question is why the trust region embedded in methods like PPO is unable to prevent the deterioration in policy by constraining its update. To address these gaps, we present the following contributions:

1. We provide the first study of feature rank and capacity loss in on-policy policy optimization, revealing that PPO agents in the Arcade Learning Environment (Bellemare et al., 2013) and MuJoCo (Todorov et al., 2012) environments are subject to representation collapse.

2. We draw connections between representation collapse, performance collapse, and trust region issues in PPO, showing that PPO's clipping becomes ineffective under poor representations and fails to prevent performance collapse, which is irrecoverable due to loss of capacity. We further isolate the breakdown of the trust region in a theoretical setting.

3. We corroborate these connections by performing interventions that regularize non-stationarity and representations and result in a better trust region and mitigation of performance collapse, incidentally giving insights on sharing an actor-critic trunk.

4. We propose *Proximal Feature Optimization* (PFO), a new regularization on the representation of the policy that regularizes the change in pre-activations. PFO strengthens our analysis by addressing the representation issues and mitigating performance collapse.

5. We open source our code providing a comprehensive and reproducible codebase for studying representation dynamics in policy optimization and a large database of run histories with extensive logging for further investigation on this topic.

## 2 Background

**Reinforcement Learning (Sutton & Barto, 2018)** We formalize our RL setting with the finite-horizon undiscounted Markov decision process, describing the interaction between an agent and an environment with finite [2] sets of states $\mathcal{S}$ and actions $\mathcal{A}$, and a reward function $r : \mathcal{S} \times \mathcal{A} \times \mathcal{S} \to \mathbb{R}$. An initial state $S_0 \in \mathcal{S}$ is sampled from the environment, then at each time step $t \in \{0, \ldots, t_{\max} - 1\}$, the agent observes the state $S_t \in \mathcal{S}$, picks an action $A_t \in \mathcal{A}$ according to its policy $\pi : \mathcal{S} \to \Delta(\mathcal{A})$ with probability $\pi(A_t|S_t)$, [3] observes the next state $S_{t+1} \in S$ sampled from the environment and receives a reward $R_{t+1} \doteq r(S_t, A_t, S_{t+1})$. We denote by $G_t \doteq \sum_{k=t}^{t_{\max}-1} R_{k+1}$ the return after the action at time step $t$. The goal of the agent is to maximize its expected return $J(\pi) \doteq \mathbb{E}_\pi\left[\sum_{t=0}^{t_{\max}-1} R_{t+1}\right] = \mathbb{E}_\pi[G_0]$ over the induced random trajectories. We discuss the choice of this setting in Appendix A.1.

**Actor-Critic Agent** We consider on-policy deep actor-critic agents which train a policy (or actor) network $\pi(\cdot; \boldsymbol{\theta})$ also denoted $\pi_{\boldsymbol{\theta}}$, and a value (or critic) network $\hat{v}(\cdot; \mathbf{w})$ that approximates the return of $\pi_{\boldsymbol{\theta}}$ at every state. At every training stage, the agent collects a batch of samples, called rollout, with its current policy $\pi_{\boldsymbol{\theta}}$, and both networks are trained with gradient descent on this data. The critic is trained to minimize the Euclidean distance to an estimator of the returns (e.g., $G_t$). We use $\lambda$-returns computed with the Generalized Advantage Estimator (GAE) (Schulman et al., 2015b). The actor is trained with the Proximal Policy Optimization (PPO) (Schulman et al., 2017).

**Proximal Policy Optimization** PPO-Clip, the most popular variant of PPO algorithms (Schulman et al., 2017), optimizes the actor by repeatedly maximizing the objective in Equation 1 at each rollout.

$$L_{\pi_{\text{old}}}^{CLIP}(\boldsymbol{\theta}) = \mathbb{E}_{\pi_{\text{old}}}\left[\sum_{t=0}^{t_{\max}-1} \min\left(\frac{\pi_{\boldsymbol{\theta}}(A_t|S_t)}{\pi_{\text{old}}(A_t|S_t)}\Psi_t, \text{clip}\left(\frac{\pi_{\boldsymbol{\theta}}(A_t|S_t)}{\pi_{\text{old}}(A_t|S_t)}, 1+\epsilon, 1-\epsilon\right)\Psi_t\right)\right] \quad (1)$$

---

[2] The pixel-based environment with discrete actions used in our experiments and our simple theoretical example in Section 3.2.1 fit the finite state and action formalism but not our continuous action space environment. We refer the reader to Szepesvári (2022) for a formalism of RL in that setting.

[3] The time step $t$ is included in the representation of $S_t$ to preserve the Markov property in finite-horizon tasks as done by Pardo et al. (2018) and is analogous to considering time-dependent policies in the classical formulation of finite-horizon MDPs.

The objective is defined for some small hyperparameter $\epsilon$; $\pi_{\text{old}}$ is the last $\pi_{\boldsymbol{\theta}}$ of the previous optimization phase, used to collect a training batch (rollout) after each optimization phase; $\Psi_t$ is an estimator of the advantage of $\pi_{\text{old}}$ (e.g., $\Psi_t = G_t - \hat{v}(S_t; \mathbf{w}_{\text{old}})$); we use the GAE in our experiments. An optimization phase consists of maximizing the objective with minibatch gradient steps over multiple epochs on the training batch. We refer to PPO-Clip as PPO and provide a pseudocode in Algorithm 1.

Intuitively, PPO aims to maximize the policy advantage $\mathbb{E}_{\pi_{\text{old}}}\left[\sum_{t=0}^{t_{\max}-1} \frac{\pi_{\boldsymbol{\theta}}(A_t|S_t)}{\pi_{\text{old}}(A_t|S_t)}\Psi_t\right]$ defined by Kakade & Langford (2002), which participates in a lower bound to the improvement of $\pi_{\boldsymbol{\theta}}$ given that it is close to $\pi_{\text{old}}$ (Schulman et al., 2015a, see Theorem 1). In this regard, a gradient step on $L_{\pi_{\text{old}}}^{CLIP}(\theta)$ would increase (resp. decrease) the probability of actions at states yielding positive (resp. negative) advantage until the ratio between the policies for those actions reaches $1 + \epsilon$ (resp. $1 - \epsilon$) at which point the gradient at those samples becomes null. This is a heuristic to ensure a trust region that keeps policies close to each other, resulting in policy improvement.

**Non-stationarity in deep RL and PPO** The actor and the critic networks are both subject to non-stationarity in deep RL. As the agent improves, it visits different states, shifting the distribution of states which makes the networks' input distribution non-stationary. This also holds for the targets to fit the critic, which change as the returns of the policy change. Unlike vanilla policy gradient (Sutton et al., 1999), and A2C (Mnih et al., 2016), PPO's objective is optimized by performing multiple epochs of minibatch gradient descent on the current collected batch, potentially making the networks more likely to be impacted by previous training rollouts. In this sense, increasing the number of epochs in PPO can cause the agent to "overfit" more to previous experience.

**Feature rank** As done in most works studying feature dynamics in deep RL (Lyle et al., 2022; Kumar et al., 2021), we refer to the activations of the last hidden layer of a network (the penultimate layer) as the features or representation learned by the network. On a batch of $N$ samples, this gives a matrix of dimensions $N \times D$ denoted by $\Phi$, where $D < N$ is the width of the penultimate layer. Several measures of the rank of this matrix have been used to quantify the "quality" of the representation (Kumar et al., 2021; Gulcehre et al., 2022; Lyle et al., 2022; Andriushchenko et al., 2023). Their absolute values differ significantly, but their dynamics are often correlated. We track all of the different rank metrics in our experiments, compare them in Appendix E, and use the *approximate rank* in our main figures for its connection to principal component analysis (PCA). Given a threshold $\delta \in \mathbb{R}$ and the singular values $\langle \sigma_i(\Phi), \ldots, \sigma_D(\Phi) \rangle$ of $\Phi$ in decreasing order, the approximate rank of $\Phi$ is $\min_k \left\{ \frac{\sum_{i=1}^k \sigma_i^2(\Phi)}{\sum_{j=1}^D \sigma_j^2(\Phi)} > 1 - \delta \right\}$ which corresponds to the smallest dimension of the subspace recovering $(1 - \delta)\%$ of the variance of $\Phi$. We use $\delta = 0.01$ i.e. the reconstruction recovers $99\%$ of the variance as done by Andriushchenko et al. (2023); Yang et al. (2020). We refer to this metric as *feature* rank with reference to the rank of the *feature* matrix when there is no ambiguity.

**Capacity loss** Target-fitting capacity (Lyle et al., 2022) is computed on checkpoints of a network undergoing some training to measure the evolution of its ability to fit some chosen target independent from its training. It is a concrete metric to evaluate plasticity. Given a fixed target (distribution over inputs and outputs) and a fixed optimization budget, a checkpoint's capacity loss is the loss from fitting the checkpoint to the target at the end of the optimization budget. Usually, the capacity of a deep RL agent is measured by its ability to fit the outputs of a model initialized randomly from the same distribution as the agent on a fixed rollout collected by this target random model (Lyle et al., 2022; Nikishin et al., 2023). We follow this practice. The data would in expectation be from the same distribution as the agent's initial checkpoint. To fit the critic, we use an $L^2$ loss on the outputs of the models. To fit the actor, we use a KL divergence between the target and the checkpoint (forward KL).

## 3 Deteriorating representations, collapse, and loss of trust

It is well-known that non-stationarity in deep RL can be a factor causing issues in representation learning. However, most of the observations have been made in value-based methods showing that value networks are prone to rank collapse, harming their expressivity, and in turn, the performance of the agent (Lyle et al., 2022; Kumar et al., 2022). Non-stationarity has been shown to impact PPO's generalization Igl et al. (2021) and performance in the long run or in a continual learning setting (Dohare et al., 2021, 2023a,b), but no evidence of representation deterioration was shown. Our motivation is to reuse the tools that showed that value-based methods are prone to representation collapse but in policy optimization methods for the first time. We focus on PPO for its popularity and its non-stationarity which is impacted and can be controlled by multi-epoch optimization.

Furthermore, a crucial question for PPO, compared to most value-based alternatives, is how the regularization implicit in PPO through its trust region interacts with representation and performance collapse. Intuitively it should prevent rapid degradation of the policy.

**Experimental setup** We begin our experiments by training PPO agents on the Arcade Learning Environment (ALE)(Bellemare et al., 2013) for pixel-based observations with discrete actions and on MuJoCo (Todorov et al., 2012) for continuous observations with continuous actions. To keep our experiments tractable, we choose the Atari-5 subset recommended by Aitchison et al. (2023) and add Gravitar to include at least one sparse-reward hard-exploration game from the taxonomy presented by Bellemare et al. (2016). For MuJoCo, we train on Ants, Half-Cheetahs, Humanoids, and Hoppers, which have varying complexity and observation and output sizes. We use the same model architectures and hyperparameters as popular implementations of PPO on ALE and MuJoCo (Raffin et al., 2021; Huang et al., 2022b); these are also the architectures and hyperparameters used by Schulman et al. (2017) in the original implementation of PPO; they do not include normalization layers. For MuJoCo we further adopt a parameterization of the output action distribution using a TanhNormal[4] with both its mean and variance depending on the state representation as done by Haarnoja et al. (2018); Andrychowicz et al. (2021). As we study the connection between performance and representation dynamics this is a more natural choice than using the commonly implemented state-independent variance which would be independent of representation dynamics. The ALE models use ReLU activations (Nair & Hinton, 2010) and the MuJoCo ones tanh; we also experiment with ReLU on MuJoCo. We use separate actor and critic models for both environments unless specified in Section 4. Details on the performance metrics and tables of all environment parameters, model architectures, and algorithm hyperparameters are presented in Appendix B. Observing that the previous findings on the feature dynamics of value-based approaches (Gulcehre et al., 2022; Lyle et al., 2022) apply to the critic of PPO as well since the loss function is the same, we focus on studying the feature dynamics of the actor unless stated otherwise in the text or figures.

We vary the number of epochs as a tool to control the effects of non-stationarity, which gives the agent a more significant number of optimization steps per rollout while not changing the optimal target it can reach due to clipping, as opposed to changing the value of $\epsilon$ in the trust region for example.[5] We keep the learning rate constant throughout training and use the same learning rate for all the epoch configurations.[6] To understand the feature dynamics, we measure different metrics that are proposed in the literature, including feature rank, number of dead neurons (Gulcehre et al., 2022), capacity loss (Lyle et al., 2022), and penultimate layer pre-activation norm. Previous work has monitored feature norm values as well (Abbas et al., 2023; Lyle et al., 2024); however, in our case, we found that as the neurons in the policy network die, the feature norm might be stable while the pre-activation norm blows up. All the metrics are computed on on-policy rollouts except for the capacity loss.

We run five seeds per hyperparameter configuration and report mean curves with min/max shaded regions unless specified otherwise. All curves, except for capacity loss, are smoothed using an exponentially weighted moving average with a coefficient of $0.05$.

### 3.1 PPO suffers from deteriorating representations

**Deteriorating representation** How do the representation metrics of a PPO agent such as the *feature rank* and the *capacity loss*, evolve during training? Are they subject to the same decline observed by Kumar et al. (2021); Lyle et al. (2022) in value-based methods? Does it affect performance?

As illustrated in Figure 1 with ALE/Phoenix as an example, we observe a consistent increase in the norm of the pre-activations of the feature layer of the policy network. Learning curves for all the ALE games and MuJoCo tasks considered can be found in Appendix D. The increase in feature norm is present in all the games/tasks considered in both environments, that is, with the two different model architectures and activation functions in the case of MuJoCo. We associate the rapid growth in the norm of the pre-activations with an eventual decline in the policy network's feature rank. We observe a rank decline in five out of six ALE games and seven out of eight MuJoCo tasks (four with ReLU and three with tanh). The same observations about the increasing norm of the pre-activations can be made about the critic network. However, its rank varies more with the sparsity of the reward: in

---

[4]We also provide evidence of collapse with the Gaussian distribution parameterization in the appendix.

[5]We show results with varying $\epsilon$ in Figure 34 of Appendix D.

[6]Although the environments we use are single-task environments to ablate additional MDP non-stationarity, they are complex enough for the agents to keep improving when trained for longer than common benchmark limits without annealing the learning rate.

most environments, its rank experiences a significant deterioration after the policy's performance declines (not the policy's rank) and rewards become sparser, and in the sparse-reward game Gravitar, the critic's rank collapses before the policy. Furthermore, capacity loss is increasing for the critic, as observed in value-based plasticity studies (Lyle et al., 2022), and we also show that is the case for the actor, for which it explodes around rank collapse.

**Worse consequences**  How does increasing the number of epochs per rollout to vary non-stationarity affect a PPO agent's representation? Does it degrade as observed in DQN and SAC agents when increasing the replay ratio (Nikishin et al., 2022; Kumar et al., 2022)?

Increasing the replay ratio in DQN and SAC deteriorates the agent's representation and, in turn, its performance (Kumar et al., 2022; D'Oro et al., 2023). This is commonly attributed to "overfitting" to previous experience (Nikishin et al., 2022). Increasing the number of epochs in PPO is analogous, and a natural hypothesis is that this would accelerate the deterioration of the policy's representation. Figure 1 shows that increasing the number of epochs accelerates the increase of pre-activations norm and the decrease of the policy's feature rank.[7] In some cases, the rank eventually collapses, coinciding with the policy's performance collapse. We observe the performance collapse in three of the six ALE games and three of the four MuJoCo tasks.

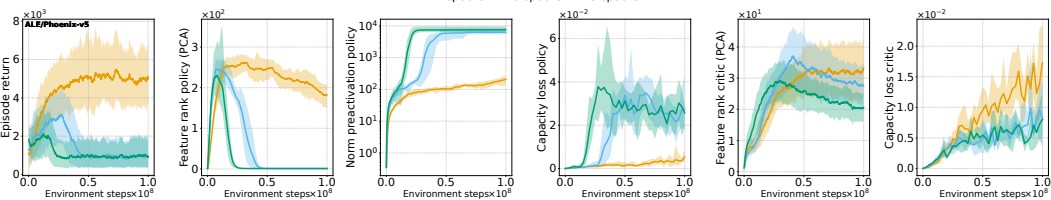

Figure 1: **Deteriorating performance and representation metrics** The policy network of a PPO-Clip agent on ALE/Phoenix-v5 is subject to a deteriorating representation. The norm of the pre-activations of the penultimate layer consistently increases, and its rank eventually decreases. Performing more optimization epochs per rollout to increase the effects of non-stationarity accelerates the growth of the norm of the pre-activations and the collapse of its rank. This ultimately leads to the collapse of the policy. This collapse is not driven by the value network, whose rank is still high. Both network's ability to fit arbitrary targets (capacity loss) is also worsening.

**Characterizing the collapse**  The collapse we observe is distinct from the typical entropy collapse. Figure 2 shows that the policy reaches a high entropy. A high overall entropy can come from an average of high-entropy states with different action distributions or trivially from the same high-entropy distribution in all states. Our analysis reveals the latter, a zero policy variance across states. This corresponds to a collapsed representation where most neurons are inactive. [8] The output thus relies solely on the bias term, as linear weights act on a null feature vector, making actions near uniform across all states and collapsing performance on complex tasks.

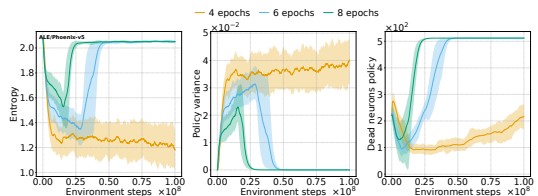

Figure 2: **Rank collapse gives a high but trivial entropy** The rank collapse of the policy gives a policy with high entropy but zero variance across states. The network outputs the same high-entropy action distribution in all states, as all the neurons in the feature layer are dead. Its output only depends on the constant bias term.

## 3.2 Collapsed representations create trust issues and unrecoverable loss

Intuitively, the heuristic trust region set by PPO-Clip should prevent sudden catastrophic changes and limit the rank collapse, which induces worse performance. However, empirically, it seems the trust region cannot mitigate the collapse. In this section, we seek to understand the interaction between the rank collapse and the trust region. We argue that as rank collapses, the clipping constraint becomes unreliable and unable to restrict learning. This is in line with previous works that have pointed out

---

[7]The collapse happens with all epochs configurations when trained for long enough as seen in Figure 32, increasing the number of epochs is a tool to observe the collapse earlier rather than a condition for it to happen.

[8]We consider a ReLU neuron as dead when its values are zero for all the samples in the batch and a tanh neuron dead when its standard deviation across samples is less than 0.001.

that probability ratios during training can go beyond the clipping limits with PPO-Clip (Engstrom et al., 2020; Wang et al., 2020; Sun et al., 2022). We believe, however, that this behavior is systematic when rank collapses and does not merely happen occasionally.

Wang et al. (2020, Theorem 2) state that when the gradients of the unclipped samples align with the gradients of clipped samples, the clipped samples' ratios will have their probabilities continue to go beyond the clip limit. They claim this condition would hold in practice because of "optimization tricks" or optimizer accumulated moments; however, there is no evidence that these factors induce the gradient alignment or that the alignment is present in practice. Our intuition is that representation degradation leads to alignment in the gradients and, therefore, a breakdown of the trust region constraint. This can create a snowball effect, preventing PPO-Clip from preventing representation collapse. We summarize this in two observations:

**Loss of trust is extreme around poor representations** The average of probability ratios outside the clipping limits (below $1 - \epsilon$ in Figure 3) significantly diverges from the clipping limit around the collapse of the agent's representation. This gives one more reason why the PPO trust region can be violated. We isolate this in a toy setting and analyze it formally in the next section. We further show in Figure 4 scatter plots of the lowest average probability ratios in runs with their associated representation metrics.[9] We observe no significant correlation in the regions where the representation is rich (high rank, low pre-activation norm), but an apparent decrease of the average of probability ratios below $1 - \epsilon$ is observed as the representation reaches poor values. Note that we characterize the collapsing regime by an extremely low rank, however, it is not straightforward to draw a line between low-rank representations beneficial for generalization and extremely low-rank representations causing aliasing as also acknowledged by Gulcehre et al. (2022), but for environments like Atari, our figures seem to draw the line at single-digit ranks, which can be related to the action space of dimension 8+.

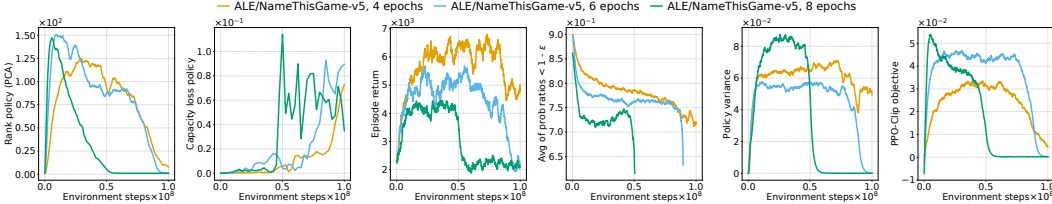

Figure 3: **Focusing on individual runs** Individual training curves on ALE/NameThisGame-v5 with different epochs per batch. Extremely low ratios are observed around the representation collapse of a PPO-Clip agent, implying that the heuristic trust region breaks down when representation power is lacking. The last-minibatch value of the PPO objective decreases towards 0 around the representation collapse, implying a reduction in the ability to improve the policy and recover, which is corroborated by the increase in capacity loss. (Ratios are trivially above $1 - \epsilon$ after collapse as a collapsed model does not change much to have values below $1 - \epsilon$.)

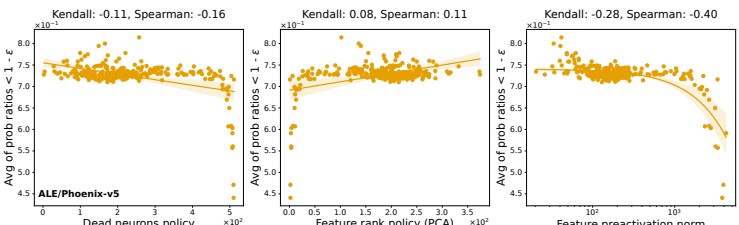

Figure 4: **Representation vs trust region** Samples from ALE/Phoenix-v5 training curves. Each point maps an average of the probability ratios below the clipping limit vs. its corresponding average representation metric (dead neurons, feature rank, feature norm). The average ratios are significantly lower around poor representations (high dead neurons, low policy rank, high feature norm) reflecting the failure of the trust region in this regime. Averages are over non-overlapping windows larger than episodes.

**Loss of plasticity makes performance collapse unrecoverable** The persistent decrease in performance overlaps with a monotonic decrease in policy variance and PPO objective. It appears that as

---

[9]20 points per run, across windows of size 1% training progress, spanning at least the horizon of the environment, so that points are well spaced in the run, with each point being the average of the window

the policy loses its ability to distinguish state, it can also ascend the PPO objective less and less at each batch (recall: after collecting a batch, the loss starts around zero with a normalized advantage, and through minibatch updates, the clipped policy advantage is ascended). Intuitively, this is implied by a loss of plasticity or a collapse in entropy (no new actions to learn from). As seen in Section 2 the entropy does not collapse, and measuring the capacity loss in Figure 3 shows that the decrease in objective gain is associated with a significant increase in capacity loss, implying loss of plasticity.

**Connecting the dots** Hence, around collapse, the representation of the policy is getting so poor that it is impacting its ability to distinguish and act differently across states; the trust region cannot prevent this catastrophic change as it also breaks down with a poor representation; finally, the policy's plasticity is also becoming so poor that the agent cannot recover by optimizing the surrogate objective.

**Implications and discussion** The causal connection we draw between the representation dynamics, the trust region, and the performance primarily holds around the collapse regime and not necessarily throughout training. However, this does not mean that one should only be concerned about the link when performance is starting to deteriorate. The representations don't collapse all of a sudden; they deteriorate throughout training until they reach collapse. Thus, mitigating representation degradation should happen throughout training and not only when around the collapsing regime. In addition, the connection gives important insights into the failure mode of the popular PPO-Clip algorithm, whose trust region is highly dependent on the representation quality, and more generally about trust-region methods which only constrain the output probabilities.

### 3.2.1 A toy setting to understand the effects of rank collapse on trust region

We present a toy example that illustrates how a collapsed representation bypasses the clipping set by PPO and cannot satisfy the trust region it seeks to set. PPO constructs a trust region around the policy $\pi_{\boldsymbol{\theta}}(\cdot|s)$ of the agent evaluated at a given state $s$, enforcing (in an approximate way) that the update computed on state $s$ can not move the policy $\pi_{\boldsymbol{\theta}}(\cdot|s)$ outside of the trust region. However, the constraint does not capture how updates computed on another state $s'$ affect the policy's probability distribution over the current state $s$. The underlying assumption is that updates computed on different states are, at least in expectation, approximately orthogonal to each other, and they do not interact. Therefore, restricting the update of the current state is sufficient to keep the policy within the region.

In our case, however, one can show that as the rank collapses or the neurons die, the representations corresponding to different states become more colinear.[10] Therefore, the gradients also become more colinear. In the extreme case, when the rank collapses to 1, or there is only one neuron alive, all representations are exactly colinear; therefore, all gradients are also. This means that even though clipping prevents the policy $\pi_{\boldsymbol{\theta}}(\cdot|s)$ on the current state $s$ from changing due to the update of that state $\nabla L(\pi_{\boldsymbol{\theta}}(\cdot|s))$, $\pi_{\boldsymbol{\theta}}(\cdot|s)$ will still change and move outside of the trust region due to the updates on other states $s'$. Leading to the trust region constraint being ineffective and not constraining the learning process in any meaningful sense. This gives a clear situation where the theorem of Wang et al. (2020) holds and can easily be analyzed as below without resorting to the theorem for an end-to-end proof or to get a better intuition.

**Formal statement of the toy setting**
Let us consider a batch containing two state-action pairs $(x, a_1)$ and $(y, a_1)$ with sampled probabilities $\pi_{\text{old}}(a_1|x)$ and $\pi_{\text{old}}(a_1|y)$ and positive estimated advantages $A(x, a_1), A(y, a_1) > 0$. Let $\phi(x), \phi(y) \in \mathbb{R}$ be fixed 1-dimensional representations of $x$, and $y$ that can be seen as the output of the (frozen) penultimate layer of a policy network with collapsed representation (all but one dead neuron), and let $\alpha \in \mathbb{R}$ such that $\phi(y) = \alpha\phi(x)$. Let $\boldsymbol{\theta} = [\theta_1, \theta_2]$, be the last layer of the network, computing the logits of two actions, $a_1$ and $a_2$, that are

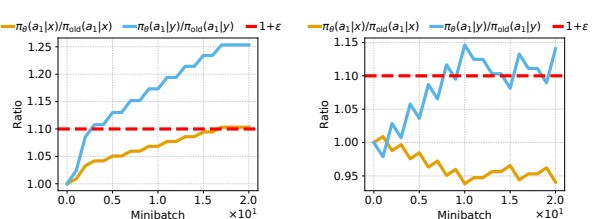

Figure 5: **Simulation of the toy setting** Left ($\alpha > 0$): a gradient on $(x, a_1)$ takes the probability of $(y, a_1)$ up and vice versa. When one is above the threshold and should not increase, the other still pushes it. Right ($\alpha < 0$): a gradient on $(x, a_1)$ takes the probability of $(y, a_1)$ down and vice versa. Both slow each down, with one forcing the other to be lower than its initial value.

fed into a softmax to compute the probabilities. I.e., $\pi_{\boldsymbol{\theta}}(a_i|s) = \frac{e^{\theta_i \phi(s)}}{e^{\theta_1 \phi(s)} + e^{\theta_2 \phi(s)}}$. Consider PPO

---

[10]The expected angle between representations shrinks to 0.

minibatch updates alternating between $(x, a_1)$ and $(y, a_1)$. Ideally, the PPO loss increases $\pi_{\boldsymbol{\theta}}(a_1|s)$ at gradients on $(x, a_1)$ until it reaches the clip ratio and similarly on $(y, a_1)$. However, we show in Appendix C that a gradient step in $(x, a_1)$ also affects $\pi_{\boldsymbol{\theta}}(a_1|y)$ and depending on $\alpha$ will increase it past its clipped ratio, or decrease it below its initial value. Essentially, when $\alpha \geq 0$, a gradient on $(x, a_1)$ increases $\theta_1^{\text{new}}$ therefore increasing both $\pi_{\boldsymbol{\theta}^{\text{new}}}(a_1|x)$ and $\pi_{\boldsymbol{\theta}^{\text{new}}}(a_1|y)$. The same holds for a gradient on $(y, a_1)$, causing one state to reach the clip limit first depending on $\alpha > 1$ but still have the other keep pushing its probability upwards. However, when $\alpha \leq 0$, a gradient on $(x, a_1)$ increases $\theta_1^{\text{new}}$ therefore increasing $\pi_{\boldsymbol{\theta}^{\text{new}}}(a_1|x)$ but decreasing $\pi_{\boldsymbol{\theta}^{\text{new}}}(a_1|y)$. For a gradient on $(y, a_1)$ it is the opposite: $\theta_1^{\text{new}}$ decreases therefore $\pi_{\boldsymbol{\theta}^{\text{new}}}(a_1|x)$ decreases and $\pi_{\boldsymbol{\theta}^{\text{new}}}(a_1|y)$ increases, causing each state to reduce the probability of the other, and depending on $\alpha < 1$ one of the probabilities will dominate and push the other one down. Figure 5 shows the evolution of the probabilities when simulating the updates empirically.

## 4 Intervening to regularize representations and non-stationarity

Having observed that PPO is affected by a frequent representation degradation that impacts its trust region heuristic and causes its performance to collapse, we turn to study interventions that aim at regularizing the representation of the policy network or reducing the non-stationarity in the optimization. We investigate whether these interventions improve the representation metrics we track and if in turn, this affects performance. We choose simple interventions that do not apply modifications to the models during training (e.g., resetting or adding neurons) or require significantly more memory (e.g., maintaining separate copies of the models). We perform interventions on the games/tasks where the collapse is the most significant. We are interested in the state of the agent at the end of the training budget. We record the performance and representation metrics for each run as averages over the last 5% of training progress. We measure the excess ratio at a timestep as the average probability ratio above $1 + \epsilon$ divided by the average probability ratio below $1 - \epsilon$ at that timestep. This metric gives an idea of how much the policy exceeds the trust region. Its average value is computed over the last 5% of training progress where the ratios are non-trivial, giving the same window at the end of training as the other metrics when there is no collapse, otherwise a window before total collapse covering 5% of training progress, as after collapse, the model does not change anymore and the ratios are trivially within the $1 + \epsilon$ and $1 - \epsilon$ limits. We give additional details on the computation of these aggregate metrics and the interventions performed in Appendix B.

**PFO: Regularizing features to mitigate trust-region issues** The motivation for our first intervention and our proposed regularization method comes from our observation that the norm of the preactivation features is consistently increasing, which can be linked to the trust-region issues discussed in Section 3. We seek to mitigate this effect in a way that is analogous to the PPO trust region, by extending the trust region to the feature space. We apply an $L^2$ loss on the difference between the pre-activated features of the optimized policy and the policy that collected the batch, as a way to keep the pre-activations of the network during an update within a trust region. We apply this regularization to the pre-activations and not the activations, as dead neurons cannot propagate gradients, and even when they do, depending on the activation function, do so with a low magnitude. The regularization is an additional loss/penalty added to the overall loss. We term this loss the Proximal Feature Optimization (PFO) loss. With $\phi_{\boldsymbol{\theta}}(s)$ as the pre-activation of the penultimate layer of the actor $\pi_{\boldsymbol{\theta}}$ given a state $s$,

$$L_{\pi_{\text{old}}}^{PFO}(\boldsymbol{\theta}) = \mathbb{E}_{\pi_{\text{old}}}\left[\sum_{t=0}^{t_{\max}-1}\|\phi_{\boldsymbol{\theta}}(S_t) - \phi_{\pi_{\text{old}}}(S_t)\|_2^2\right]. \tag{2}$$

We apply two versions of PFO: one on only the penultimate layer's pre-activations and one on all the pre-activations until the penultimate layer. In the scope of this work, we do not tune the coefficient of PFO; we pick the closest power of 10 that sets the magnitude of this loss to a similar magnitude of the clipped PPO objective tracked on the experiments without intervention. This gives a coefficient of 1 for ALE, 1 for MuJoCo with tanh, and 10 with ReLU. The goal is not necessarily to obtain better performance but to see if PFO improves the representations learned by PPO and if, in turn, it affects its trust region and performance. As shown in Figure 6, the regularization of PFO effectively brings the norm of the preactivation down, the number of dead neurons down, the capacity loss down, and the rank up. This coincides with a significant decrease in the excess probability ratio, especially in the upper tail. More importantly, we also see a significant increase in the lower tail of the returns where no collapse in performance is observed anymore on ALE/NameThisGame and ALE/Phoenix, with a

slight increase in the upper tail showing that PFO can increase performance. Among the interventions we have tried, PFO provided the most consistent improvements in representation and trust region.

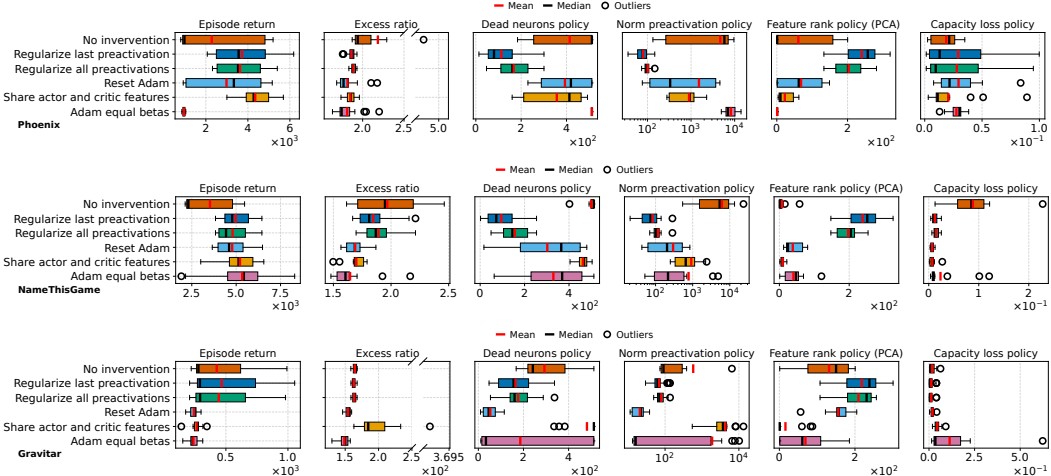

Figure 6: **Effects of regularizing features and non-stationarity** *Top & Middle: ALE/Phoenix-v5 & ALE/NameThisGame-v5.* Regularizing the difference between the features of consecutive policies with PFO results in better representations, a lower trust-region excess, and mitigates performance collapse. The same applies to sharing the actor-critic trunk. *Bottom: ALE/Gravitar.* Sharing the feature trunk between the actor and the critic results in a worse policy representation as the value network is subject to rank collapse due to reward sparsity. A boxplot includes 15 runs with different epochs.

**Sharing the actor-critic trunk** In deep RL, the decision to use the same feature network trunk for both the actor and the critic is not trivial. Depending on the complexity of the environment, it can significantly change the performance of a PPO agent (Andrychowicz et al., 2021; Huang et al., 2022a). We, therefore, attempt to draw a connection between sharing the feature trunk, the resulting representation, and its effects on the PPO objective. In this intervention, we make the actor and the critic share all the layers except their respective output layers and backpropagate the gradients from both the value and policy losses to the shared trunk. Figure 6 shows that the value loss acts as a regularizer, which decreases the feature rank and, depending on the reward's sparsity, gives two distinct effects. In dense-reward environments such as ALE/Phoenix and ALE/NameThisGame, the ranks are concentrated at low but non-trivial values: the upper tail significantly decreases compared to the baselines while the lower tail increases. This coincides with a lower feature norm, lower excess probability ratio, and, in turn, a high tail for the returns. It also increases performance in some cases. However, the opposite is true in the sparse-reward environment Gravitar: the rank completely collapses, and the feature norms and excess ratios are very high, collapsing the model's performance. This is consistent with the observations made in the plasticity works studying value-based methods: they show that sparse rewards deteriorate the rank of the value network, and we show that when shared in an actor-critic architecture they, in turn, deteriorate the policy. It is important to note that this distinction using the reward sparsity holds when comparing environments from the same family (e.g., ALE), but may not hold otherwise (e.g., comparing an ALE and a MuJoCo environment). We provide training curves showing the difference in the evolution of the feature rank when sharing the actor-critic trunk in Appendix D. To further strengthen this observation we run an intervention on ALE/Phoenix (a dense reward environment), with a reward mask randomly masking a reward with 90% chance, comparing the effects of sharing the actor-critic trunk. As expected, while with dense rewards, sharing the trunk is beneficial in ALE/Phoenix (Appendix Figure 21), with the sparse reward, the opposite is true: sharing the trunk is detrimental (Appendix Figure 35).

**Adapting Adam** Asadi et al. (2023) argue that as the targets of the value function change with the changing policy rollouts, the old moments accumulated by Adam become harmful to fit the new targets and find that resetting the moments of Adam helps performance in DQN-like algorithms. As the PPO objective creates a dependency on the previous policy, and more generally, in the policy gradient, the advantages change with the policy, the same argument about Adam moments can be made for PPO. Furthermore, Dohare et al. (2023b); Lyle et al. (2023) advocate for decaying the second moment of Adam faster than its default decay of 0.999 when training under non-stationarity

and set it to match the decay of the first moment. Therefore, we experiment with both resetting Adam's moments after each batch collection (to avoid tuning its frequency) and setting the second moment to decay at the (smaller) default decay of the first moment for both the actor and the critic; the moments are thus only accumulated over the epochs on the same batch in the former and over shorter batch sequences in the latter. We observe in Figure 6 and Appendix D that these interventions reduce the feature norm and increase the feature rank on ALE, which also reduces the excess probability ratio and, in some cases, improves performance; however, they are not sufficient to prevent collapse and, like sharing the actor-critic trunk, result in poor performance on ALE/Gravitar.

## 5    Related Work

Our work is complementary to various other works studying the plasticity and representation dynamics of neural networks trained under non-stationarity. Kumar et al. (2023) provide a comprehensive comparison and categorization of methods used to mitigate plasticity loss in continual supervised learning tasks and their effects on representations. Our work provides insights into the transferability of some of these solutions to RL and tools to evaluate their impact on trust region methods. Sokar et al. (2023) provide an alternative characterization of plasticity loss in RL using dormant neurons and observe an increase in dormant neurons for non-stationary objectives. Abbas et al. (2023) study representation metrics such as feature norms and observe a decrease of the norm due to dying neurons. Like in the work of Lyle et al. (2022), both studies only include value-based methods. In this work, we study dead units and capacity loss as Lyle et al. (2022) and provide corroboration of the dying units phenomenon in policy optimization methods and, taking the dying neurons out of the equation, find that the norm of preactivations actually blows up.

Other feature regularizations similar to PFO have been studied in value-based offline RL. Kumar et al. (2022) propose DR3, which counteracts an implicit regularization in TD learning by minimizing the dot product between the features of the estimated and target states. Ma et al. (2023) propose Representation Distinction (RD) which tries to avoid unwanted generalization by minimizing the dot product between the features of state-action pairs sampled from the learned policy and those sampled from the dataset or an OOD policy. Both are related to PFO as the methods directly tackle an undesired feature learning dynamic, but there is no motivation for DR3 or RD in online RL, and PFO is conceptually different. The implicit regularization that DR3 counteracts is not present in on-policy RL as shown by Kumar et al. (2022) in the SARSA experiment, and PFO differs from DR3 as it extends a trust region rather than counteracts an implicit bias. Similarly, the overestimation studied by Ma et al. (2023) in the vicious backup-generalization cycle is broken by on-policy data, and RD regularizes state features between the learned policy and the dataset policy, not consecutive policies.

## 6    Conclusion and Discussion

**Conclusion**  In this work, we provide evidence that the representation deterioration under non-stationarity observed by previous work in value-based methods generalizes to PPO agents in ALE and MuJoCo with their common model architectures and is connected to performance collapse. This brings a novel perspective to previous works that showed that PPO agents lose plasticity throughout training. We show that this is particularly concerning for the heuristic trust region set by PPO-Clip, which fails to prevent collapse as it becomes less effective when the agent's representation becomes poor. Finally, we present Proximal Feature Optimization (PFO), a simple novel auxiliary loss based on regularizing the evolution of features that mitigates representation degradation and, along with other interventions, shows that controlling representation mitigates performance collapse.

**Limitations and open questions**  In this work, we study the common architecture and optimizer of PPO agents in ALE and MuJoCo consisting of relatively small models without normalization layers, weight decay, or memory (e.g., not using Transformers and RNNs). Despite our best attempts, as with any other empirical machine learning work, the generalization of our results to other settings is not fully known. Still, this work should raise awareness about the representation collapse phenomenon observed in PPO and encourage future work to monitor representations when training PPO agents, as it can help diagnose performance collapse. We have focused on simple interventions that regularize non-stationarity and representations to highlight the effects of non-stationarity and the connection between representation, trust region, and collapse, but exploring interventions on plasticity is also valuable, as these may also influence the same dynamics. We believe further studies to analyze this problem, both empirically and particularly theoretically, to understand the reasons driving representation deterioration to be valuable. We hope that our study encourages work in this direction.

## Acknowledgments and Disclosure of Funding

We extend our gratitude to the reviewers for their valuable insights, which significantly enhanced the clarity and rigor of this work. We are particularly grateful to the area chair for their guidance which shaped the final version of this paper. We thank the SCITAS team at EPFL for the access to the beta testing phase of their new cluster. We are also grateful to Vincent Moens for his support with the TorchRL library. Finally, we thank `nimble.ai` for their generous gift to the CLAIRE Lab, which supported D.P.'s Master's project.

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

# A  Additional background

## A.1  Reinforcement Learning

The undiscounted formulation presented in the background (Section 2) has also been used by Schulman et al. (2015b) and does not limit the use of a discount factor to discount future rewards; for that purpose, as we consider a finite-horizon setting, we can assume that discounting would already be present in the reward which depends on time through the state. This allows to isolate the discount factor $\gamma$ for the purpose of the value estimation with GAE which serves as a trade-off between the bias and the variance in the estimator, in addition to $\lambda$ used for the $\lambda$-returns that combine multiple $n$-step returns. More importantly, this also allows us to reuse the policy gradient and PPO losses without discount factors, as the deep RL community is used to them while avoiding their incorrect use in the discounted setting as pointed out by Nota & Thomas (2020). In any case, our results can also be translated to the discounted setting using a biased gradient estimator (missing a discount factor), being the typical setting considered in deep RL works.

# B  Experiment details

## B.1  Code and run histories

Our codebase is publicly available at `https://github.com/CLAIRE-Labo/no-representation-no-trust`. It includes the development environment distributed as a Docker image for GPU-accelerated machines and a Conda environment for MPS-accelerated machines, the training code, scripts to run all the experiments, and the notebook that generated the plots. The codebase uses TorchRL (Bou et al., 2024) and provides a comprehensive toolbox to study representation dynamics in policy optimization. We also provide modified scripts of CleanRL (Huang et al., 2022b) to replicate the collapse observed in this work and ensure it is not a bug from our novel codebase.

The code repository contains links to the Weights&Biases (W&B) project with all of our run histories, a summary W&B report of the runs, and a W&B report with the replication with CleanRL.

Runs are fully reproducible on the same acceleration device on which they were run. In particular, we have reproduced our results on three different clusters with the same NVIDIA GPU device.

## B.2  Additional details on our experimental setup

We conduct experiments on an environment with pixel-based observations and discrete actions and an environment with continuous observations and actions, each with a different model architecture. For the discrete action case, we use the Arcade Learning Environment (ALE)(Bellemare et al., 2013) with the specification recommended by Machado et al. (2018) in v5 on Gymnasium (Towers et al., 2023). That is, with a sticky action probability of 0.25 as the only form of environment stochasticity, using only the game-over signal for termination (as opposed to end-of-life signals) with the default maximum of $108 \times 10^3$ environment frames per episode and reporting performance over training episodes (i.e., with sampling according to the policy distribution as opposed to taking the mode action). We train all models for 100 million environment frames. We use standard algorithmic choices to make our setting and results relevant to previous work. This includes taking only the sign of rewards (clipping) and frame skipping. We use a frame skip of 3, as opposed to the standard value of 4, due to limitations in the ALE-v5 environment, which does not implement frame pooling.[11] We use the standard architecture of Mnih et al. (2015) consisting of convolutional layers followed by linear layers, all with ReLU activations, and no normalization layers. We also use Mnih et al. (2015)'s standard observation transformations with a resizing to 84x84, grayscaling, and a frame stacking of 4.

For the continuous case, we use MuJoCo (Todorov et al., 2012) with v4 on Gymnasium (Towers et al., 2023) with the default maximum of 1,000 environment frames to mark episode termination. Similarly to Atari, we report performance as the average episode return over training episodes. We

---

[11]That is taking the max over the last two skipped and unskipped frames to capture elements that only appear in even or odd frames of the game (`https://github.com/Farama-Foundation/Arcade-Learning-Environment/issues/467`). Using an odd frame skip value alleviates the issue.

train all models for 5 million environment frames. We standardize the observations (subtract mean and divide by standard deviation) according to an initial rollout of 4,000 environment steps (at least four episodes). The standardization parameters are kept the same to avoid adding non-stationarity. We use the same architecture as Schulman et al. (2017), with only linear layers, tanh activations, and no normalization layers. We also experiment with ReLU activations. The network outputs a mean and a standard deviation (with softplus), both conditioning on the observation independently for each action dimension, which are then used to create a TanhNormal distribution, similarly to Haarnoja et al. (2018).

To measure the capacity loss of a checkpoint, we use the same optimization hyperparameters used to train the checkpoint, i.e. the same batch size and learning rate. The optimizer is reconstructed from its initial state (loading the optimizer state is also a valid design choice). The dataset sizes and fitting budgets for capacity are listed in Tables 2 and 3.

We provide a high-level pseudocode for PPO in Algorithm 1 and list all hyperparameters considered in Tables 2 and 3.

---

**Algorithm 1** High-level Pseudocode for PPO

$N$: number of environments in parallel.
$B_{\mathrm{env}}$: agent steps per environment to collect in a batch.
$K$: number of optimization epochs per batch.

$L_{\pi_{\mathrm{old}}}^{\mathrm{CLIP}}(\boldsymbol{\theta})$: PPO-Clip objective.
$H(\boldsymbol{\theta})$: entropy bonus/loss; $c_H$: entropy bonus coefficient.
$L^{VF}(\mathbf{w})$: critic loss ($L^2$ to GAE); $c_{VF}$: critic loss coefficient.

1: **while** collected environment steps $\leq$ total environment steps **do**
   Collect a batch of interaction steps of size $B = N \times B_{\mathrm{env}}$ and computes advantages.
2:    **for** actor $= 1$ **to** $N$ **do**
3:        Run policy $\pi_{\mathrm{old}}$ in environment for $B_{\mathrm{env}}$ agent steps.
4:        Compute advantage estimates $\Psi_1^{\mathrm{actor}}, \ldots, \Psi_{B_{\mathrm{env}}}^{\mathrm{actor}}$ with GAE.
5:    **end for**
6:    Minimize overall policy and value loss $(-L_{\pi_{\mathrm{old}}}^{\mathrm{CLIP}}(\boldsymbol{\theta}) - c_H H(\boldsymbol{\theta}) + c_{VF} L^{VF}(\mathbf{w}))$ with autograd on the collected batch over $K$ epochs with minibatch size $M \leq B$.
7:        $\pi_{\mathrm{old}} \leftarrow \pi_{\boldsymbol{\theta}}$
8: **end while**

---

**Proximal Feature Regularization** With a coefficient $c_{PFO}$, the PFO loss is added to the overall loss $(-L_{\pi_{\mathrm{old}}}^{\mathrm{CLIP}}(\boldsymbol{\theta}) + c_{PFO} L_{\pi_{\mathrm{old}}}^{PFO}(\boldsymbol{\theta}) - c_H H(\boldsymbol{\theta}) + c_{VF} L^{VF}(\mathbf{w}))$ optimized with autograd over multiple minibatch epochs.

### B.3   Additional details on metrics used in the figures

**Training curves** A point in the training curves in Figures 1, 2, 3, before aggregating seeds and smoothing, corresponds to an average value over the last batch collected at the time of logging for metrics available at every batch (feature rank, entropy, etc.) or the latest batch where the metric was available at the time of logging for the episodic return (as it's only available when episodes finish, and it requires multiple batches to finish an episode). E.g., in Figure 1 on ALE, a feature rank corresponds to the average feature rank over all the states in the last batch collected at the time of logging and is logged every 0.1% of the batches (i.e. every 6,144 env steps); A return corresponds to the average return across all workers that had episodes finished in the latest batch containing finished episodes at the time of logging.

**Figure 4** A window of size 1% of training progress represents approximately 1 million training steps on ALE and 50,000 training steps on MuJoCo We average the metrics per window and then take the 20 windows with the lowest average probability ratios below $1 - \epsilon$. The probability ratios in a run can be trivially within the $1 - \epsilon$ region after the model collapses, resulting in less than 20 points if the model collapses before 20% of the training progress. When all runs give 20 points, we can observe 300 points in total per scatter plot.

Table 1: Hyperparameters for the toy setting in Figure 5.

| **Environment** | |
|---|---|
| $\phi(x)$ | Sampled from a Normal distribution |
| $\phi(y)$ | $\alpha\phi(x)$ |
| $\alpha$ | 3 (overshoot), -1 (interfere) |
| $A(x, a_1), A(y, a_1)$ | 1 |

| **Policy** | |
|---|---|
| Network | 2 output neurons representing the 2 logits + Softmax |

| **Optimization** | |
|---|---|
| Clipping epsilon (PPO-Clip) | 0.1 |
| Optimizer | SGD |
| Learning rate | 1.5 |
| Minibatch size | 1 |
| Number of epochs | 10 |
| Number of steps | 20 alternating between $x$ and $y$ |

**Figure 6** A window of size 5% of training progress represents approximately five million training steps in ALE and captures at least five episodes per environment so in total at least 40 episodes. For MuJoCo this represents approximately 256,000 training steps and captures at least 128 episodes per environment so in total at least 256 episodes.

When a model collapses, it typically doesn't change anymore so its optimization trivially gives ratios within the clipping limits (no value above $1 + \epsilon$ and below $1 - \epsilon$ is logged). In that case, we are more interested in the evolution of the excess ratio before the ratios become trivial. Therefore, the upper limit of the 5% of training progress is taken such that it is the latest timestep where there are at least 10 non-trivial ratios, i.e. 10 logged excess ratios. This coincides with a window before the collapse of the model capturing the values we are interested in. Note that when a model collapses this window may not coincide with the window used to report the other metrics such as the average return, however, these other metrics typically do not change after a collapse, so it is more robust to capture them at the end of training rather than looking for an arbitrary window after the collapse. We give training curves similar to Figure 1 with the interventions performed.

In MuJoCo, with continuous action distributions the ratios diverge to infinity and 0 before collapse therefore to get meaningful plots, we clip average probability ratios above $1 + \epsilon$ and below $1 - \epsilon$ to $10^{12}$ and $10^{-12}$, respectively, before computing the average excess ratio.

We group the different epoch configurations of an intervention on the same environment, giving 15 runs per boxplot (three epochs with five seeds each). The right (resp. left) whiskers are determined by the highest (resp. lowest) observed datapoint below Q3 + 1.5 IQR (default of Matplotlib). The outliers are points outside of the whiskers.

## B.4  Statistical significance

Stochasticity in our experiments arises from network initialization, environment transitions (e.g., sticky actions in ALE), agent action sampling, and minibatch sampling for optimization. A seed fully controls the sequence of randomness in a run with the same hyperparameter configuration. We repeat each configuration with five seeds using the same collection of seeds, resulting in the same initialization of the networks and environments for a given seed across configurations. This form of repeated measures allows us to compare the configurations with lower variance as they share the same initial conditions, hence requiring a lower number of seeds.

In Figures 1 and 2, we aggregate the five runs of each experiment into mean curves with min/max shaded areas. The use of min/max error bars allows us to demonstrate the full range of observed outcomes, although it may result in shaded areas that overlap more than with other types of error bars. Most of the claims we make based on those figures do not rely on non-overlapping shaded areas and are instead stronger when the max or min boundaries are consistent with the observation made

(min boundary of feature norm increasing, max boundary of feature rank decreasing). Otherwise, we made comparative claims when shaded areas did not overlap (feature rank decreasing faster with more epochs and more non-stationarity).

Figure 3 displays individual seeds to zoom on single-run dynamics around collapse. It is used for an illustrative purpose to provide intuition and does not depend on the number of runs or statistical aggregation of results. The main claim made with the intuition (breakdown of the trust region) is backed by Figure 4, which includes 300 points per plot per environment, subsampled from 15 training curves per environment.

To evaluate the effects of the interventions in Figure 6, we show boxplots to give a complete idea of the distribution of the data which is formed by grouping the different configurations in the same environment. Each boxplot contains 15 runs. We make claims such as preventing collapse using the tails and medians and claims about lower excess ratio and higher rank using the interquartile range. Without a clear intuition about the distribution of combined configurations per environment, we consider this approach appropriate for comparing interventions.

In summary, we believe our experimental design provides a balanced tradeoff between statistical significance and richness of claims. The computational cost of running more seeds may not yield proportionately valuable insights.

## B.5 Hardware and runtime

The experiments in this project took a total of ~11,300 GPU hours on NVIDIA V100 and A100 GPUs (ALE) and ~25,500 CPU hours (MuJoCo). A run on ALE takes around 10 hours on an A100 and 16 hours on a V100. A run on MuJoCo takes around 5 hours on 6 CPUs.

Table 2: Hyperparameters for ALE.

### Environment

| | |
|---|---|
| Repeat action probability (Sticky actions) | 0.25 |
| Frameskip | 3 |
| Max environment steps per episode | 108,000 |
| Noop reset steps | 0 |

### Observation transforms

| | |
|---|---|
| Grayscale | True |
| Resize width ('resize_w') | 84 |
| Resize height ('resize_h') | 84 |
| Frame stack | 4 |
| Normalize observations | False |

### Reward transforms

| | |
|---|---|
| Sign | True |

### Collector

| | |
|---|---|
| Total environment steps | 100,000,000 |
| Num envs in parallel | 8 |
| Num envs in parallel capacity | 1 |
| Agent steps per batch | 10,24 (128 per env) |
| Total agent steps capacity | 36,000 (at least one full episode) |

### Models (actor and critic)

| | |
|---|---|
| Activation | ReLU |
| **Convolutional Layers** | |
| Filters | [32, 64, 64] |
| Kernel sizes | [8, 4, 3] |
| Strides | [4, 2, 1] |
| **Linear Layers** | |
| Number of layers | 1 |
| Layer size | 512 |

### Optimization

| | |
|---|---|
| **Advantage estimator** | |
| Advantage estimator | GAE |
| Gamma | 0.99 |
| Lambda | 0.95 |
| **Value loss** | |
| Value loss coefficient | 0.5 |
| Loss type | L2 |
| **Policy loss** | |
| Normalize advantages | minibatch normalization |
| Clipping epsilon | 0.1 |
| Entropy coefficient | 0.01 |
| Feature regularization coefficient | 1 (last pre-activation), 10 (all pre-activations) |
| **Optimizer (actor and critic)** | |
| Optimizer | Adam |
| Learning rate | 0.00025 |
| Betas | (0.9, 0,999), (0.9, 0,9) for the intervention |
| Max grad norm | 0.5 |
| Annealing linearly | False |
| Number of epochs | 4, 6, 8 |
| Number of epochs capacity fit | 1 |
| Minibatch size | 256 |

### Logging (% of the total number of batches)

| | |
|---|---|
| Training | every 0.1% (~100,000 env steps) |
| Capacity | every 2.5% (41 times in total) |

Table 3: Hyperparameters for MuJoCo.

| **Environment** | |
|---|---|
| Frameskip | 1 |
| Max env steps per episode | 1,000 |
| Noop reset steps | 0 |

| **Observation transforms** | |
|---|---|
| Normalize observations | True (from initial steps collected by uniform policy) |
| Initial random steps for normalization | 4000 (at least 4 episodes) |

| **Collector** | |
|---|---|
| Total environment steps | 5,000,000 |
| Num envs in parallel | 2 |
| Num envs in parallel capacity | 4 |
| Agent steps per batch | 2048 (1024 per env) |
| Total environment steps capacity | 4,000 (at least 4 full episodes) |

| **Models (actor and critic)** | |
|---|---|
| Activation | Tanh, ReLU |
| **Convolutional layers** | |
| Number of Layers | 0 |
| **Linear layers** | |
| Number of layers | 2 |
| Layer size | 64 |

| **Optimization** | |
|---|---|
| **Advantage estimator** | |
| Advantage estimator | GAE |
| Gamma | 0.99 |
| Lambda | 0.95 |
| **Value loss** | |
| Value coefficient | 0.5 |
| Loss type | L2 |
| **Policy loss** | |
| Normalize advantages | minibatch normalization |
| Clipping epsilon (PPO-Clip) | 0.2 |
| Entropy coefficient | 0.0 |
| Feature regularization coefficient | 1 (tanh), 10 (ReLU) |
| **Optimizer (actor and critic)** | |
| Optimizer | Adam |
| Learning rate | 0.0003 |
| Betas | (0.9, 0.999), (0.9, 0.9) for the intervention |
| Max grad norm | 0.5 |
| Annealing linearly | False |
| Number of epochs | 10, 15, 20 |
| Number of epochs capacity fit | 4 |
| Minibatch size | 64 |

| **Logging (% of the total number of batches)** | |
|---|---|
| Training | every 0.1% (6,144 env steps) |
| Capacity | every 2.5% (41 times in total) |

## C Toy setting derivation details

The derivatives of the softmax probability $\pi_{\boldsymbol{\theta}}(a_1|s)$ with respect to $\theta_1$ and $\theta_2$ are as follows:

$$\frac{\partial \pi_{\boldsymbol{\theta}}(a_1|s)}{\partial \theta_1} = \frac{\partial}{\partial \theta_1}\left(\frac{e^{\theta_1 \phi(s)}}{e^{\theta_1 \phi(s)} + e^{\theta_2 \phi(s)}}\right) = \phi(s) \cdot \frac{e^{\theta_1 \phi(s)} \cdot e^{\theta_2 \phi(s)}}{(e^{\theta_1 \phi(s)} + e^{\theta_2 \phi(s)})^2} \tag{3}$$

$$\frac{\partial \pi_{\boldsymbol{\theta}}(a_1|s)}{\partial \theta_2} = \frac{\partial}{\partial \theta_2}\left(\frac{e^{\theta_1 \phi(s)}}{e^{\theta_1 \phi(s)} + e^{\theta_2 \phi(s)}}\right) = -\phi(s) \cdot \frac{e^{\theta_1 \phi(s)} \cdot e^{\theta_2 \phi(s)}}{(e^{\theta_1 \phi(s)} + e^{\theta_2 \phi(s)})^2} \tag{4}$$

The update rule for each parameter $\theta_i$ in $\theta$ with SGD is $\theta_i^{\text{new}} = \theta_i + \eta \frac{\partial L}{\partial \theta_i}$ where $\eta$ is the learning rate. Therefore, given the partial derivatives, the updated values for $\theta_1$ and $\theta_2$ after taking a gradient step are (if the probability is still inferior to $1 + \epsilon$, otherwise the gradient is 0)

$$\theta_1^{\text{new}} = \theta_1 + \eta \cdot \frac{A(s,a_1)}{\pi_{\text{old}}(a_i|s)} \cdot \left(\phi(s) \cdot \frac{e^{\theta_1 \phi(s)} \cdot e^{\theta_2 \phi(s)}}{(e^{\theta_1 \phi(s)} + e^{\theta_2 s})^2}\right) \quad \text{and} \quad \theta_2^{\text{new}} = \theta_2 - \eta \cdot \frac{A(s,a_1)}{\pi_{\text{old}}(a_i|s)} \cdot \left(\phi(s) \cdot \frac{e^{\theta_1 \phi(s)} \cdot e^{\theta_2 \phi(s)}}{(e^{\theta_1 \phi(s)} + e^{\theta_2 \phi(s)})^2}\right)$$

Hence,

$$\theta_1^{\text{new}} = \theta_1 + \delta_s \quad \text{with } \delta_s = \eta \cdot \frac{A(s,a_1)}{\pi_{\text{old}}(a_i|s)} \cdot \left(\phi(s) \cdot \frac{e^{\theta_1 \phi(s)} \cdot e^{\theta_2 \phi(s)}}{(e^{\theta_1 \phi(s)} + e^{\theta_2 \phi(s)})^2}\right)$$

$$\theta_2^{\text{new}} = \theta_2 - \delta_s$$

Let $\alpha \geq 0$ and without loss of generality, let's take $\alpha \geq 1$. After a gradient step on $x$ one has

$$\begin{aligned}
\pi_{\boldsymbol{\theta}^{\text{new}}}(a_1|x) &= \frac{e^{\theta_1^{\text{new}} \phi(x)}}{e^{\theta_1^{\text{new}} \phi(x)} + e^{\theta_2^{\text{new}} \phi(x)}} \\
&= \frac{e^{(\theta_1 + \delta_x)\phi(x)}}{e^{(\theta_1 + \delta_x)\phi(x)} + e^{(\theta_2 - \delta_x)\phi(x)}} \\
&= \frac{e^{\theta_1 \phi(x)}}{e^{\theta_1 \phi(x)} + e^{(\theta_2 - 2\delta_x)\phi(x)}} \\
&= \frac{e^{\theta_1 \phi(x)}}{e^{\theta_1 \phi(x)} + e^{\theta_2 \phi(x) - 2\delta_x \phi(x)}} \\
&\geq \frac{e^{\theta_1 \phi(x)}}{e^{\theta_1 \phi(x)} + e^{\theta_2 \phi(x)}} \quad (\text{since } -2\delta_x \phi(x) \leq 0) \\
&= \pi_{\boldsymbol{\theta}}(a_1|x)
\end{aligned}$$

$$\begin{aligned}
\pi_{\boldsymbol{\theta}^{\text{new}}}(a_1|y) &= \frac{e^{\theta_1^{\text{new}} \alpha\phi(x)}}{e^{\theta_1^{\text{new}} \alpha\phi(x)} + e^{\theta_2^{\text{new}} \alpha\phi(x)}} \\
&= \frac{e^{(\theta_1 + \delta_x)\alpha\phi(x)}}{e^{(\theta_1 + \delta_x)\alpha\phi(x)} + e^{(\theta_2 - \delta_x)\alpha\phi(x)}} \\
&= \frac{e^{\theta_1 \alpha\phi(x)}}{e^{\theta_1 \alpha\phi(x)} + e^{(\theta_2 - 2\delta_x)\alpha\phi(x)}} \\
&= \frac{e^{\theta_1 \alpha\phi(x)}}{e^{\theta_1 \alpha\phi(x)} + e^{\theta_2 \alpha\phi(x) - 2\delta_x \alpha\phi(x)}} \\
&\geq \frac{e^{\theta_1 \alpha\phi(x)}}{e^{\theta_1 \alpha\phi(x)} + e^{\theta_2 \alpha\phi(x)}} \quad (\text{since } -2\delta_x \alpha\phi(x) \leq 0) \\
&= \pi_{\boldsymbol{\theta}}(a_1|y)
\end{aligned}$$

And after a gradient step on $y$:

$$\pi_{\boldsymbol{\theta}^{\text{new}}}(a_1|x) = \frac{e^{\theta_1^{\text{new}}\phi(x)}}{e^{\theta_1^{\text{new}}\phi(x)} + e^{\theta_2^{\text{new}}\phi(x)}}$$

$$= \frac{e^{(\theta_1+\delta_y)\phi(x)}}{e^{(\theta_1+\delta_y)\phi(x)} + e^{(\theta_2-\delta_y)\phi(x)}}$$

$$= \frac{e^{\theta_1\phi(x)}}{e^{\theta_1\phi(x)} + e^{(\theta_2-2\delta_y)\phi(x)}}$$

$$= \frac{e^{\theta_1\phi(x)}}{e^{\theta_1\phi(x)} + e^{\theta_2\phi(x)-2\delta_y\phi(x)}}$$

$$\geq \frac{e^{\theta_1\phi(x)}}{e^{\theta_1\phi(x)} + e^{\theta_2\phi(x)}} \quad (\text{since } -2\delta_y\phi(x) \leq 0)$$

$$= \pi_{\boldsymbol{\theta}}(a_1|x)$$

$$\pi_{\boldsymbol{\theta}^{\text{new}}}(a_1|y) = \frac{e^{\theta_1^{\text{new}}\alpha\phi(x)}}{e^{\theta_1^{\text{new}}\alpha\phi(x)} + e^{\theta_2^{\text{new}}\alpha\phi(x)}}$$

$$= \frac{e^{(\theta_1+\delta_y)\alpha\phi(x)}}{e^{(\theta_1+\delta_y)\alpha\phi(x)} + e^{(\theta_2-\delta_y)\alpha\phi(x)}}$$

$$= \frac{e^{\theta_1\alpha\phi(x)}}{e^{\theta_1\alpha\phi(x)} + e^{(\theta_2-2\delta_y)\alpha\phi(x)}}$$

$$= \frac{e^{\theta_1\alpha\phi(x)}}{e^{\theta_1\alpha\phi(x)} + e^{\theta_2\alpha\phi(x)-2\delta_y\alpha\phi(x)}}$$

$$\geq \frac{e^{\theta_1\alpha\phi(x)}}{e^{\theta_1\alpha\phi(x)} + e^{\theta_2\alpha\phi(x)}} \quad (\text{since } -2\delta_y\alpha\phi(x) \leq 0)$$

$$= \pi(a_1, \alpha x, \boldsymbol{\theta})$$

$$= \pi_{\boldsymbol{\theta}}(a_1|y)$$

Let $\alpha \leq 0$ and without loss of generality, let's take $\alpha \leq 1$, after a gradient step on $x$ one has

$$\pi_{\boldsymbol{\theta}^{\text{new}}}(a_1|x) = \frac{e^{\theta_1^{\text{new}}\phi(x)}}{e^{\theta_1^{\text{new}}\phi(x)} + e^{\theta_2^{\text{new}}\phi(x)}}$$

$$= \frac{e^{(\theta_1+\delta_x)\phi(x)}}{e^{(\theta_1+\delta_x)\phi(x)} + e^{(\theta_2-\delta_x)\phi(x)}}$$

$$= \frac{e^{\theta_1\phi(x)}}{e^{\theta_1\phi(x)} + e^{(\theta_2-2\delta_x)\phi(x)}}$$

$$= \frac{e^{\theta_1\phi(x)}}{e^{\theta_1\phi(x)} + e^{\theta_2\phi(x)-2\delta_x\phi(x)}}$$

$$\geq \frac{e^{\theta_1\phi(x)}}{e^{\theta_1\phi(x)} + e^{\theta_2\phi(x)}} \quad (\text{since } -2\delta_x\phi(x) \leq 0)$$

$$= \pi_{\boldsymbol{\theta}}(a_1|x)$$

$$
\begin{aligned}
\pi_{\boldsymbol{\theta}^{\text{new}}}(a_1, y) &= \frac{e^{\theta_1^{\text{new}}\alpha\phi(x)}}{e^{\theta_1^{\text{new}}\alpha\phi(x)} + e^{\theta_2^{\text{new}}\alpha\phi(x)}} \\
&= \frac{e^{(\theta_1+\delta_x)\alpha\phi(x)}}{e^{(\theta_1+\delta_x)\alpha\phi(x)} + e^{(\theta_2-\delta_x)\alpha\phi(x)}} \\
&= \frac{e^{\theta_1\alpha\phi(x)}}{e^{\theta_1\alpha\phi(x)} + e^{(\theta_2-2\delta_x)\alpha\phi(x)}} \\
&= \frac{e^{\theta_1\alpha\phi(x)}}{e^{\theta_1\alpha\phi(x)} + e^{\theta_2\alpha\phi(x)-2\delta_x\alpha\phi(x)}} \\
&\leq \frac{e^{\theta_1\alpha\phi(x)}}{e^{\theta_1\alpha\phi(x)} + e^{\theta_2\alpha\phi(x)}} \quad (\text{since } -2\delta_x\alpha\phi(x) \geq 0) \\
&= \pi_{\boldsymbol{\theta}}(a_1|y)
\end{aligned}
$$

And after a gradient step on $y$:

$$
\begin{aligned}
\pi_{\boldsymbol{\theta}^{\text{new}}}(a_1|x) &= \frac{e^{\theta_1^{\text{new}}\phi(x)}}{e^{\theta_1^{\text{new}}\phi(x)} + e^{\theta_2^{\text{new}}\phi(x)}} \\
&= \frac{e^{(\theta_1+\delta_y)\phi(x)}}{e^{(\theta_1+\delta_y)\phi(x)} + e^{(\theta_2-\delta_y)\phi(x)}} \\
&= \frac{e^{\theta_1\phi(x)}}{e^{\theta_1\phi(x)} + e^{(\theta_2-2\delta_y)\phi(x)}} \\
&= \frac{e^{\theta_1\phi(x)}}{e^{\theta_1\phi(x)} + e^{\theta_2\phi(x)-2\delta_y\phi(x)}} \\
&\leq \frac{e^{\theta_1\phi(x)}}{e^{\theta_1\phi(x)} + e^{\theta_2\phi(x)}} \quad (\text{since } -2\delta_y\phi(x) \geq 0) \\
&= \pi_{\boldsymbol{\theta}}(a_1|x)
\end{aligned}
$$

$$
\begin{aligned}
\pi_{\boldsymbol{\theta}^{\text{new}}}(a_1|y) &= \frac{e^{\theta_1^{\text{new}}\alpha\phi(x)}}{e^{\theta_1^{\text{new}}\alpha\phi(x)} + e^{\theta_2^{\text{new}}\alpha\phi(x)}} \\
&= \frac{e^{(\theta_1+\delta_y)\alpha\phi(x)}}{e^{(\theta_1+\delta_y)\alpha\phi(x)} + e^{(\theta_2-\delta_y)\alpha\phi(x)}} \\
&= \frac{e^{\theta_1\alpha\phi(x)}}{e^{\theta_1\alpha\phi(x)} + e^{(\theta_2-2\delta_y)\alpha\phi(x)}} \\
&= \frac{e^{\theta_1\alpha\phi(x)}}{e^{\theta_1\alpha\phi(x)} + e^{\theta_2\alpha\phi(x)-2\delta_y\alpha\phi(x)}} \\
&\geq \frac{e^{\theta_1\alpha\phi(x)}}{e^{\theta_1\alpha\phi(x)} + e^{\theta_2\alpha\phi(x)}} \quad (\text{since } -2\delta_y\alpha\phi(x) \leq 0) \\
&= \pi_{\boldsymbol{\theta}}(a_1|y)
\end{aligned}
$$

# D   Main paper figures on all environments

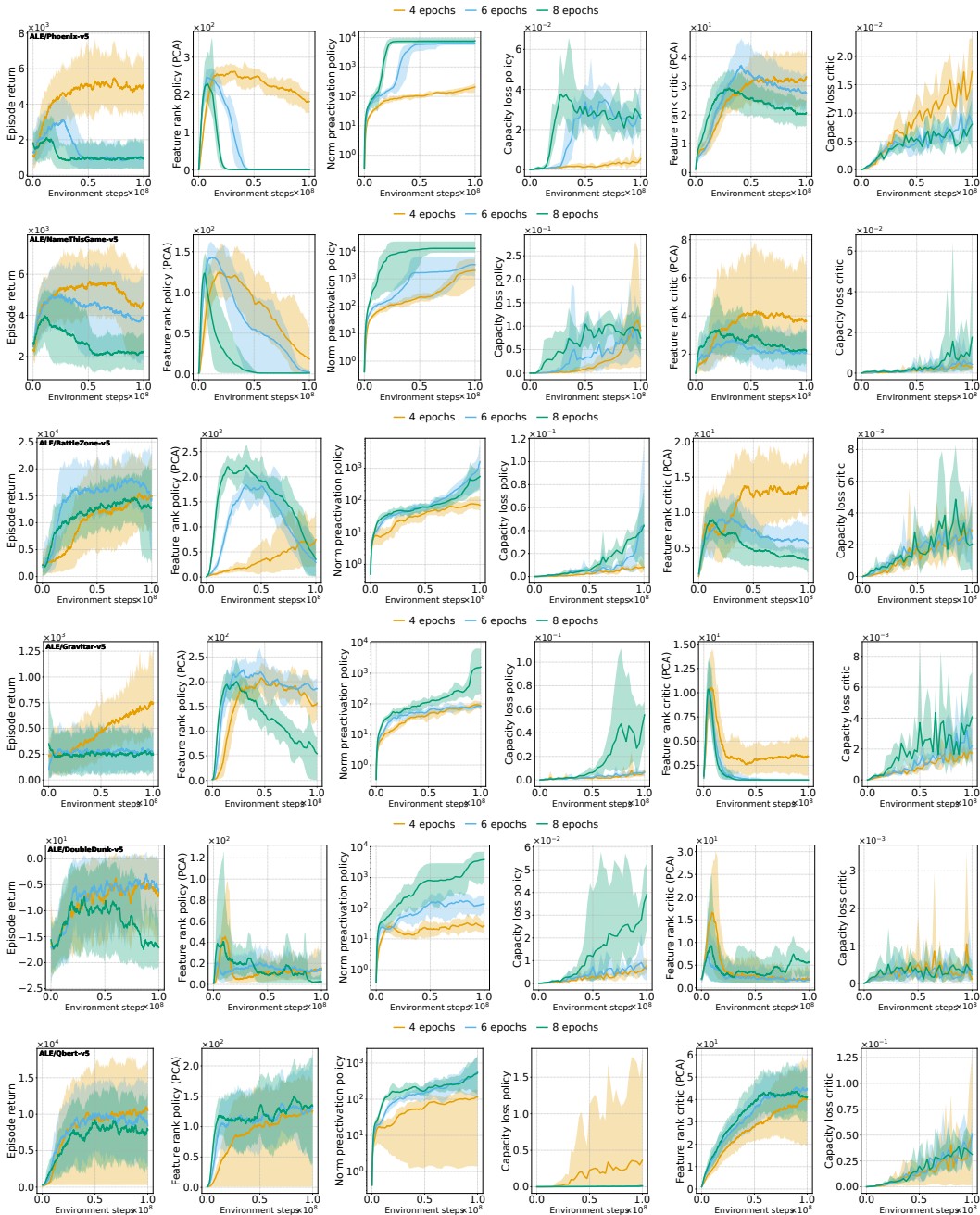

Figure 7: Figure 1 on ALE. QBert is the only game where rank decline and collapse are not observed, apart from an outlier run that collapsed at initialization. The performance of the policy should be taken into consideration when comparing the capacity loss of the critic. E.g., for Phoenix, the capacity loss of the critic associated with the policy that doesn't collapse ends up higher than that of the policies that do collapse.

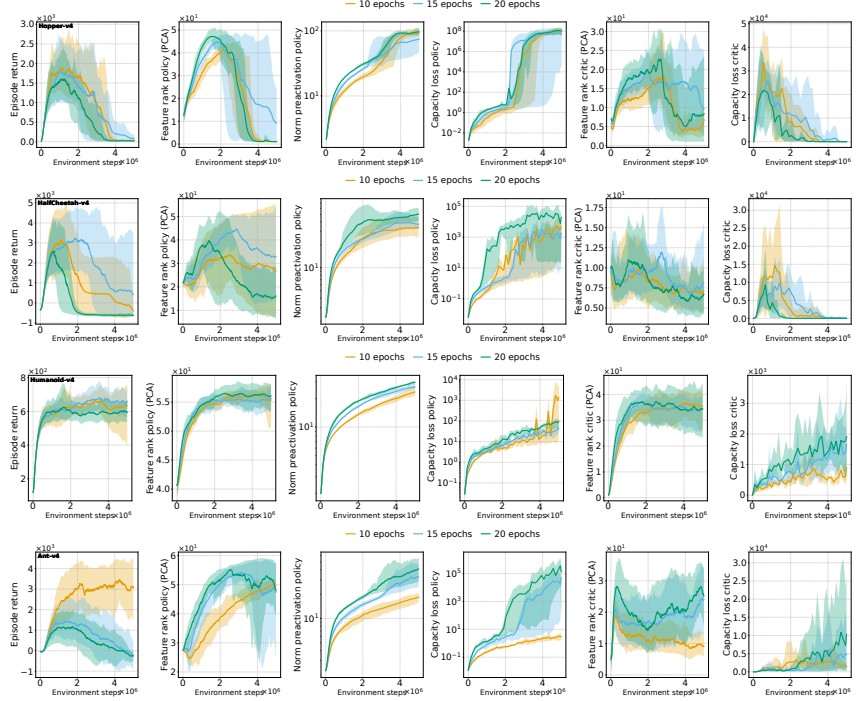

Figure 8: Figure 1 on MuJoCo with the tanh activation.

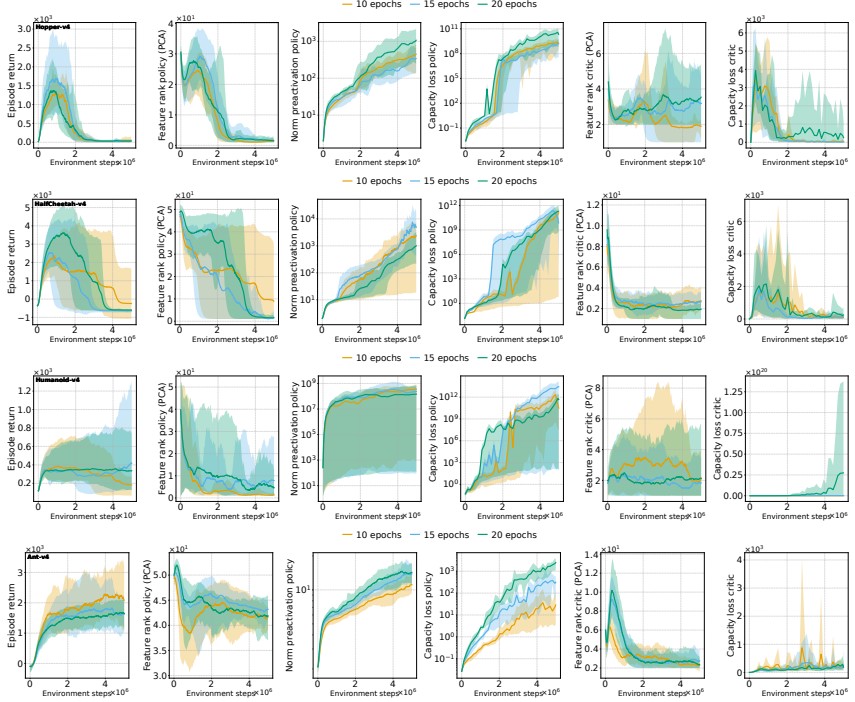

Figure 9: Figure 1 on MuJoCo with the ReLU activation.

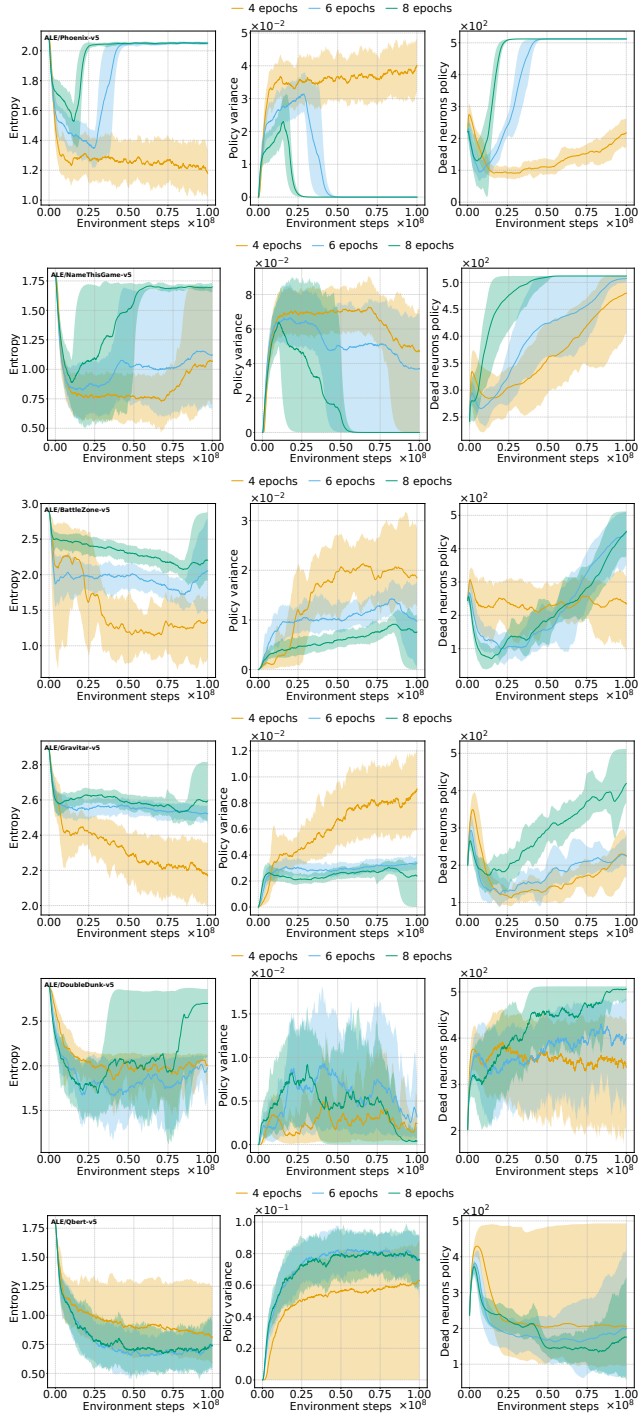

Figure 10: Figure 2 on ALE.

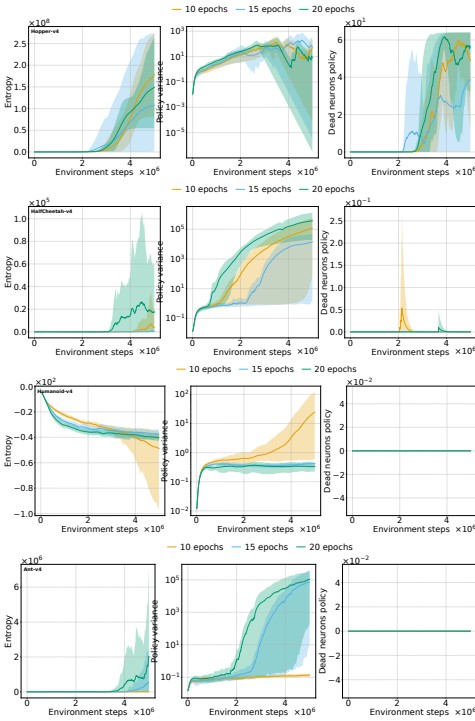
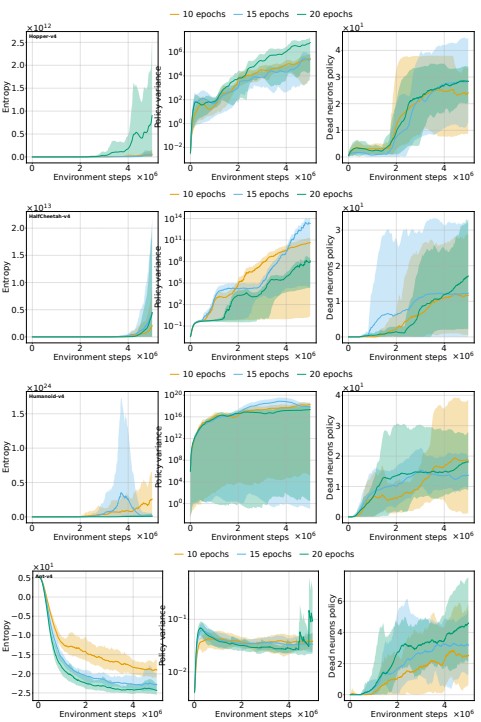

Figure 11: Figure 2 on MuJoCo with the tanh activation. With a continuous action distribution, the policy variance can either drop or explode. Dead neurons for the tanh activation are hard to compute as they are dependent on an arbitrary threshold.

Figure 12: Figure 2 on MuJoCo with the ReLU activation.

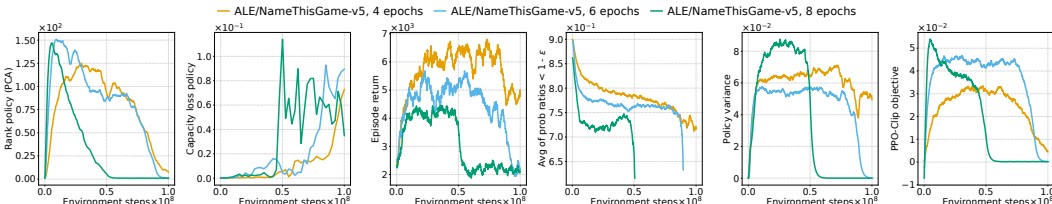

Figure 13: Figure 3 on ALE. (No other environments considered; same figure as Figure 3).

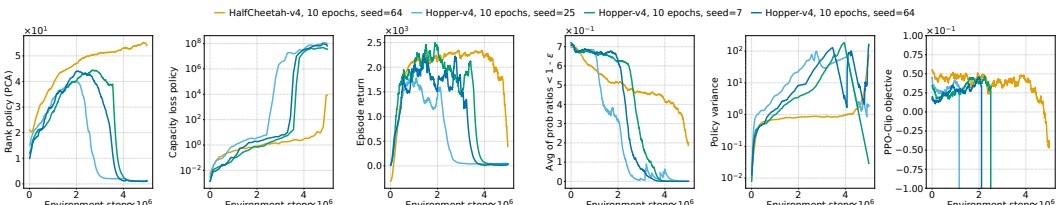

Figure 14: Figure 3 on MuJoCo with the tanh activation. The PPO-Clip objective explodes in the negative direction after collapse so we clip the y-axis of that plot to −1.

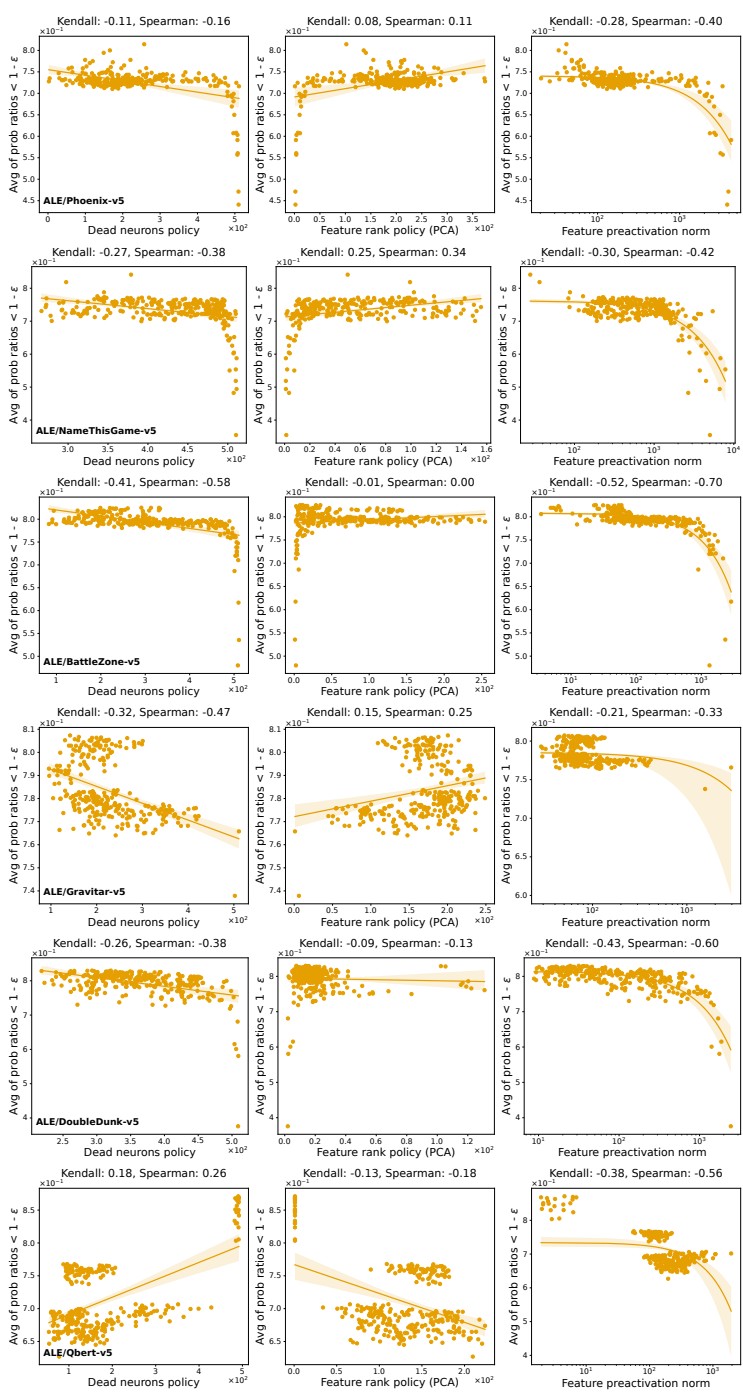

Figure 15: Figure 4 ALE. Qbert and Gravitar do not have runs with poor representation regions (dead neurons > 510) to exhibit the correlation around collapse. Qbert has one outlier where the agent collapsed at the very beginning of the training and kept a high (but lower than 510) number of dead neurons and a trivial rank, but a low excess ratio.

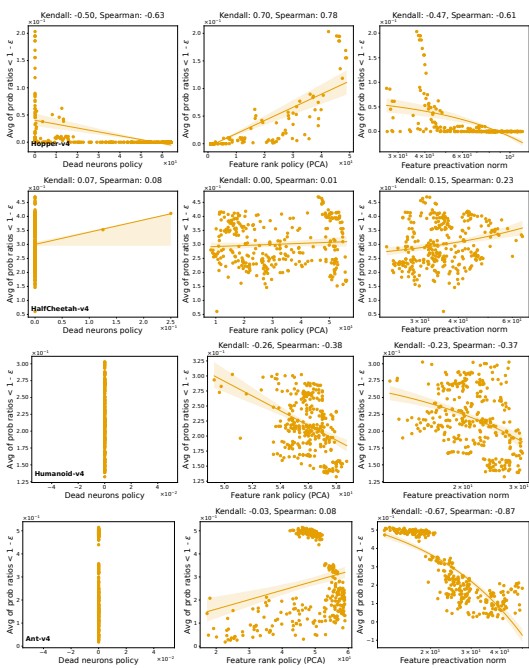

Figure 16: Figure 4 on MuJoCo with the tanh activation. Dead neurons for the tanh activation are hard to compute as they are dependent on an arbitrary threshold. In Humanoid the rank does not arrive at low values to exhibit the correlation around collapse.

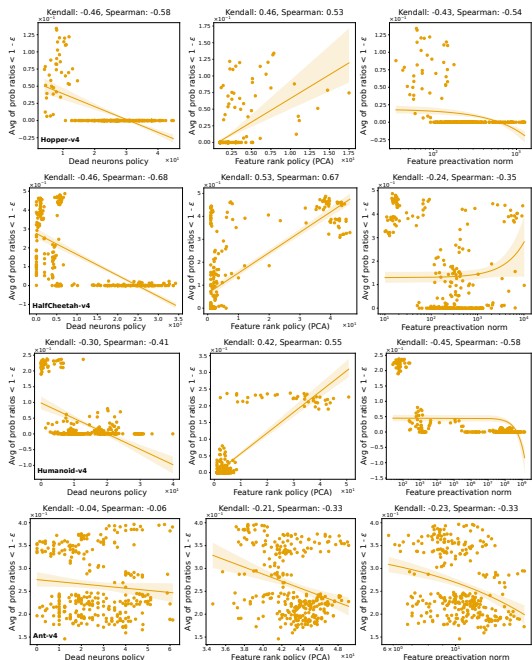

Figure 17: Figure 4 on MuJoCo with the ReLU activation. In Ant, the rank does not arrive at low values to exhibit the correlation around collapse.

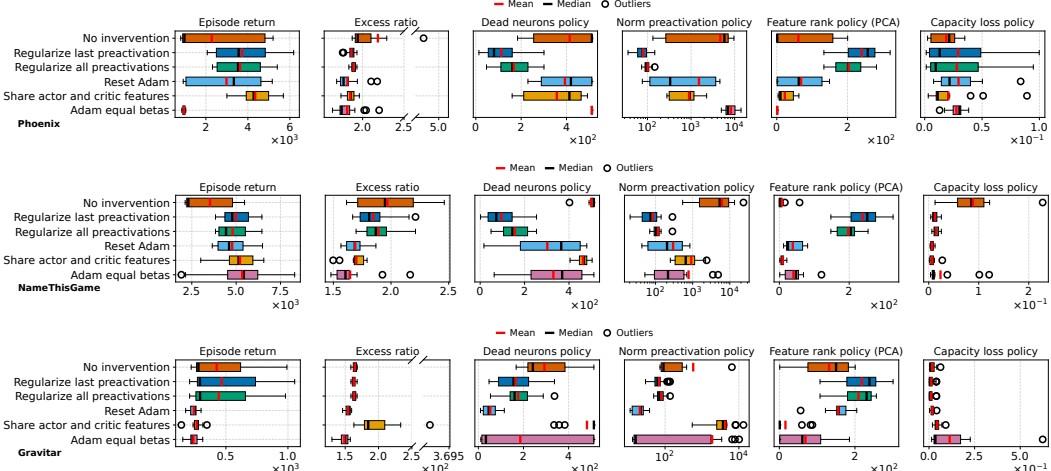

Figure 18: Figure 6 on ALE. The tails of the capacity loss on Phoenix with interventions can be higher than without interventions on the runs where the models collapse too early without interventions, leading to the capacity loss of the non-collapsed models with interventions eventually becoming higher. This can be observed from the training curves with interventions. Nevertheless, their medians are lower.

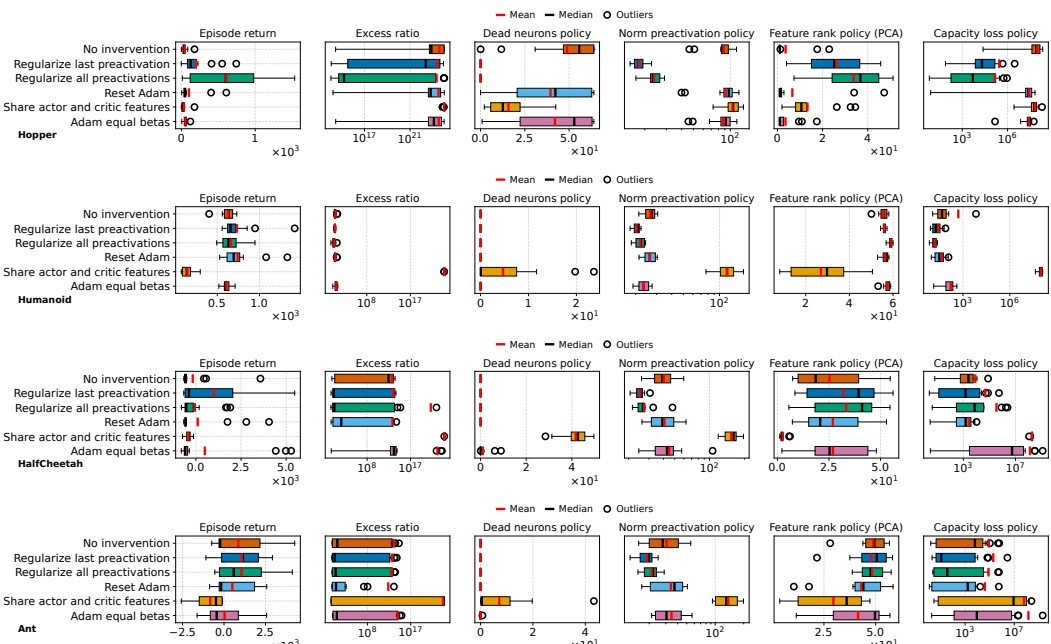

Figure 19: Figure 6 on MuJoCo with the tanh activation.

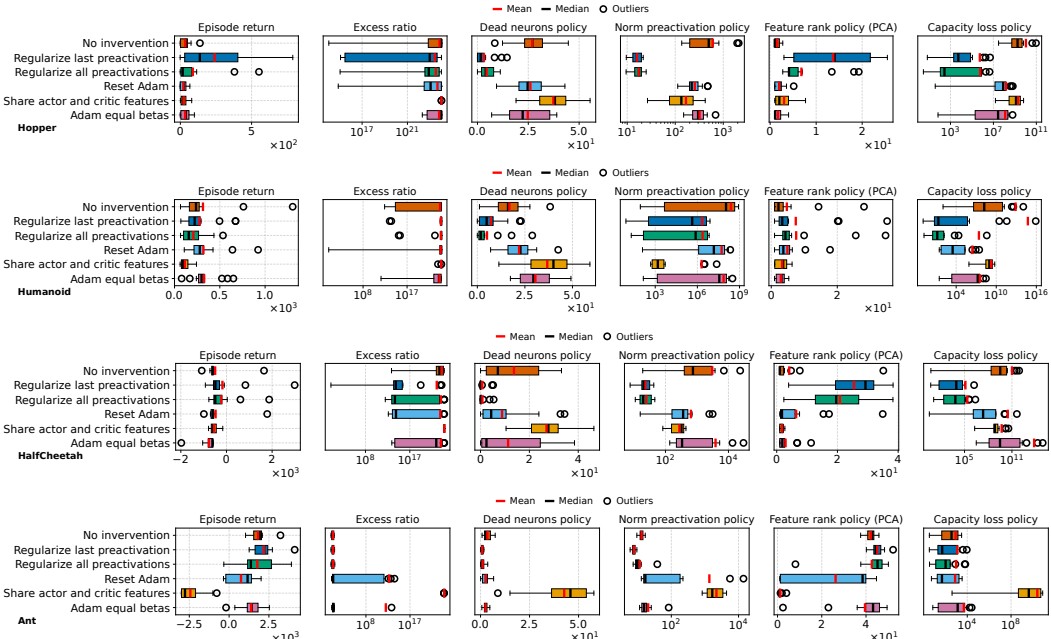

Figure 20: Figure 6 on MuJoCo with the ReLU activation.

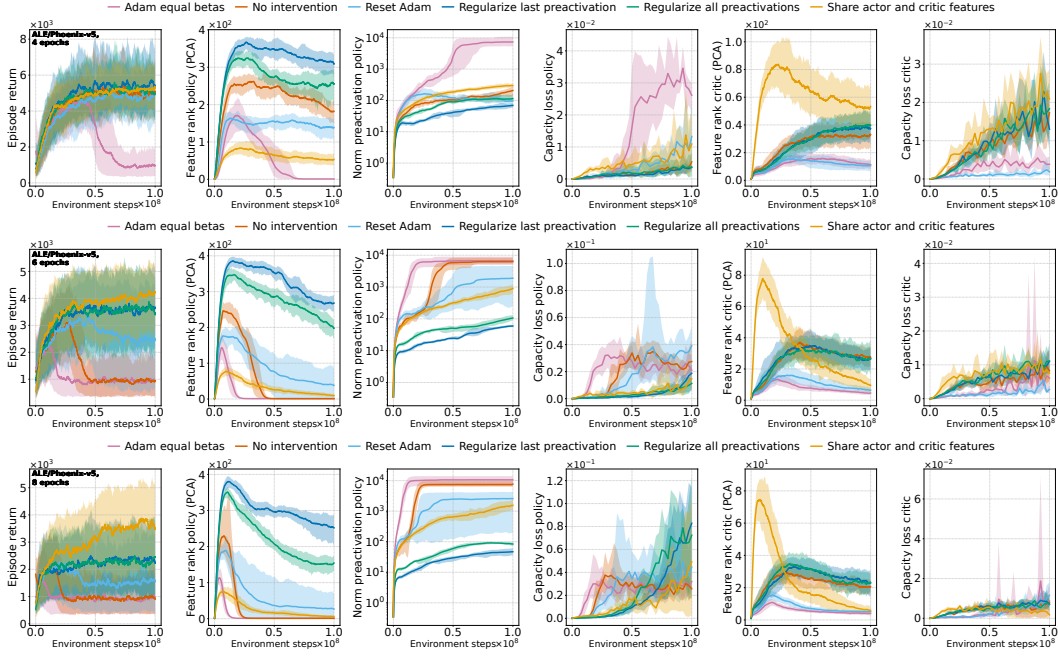

Figure 21: Figure 1 on ALE/Phoenix-v5 with interventions.

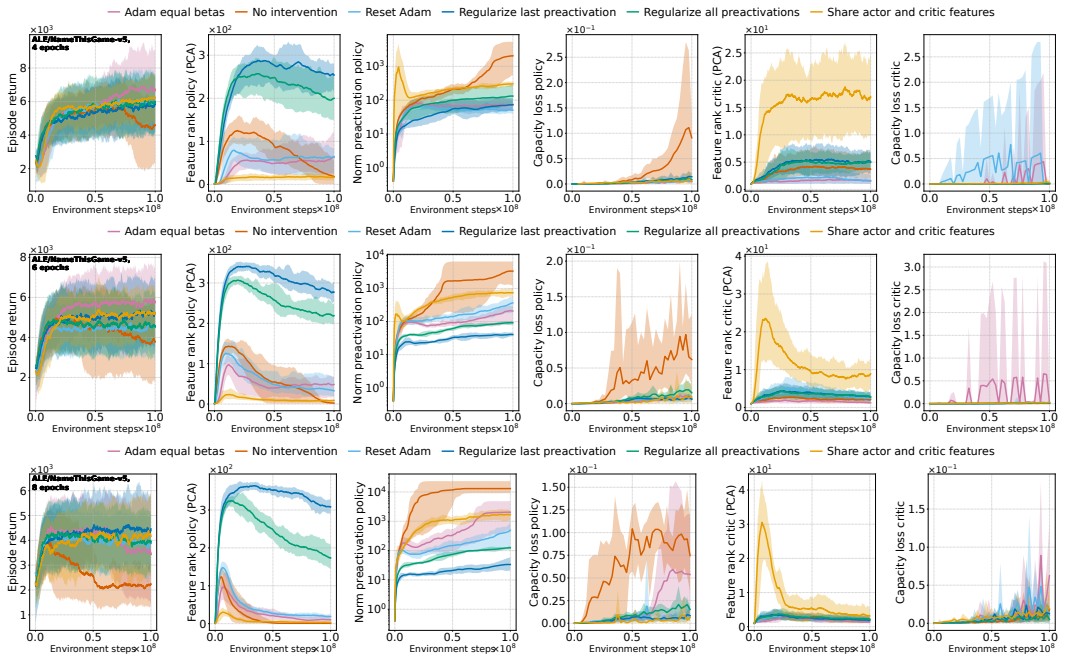

Figure 22: Figure 1 on ALE/NameThisGame-v5 with interventions.

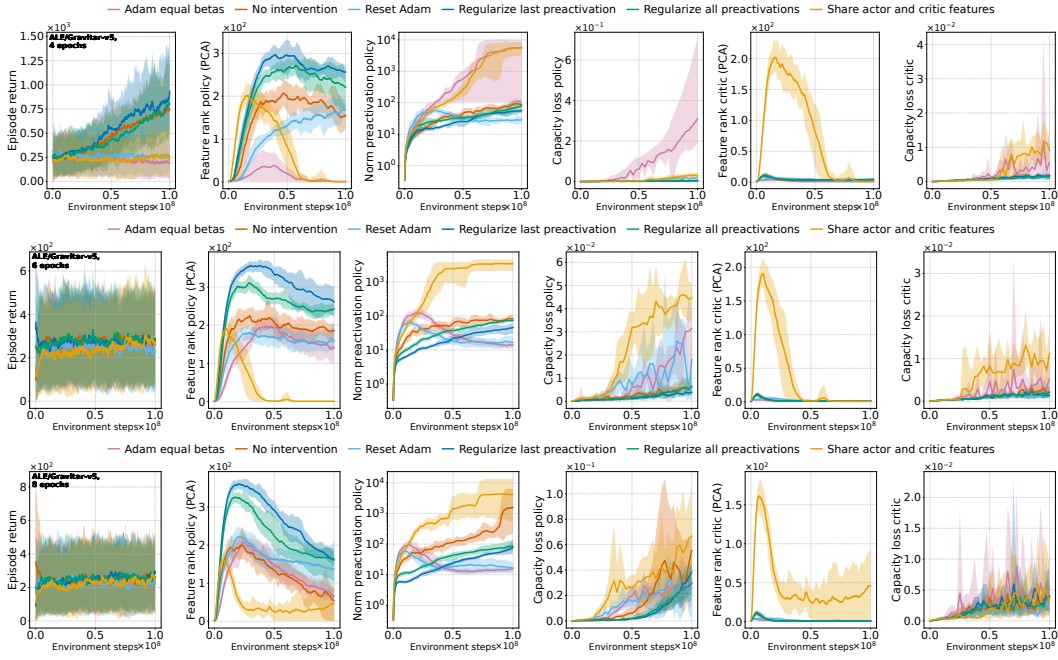

Figure 23: Figure 1 on ALE/NameThisGame-v5 with interventions.

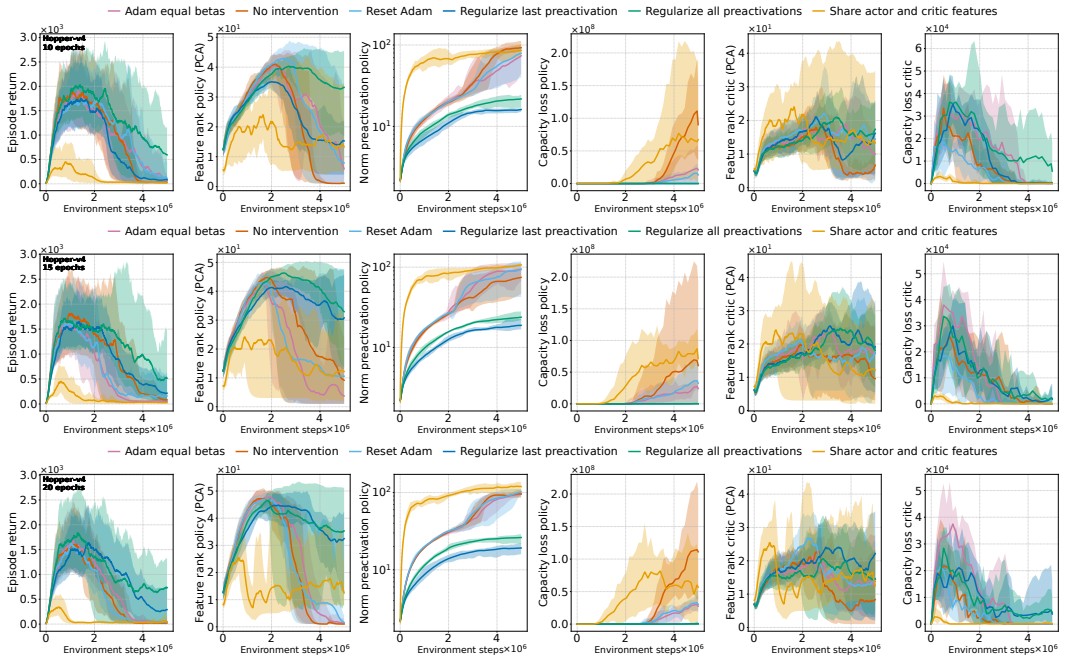

Figure 24: Figure 1 on MuJoCo Hopper with the tanh activation.

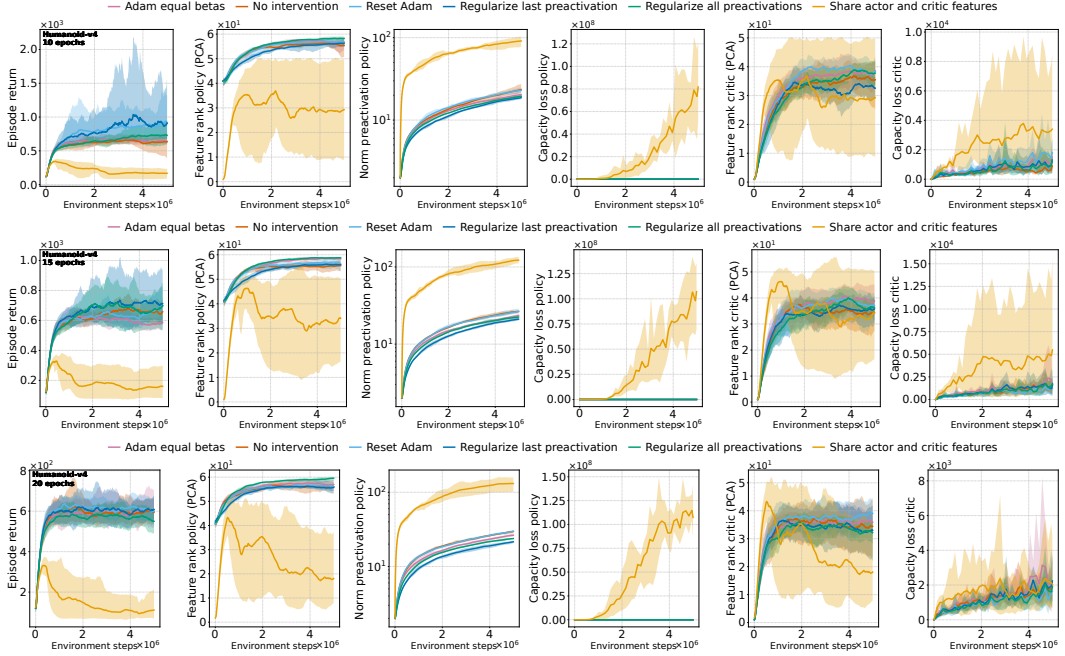

Figure 25: Figure 1 on MuJoCo Humanoid with the tanh activation.

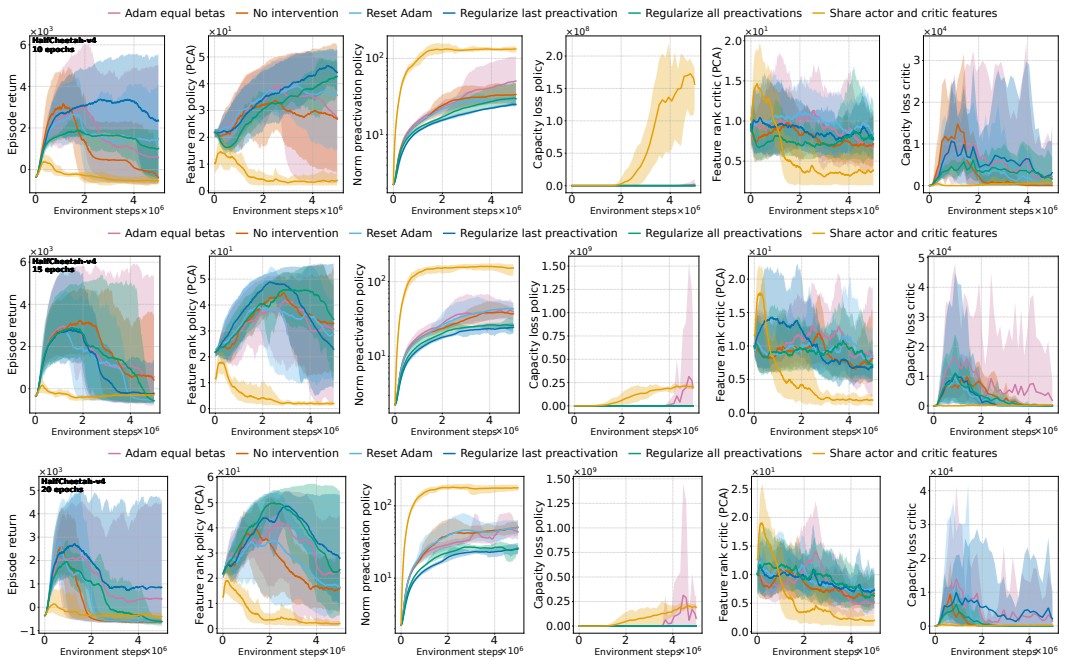

Figure 26: Figure 1 on MuJoCo HalfCheetah with the tanh activation.

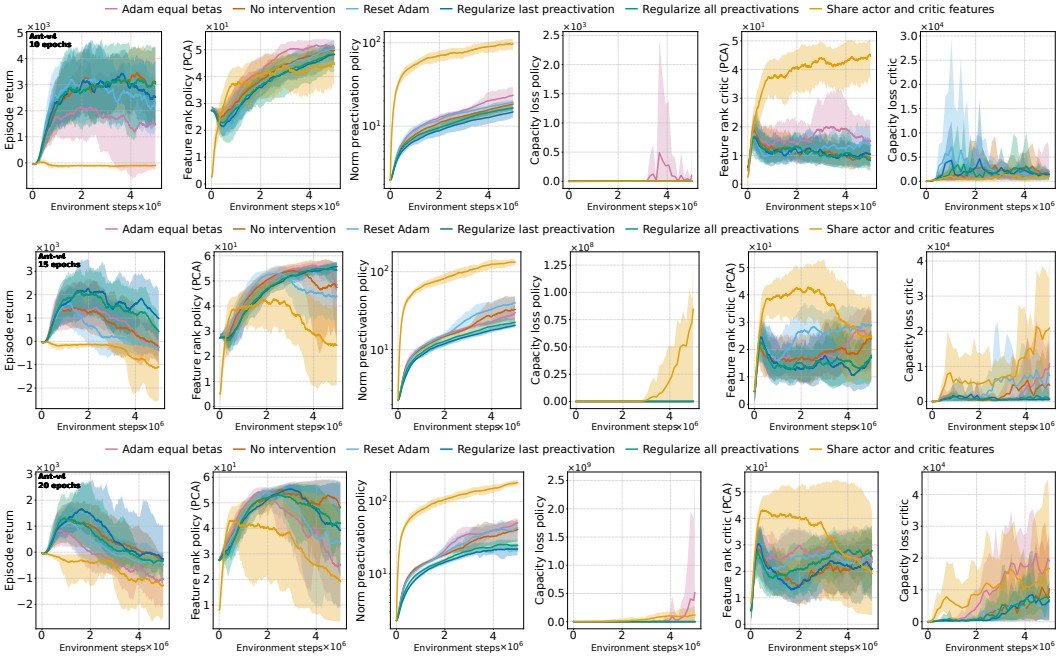

Figure 27: Figure 1 on MuJoCo Ant with the tanh activation.

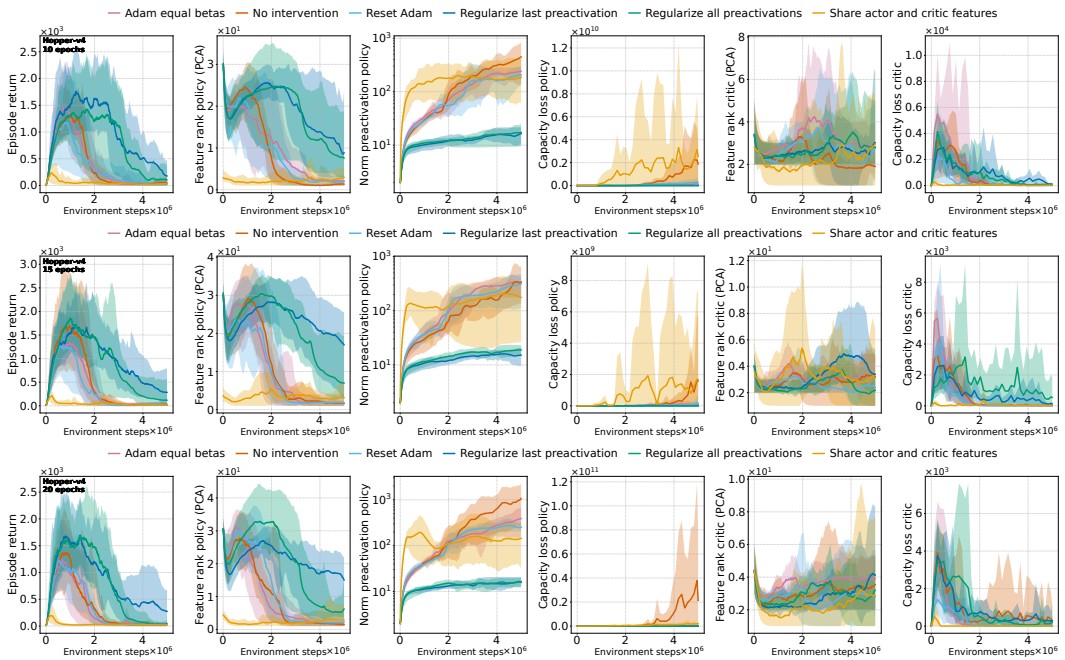

Figure 28: Figure 1 on MuJoCo Hopper with the ReLU activation.

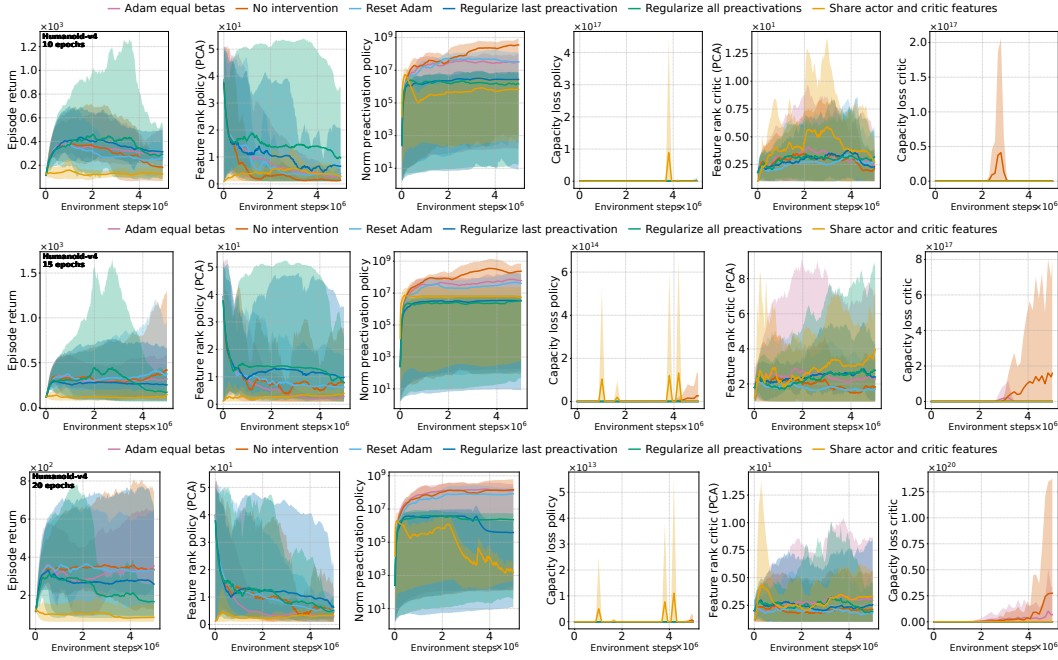

Figure 29: Figure 1 on MuJoCo Humanoid with the ReLU activation.

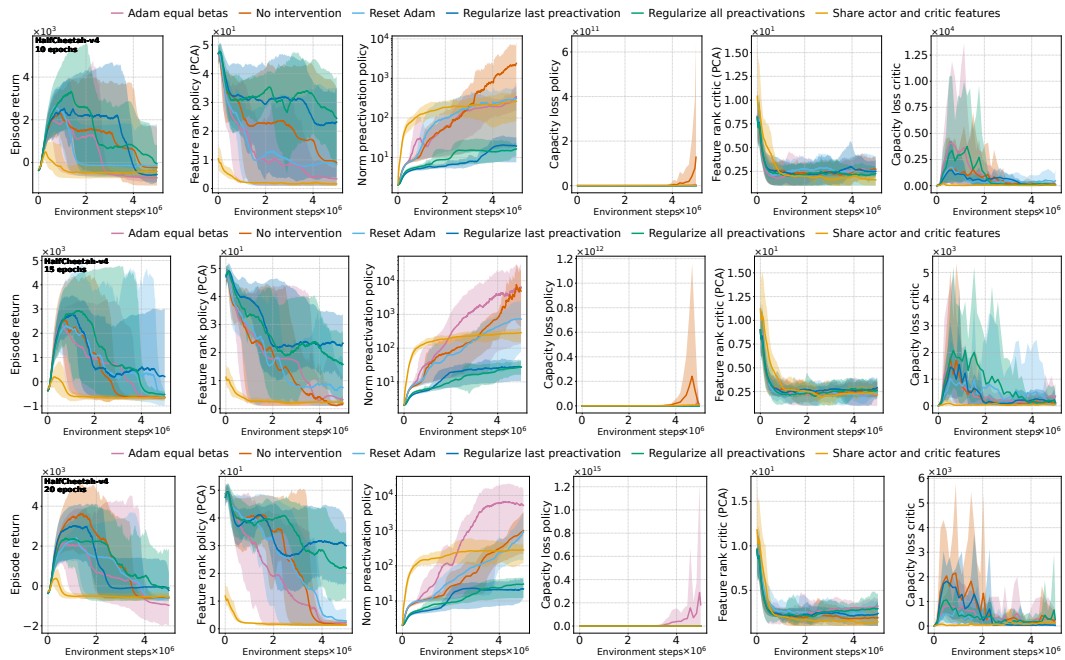

Figure 30: Figure 1 on MuJoCo HalfCheetah with the ReLU activation.

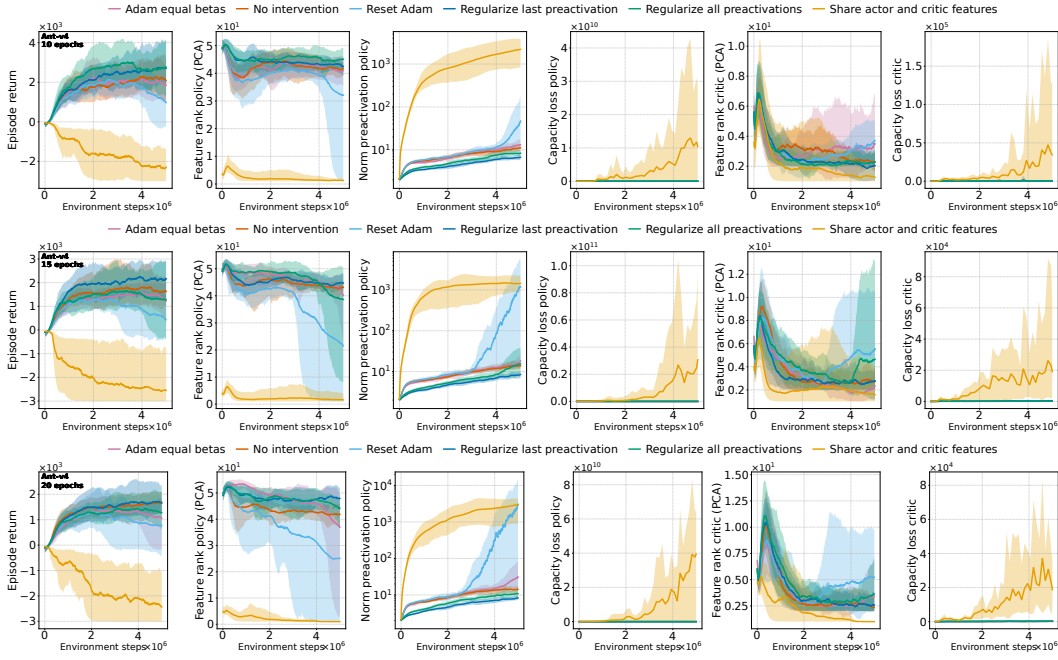

Figure 31: Figure 1 on MuJoCo Ant with the ReLU activation.

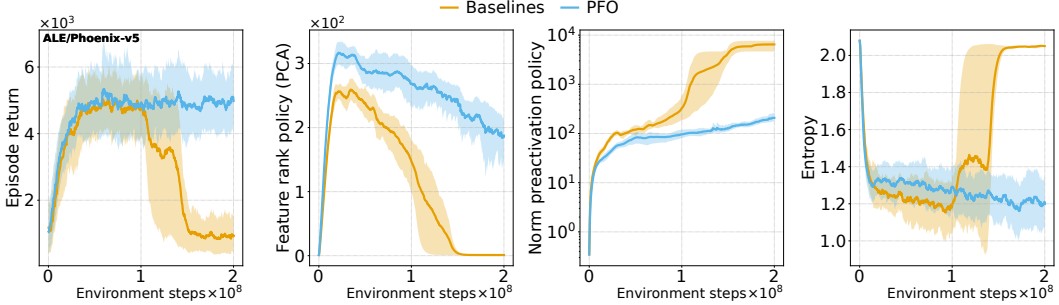

Figure 32: PPO on ALE/Phoenix-v5 collapses with its standard tuned hyperprameters from (Schulman et al., 2017) (4 epochs) when training for 200M steps. Regularizing with PFO mitigates the collapse (applied on the last pre-activation).

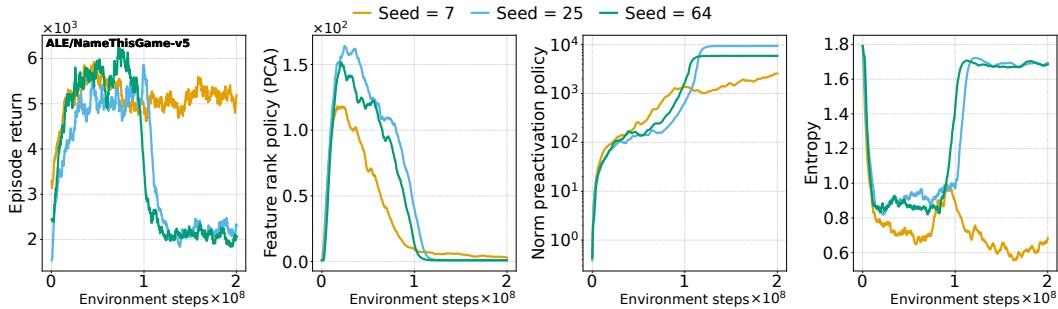

Figure 33: Three seeds of PPO on ALE/NameThisGame-v5 showing that it also collapses with its standard tuned hyperprameters from (Schulman et al., 2017) (4 epochs) when trained for long enough.

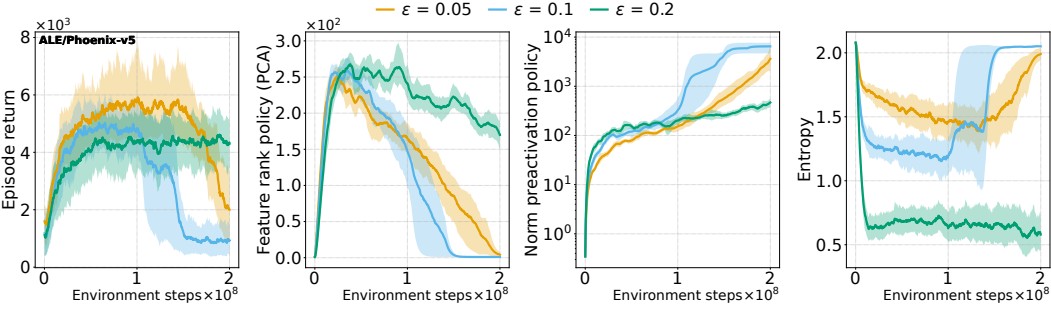

Figure 34: Varying $\varepsilon$ cannot be used as a reliable tool to study collapse as monotonic changes in $\varepsilon$ yield non-monotonic collapse speeds, unlike when varying the number of epochs.

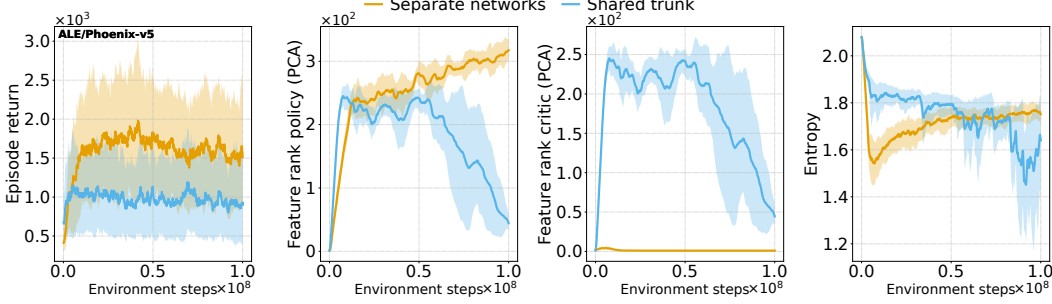

Figure 35: PPO with a shared actor-critic on ALE/Phoenix-v5 with a sparse reward (random masking with probability 0.9) with the standard hyperparameters (4 epochs). While with dense rewards, sharing the trunk was beneficial in ALE/Phoenix (Figure 21 Appendix), with the sparse reward, the opposite is true: sharing the trunk is detrimental.

# E    Measuring and comparing rank dynamics

Several matrix rank approximations have been used in the deep learning literature, and more specifically the deep RL literature, to measure the rank of the representation of features learned by a deep network. In complement to the background presented in section 2, we give here all the rank metrics we have tracked in this work and their correlations, showing that although their absolute values differ, their dynamics tend to describe the same evolution.

## E.1    Definitions of different rank metrics

Essentially, the main difference between the rank metrics considered in the literature is whether they apply a *relative* thresholding of the singular values or an *absolute* one. Their implementation can be found under `src/po_dynamics/modules/metrics.py` in our codebase.

Referring by $\Phi$ the $N \times D$ matrix of representations as in Section 2, and letting $\delta = 0.01$ be the threshold, and $\langle \sigma_i(\Phi), \ldots, \sigma_D(\Phi) \rangle$ the singular values of $\Phi$ in decreasing order, the different rank definitions are as follows.

**Effective rank (Roy & Vetterli, 2007)**   A relative measure of the rank. Let $H(p_1, \ldots, p_k)$ denote the Shannon entropy of a probability distribution over $k$ events and $\|\boldsymbol{\sigma}\|_1$ be the sum of the singular values. Let $\tilde{\sigma}_i(\Phi) = \frac{\sigma_i(\Phi)}{\|\boldsymbol{\sigma}\|_1}$ be the normalized singular values. The effective rank is

$$\exp(H(\tilde{\sigma}_1(\Phi), \ldots, \tilde{\sigma}_D(\Phi)))\}$$

This rank measure has also been used in deep learning by Huh et al. (2023).

**Approximate rank (PCA)**   A relative measure of the rank. Intuitively this rank measures the number of PCA values that together explain 99% of the variance of the matrix. This can also be viewed as the lowest-rank reconstruction of the feature matrix with an error lower than 1%. [12] It is also used in RL by Yang et al. (2020).

$$\min_k \left\{ \frac{\sum_{i=1}^k \sigma_i^2(\Phi)}{\sum_{j=1}^D \sigma_j^2(\Phi)} > 1 - \delta \right\}$$

**srank (Kumar et al., 2021)**   A relative measure of the rank. This is a relative thresholding of the singular values, similar to the approximate rank but with no connection to low-rank reconstruction or variance of the feature matrix.

$$\min_k \left\{ \frac{\sum_{i=1}^k \sigma_i(\Phi)}{\sum_{j=1}^D \sigma_j(\Phi)} > 1 - \delta \right\}$$

**Feature Rank (Lyle et al., 2022)**   An absolute measure of the rank. The number of singular values of the normalized $\Phi$ that are larger than a threshold $\delta$.

$$\left| \left\{ \frac{\sigma_i(\Phi)}{\sqrt{N}} > \delta \text{ for } i \in \{1, \ldots, D\} \right\} \right|$$

**PyTorch rank**   An absolute measure of the rank.   This is the rank computed by `torch.linalg.matrix_rank` and `torch.linalg.matrix_rank`. Let $\epsilon$ be the smallest difference possible between points of the data type of the singular values, i.e. for `torch.float32` that is $1.19209e^{-7}$. This rank is computed as follows.

$$\left| \left\{ \frac{\sigma_i(\Phi)}{\sigma_1 \times N} > \epsilon \text{ for } i \in \{1, \ldots, D\} \right\} \right|$$

It also appears in Press et al. (2007) in the discussion of SVD solutions for linear least squares.

---

[12] https://github.com/epfml/ML_course/blob/94d3f8458e31fb619038660ed2704cef3f4bb512/lectures/12/lecture12b_pca_annotated.pdf

## E.2 Correlations between the rank metrics

We compute various correlation coefficients and distance measures between the rank metrics. To compute a correlation/distance on a pair of rank metrics $(X, Y)$, we take for each training run the set $\{(x_t, y_t)t \in \{0, ..., T\}\}$ of coinciding values of the curves of the two rank metrics during the run that had $T$ logged steps, compute the correlation/distance on this set, and average the correlation/distance values across all considered runs. We also compute the worst correlation/distance between each rank metric pair for a worst-case analysis. We separate the average values and worst-case values by environment (ALE vs. MuJoCo) for a more granular analysis. We consider all the runs without the interventions and exclude a few runs where the models collapse since the beginning of training, giving constant trivial ranks, as these result in undefined or trivial correlation coefficients.

We compute Kendall's $\tau$ coefficient (Kendall, 1938), Spearman's $\rho$ coefficient (Spearman, 1987), the Pearson correlation coefficient, and a normalized L2-distance computed as $\frac{\sqrt{\sum_{t=1}^{T}(x_t - y_t)^2}}{\sqrt{T} \times L}$ where $L$ is the width of the feature layer considered (i.e., 512 for ALE and 64 for MuJoCo).

**Results** We visualize the correlation/distance between the pairs of ranks as heatmaps annotated with averages and standard deviations. Overall, the metrics are highly correlated with average correlation the coefficients varying between 0.99 and 0.51. Individually, no rank metric correlates significantly more on average with the other metrics. Interestingly, from the average correlations, we clearly see two consistent clusters of stronger correlations between the relative rank metrics (approximate rank (PCA) and Effective rank (Roy & Vetterli, 2007)) and absolute rank metrics (Feature Rank (Lyle et al., 2022) and PyTorch rank). The srank (Kumar et al., 2021) which is technically a relative metric, but with a weak normalization rationale, correlates more with the relative metrics on MuJoCo with tanh but more with the absolute metrics on ALE and MuJoCo with ReLU.

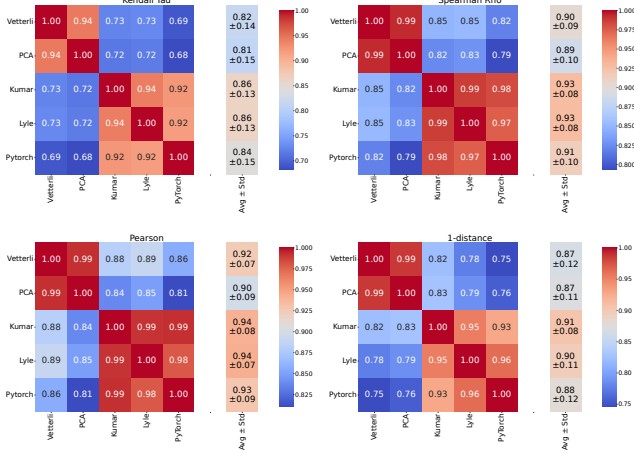

Figure 36: Average correlation between rank metrics on MuJoCo ALE.

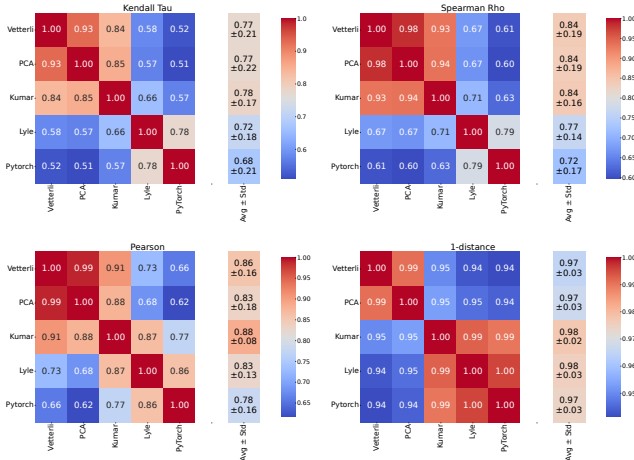

Figure 37: Average correlation between rank metrics on MuJoCo with the tanh activation.

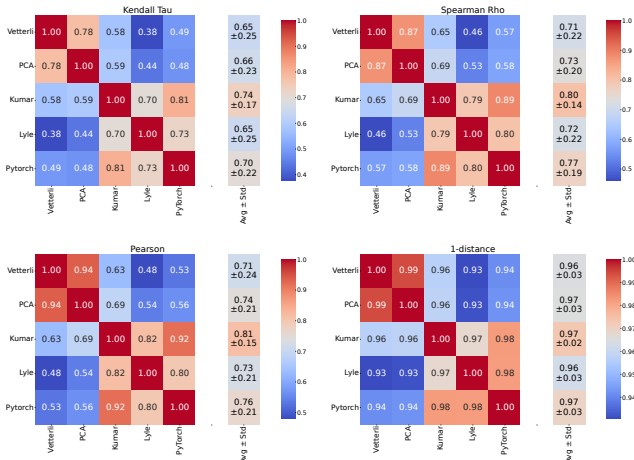

Figure 38: Average correlation between rank metrics on MuJoCo with the ReLU activation.

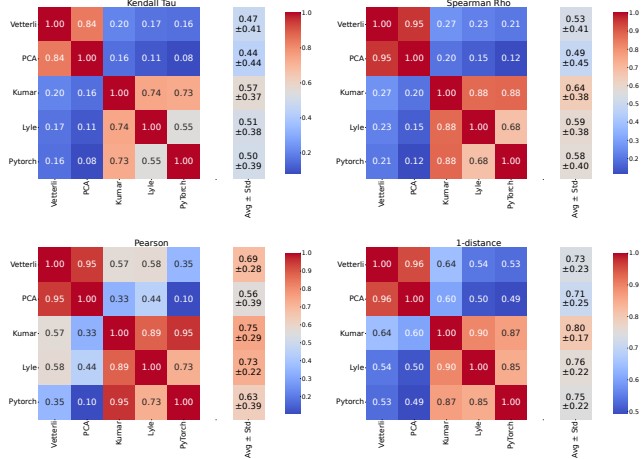

Figure 39: Worst-case correlations between rank metrics on ALE.

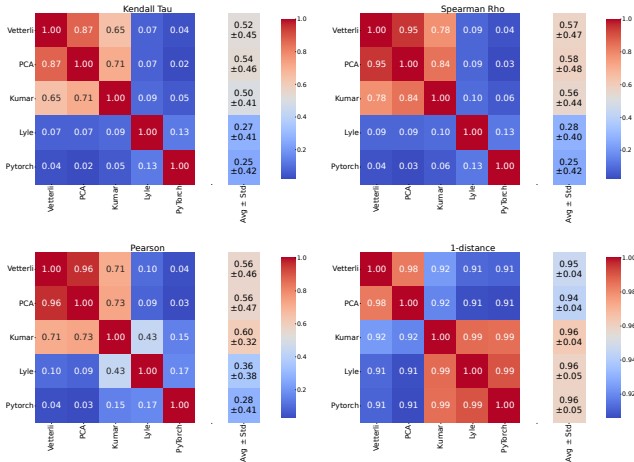

Figure 40: Worst-case correlations between rank metrics on MuJoCo with the tanh activation.

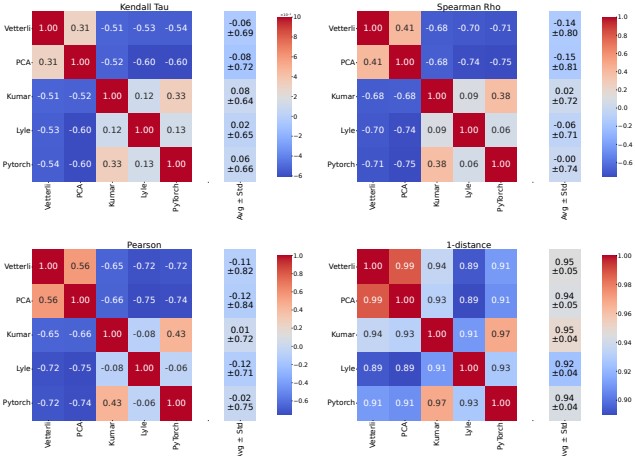

Figure 41: Worst-case correlations between rank metrics on MuJoCo with the ReLU activation.

