# OpenReview forum: "No Representation, No Trust: Connecting Representation, Collapse, and Trust Issues in PPO"
_NeurIPS.cc/2024/Conference — NeurIPS 2024 poster_

### Official Review · Reviewer_4jUo · 2024-06-17

**Soundness:** 3
**Presentation:** 3
**Contribution:** 2
**Rating:** 6
**Confidence:** 5

**Summary:**

The paper investigates the phenomenon of feature collapse in the popular on-policy RL algorithm PPO. It demonstrates that increasing the sample reuse (number of epochs) in PPO deteriorates the feature rank and plasticity. It also finds that the clipping operation for the conservative update in PPO does not prevent this feature collapse. Furthermore, it shows that feature collapse leads to the degradation of the trust region, and vice versa. Finally, the paper proposes several techniques to prevent feature collapse, including feature regularization, network sharing, and Adam hyperparameter adjustments, though these techniques do not fully resolve the issue.

**Strengths:**

- To the best of my knowledge, this is the first work to investigate the phenomenon of feature collapse in on-policy algorithms.
- The experimental design is strong. They appropriately select benchmarks: Atari for discrete action spaces and MuJoCo for continuous action spaces. Additionally, the experiments are extensive and well-conducted.
- The connection between feature collapse and trust region is intriguing and distinguishes this work from previous studies on feature collapse in value-based methods.

**Weaknesses:**

- They demonstrated that training policy and value networks for more epochs leads to feature collapse, but this is not typical in PPO settings. It is uncommon to see PPO run for 6 or 8 epochs (Figure 1, 2, and 3).
- While they attempt to explain feature collapse in the critic networks, the explanation is insufficient and requires further elaboration  (Section 3.1).
- While they emphasize that the focus of the paper is not on improving the performance of PPO, they do not provide a clear solution to mitigate feature collapse in PPO (Figure 6).

**Questions:**

- In Section 3.1 and Figures 7, 8, and 9, there appears to be no clear correlation between the number of epochs and the plasticity loss of the critic network, which is counterintuitive since it contrasts with findings from value-based plasticity studies. Could you explain more about this?
- In Section 3.2, you state that the clipped sample’s ratio will have their probabilities continue to go beyond the clip limit in PPO. How about replacing the clipping operation with a KL penalty? This technique was proposed in the original PPO paper [1], and it has shown no significant difference in performance compared to the clipping operation in some environments [2]. What impact do you think replacing the clipping operation with a KL penalty would have on feature collapse?
- Regarding PFO, instead of applying regularization in the feature space to ensure the current representation does not deviate from previous representations, how about applying regularization in the weight space using EWMA update?
- It would be beneficial to investigate Phasic Policy Gradient (PPG) [2]. It trains the actor network to predict the value of the old policy, which is similar to PFO. It would be interesting to analyze feature collapse in PPG to see if similar patterns emerge as observed in PFO.

[1] John Schulman et al., Proximal Policy Optimization Algorithms, arXiv 2017 \
[2] Karl Cobbe et al., Phasic Policy Gradient, ICML 2020

---

> ### Author Rebuttal · Authors · 2024-08-07
>
> >**Q1.** … leads to feature collapse, but … it is uncommon to see PPO run for 6 or 8 epochs (Figure 1, 2, and 3).
>
> **A1.** **The collapse phenomenon we show is certainly not exclusive to running with more epochs. The phenomenon does happen with the “standard” hyperparameters of the environments we study.**
> This is the case on MuJoCo, and on ALE in Figures 1, 2, and 3, mentioned by the reviewer, we see the start of the collapse at 100M steps, which motivates us to increase the number of epochs to accelerate the collapse, and allows us to show that the phenomenon is due to non-stationarity and the inability of the policy to adapt to new targets.
> In this sense, **increasing the number of epochs is a tool in our analysis and is not a condition for collapse.**
> To observe collapse with the “standard” hyperparameters in Figures 1, 2, and 3 it suffices to train for longer. **We show this on ALE/Phoenix and ALE/NameThisGame with 200M steps in Figures 34 and 35 in the PDF of the rebuttal.**
>
> >**Q2.** While they attempt to explain feature collapse in the critic, the explanation is insufficient
>
> **A2.** **We do not attempt to explain the collapse of the critic network (see lines 147-149)** which has been studied in previous works, e.g., Kumar et al., 2022 through implicit regularization and Lyle et al., 2022 who argue that sparser targets are more detrimental.
> **We make a few observations** about the representation dynamics of the critic network **in relation to the sparsity of the environment** (lines 178-184), which then allow us to draw conclusions about sharing an actor-critic trunk in section 4 (lines 339-355).
>
> If the reviewer meant the actor network, then we do describe its features and dynamics and connect them to non-stationarity. By showing that this collapse also holds in on-policy policy optimization in addition to offline, off-policy, and value-based RL as shown by previous work, we strengthen the hypothesis of non-stationarity being a major cause of the phenomenon.
>
> Ultimately, discovering deeper causes beyond non-stationarity remains an open question as discussed in our conclusion (389-391).
>
> >**Q3.** While they emphasize that the focus of the paper is not on improving the performance of PPO, they do not provide a clear solution to mitigate feature collapse in PPO (Figure 6).
>
> **A3.** In this work, we study interventions that have been shown in other settings to help and present their effects on the representation dynamics of PPO. This serves two purposes. 1) **showing which ones can help** and under which conditions (which environment features), and 2) when they do help and improve the trust region at the same time, **strengthen the relation we have drawn between representation collapse and trust-region collapse.**
> We have shown that sharing the actor-critic network when the environment has a rich reward signal does mitigate collapse (Figure 6, top and middle), and applying PFO also mitigates collapse when sharing the actor-critic trunk does not (Figure 6, 19, 20).
>
> However, like any other empirical study, and as mentioned in our limitations, there is no guarantee on how well our results would generalize to other settings. **We hope that our paper encourages further theoretical studies, and only then will we have provably or “clearly” mitigated the problem.**
>
> >**Q4.** … no clear correlation between the number of epochs and the plasticity loss of the critic network … it contrasts with findings from value-based plasticity studies.
>
> **A4.** **Runs with higher epochs tend to have higher plasticity loss which can be explained by a stronger overfitting to previous targets and is consistent with the findings from value-based methods.**
>
> **There are some exceptions,** which have probably raised the reviewer’s attention, which can be explained by the behavior of the policy. When the policy collapses earlier in a run with a higher number of epochs, the critic's plasticity loss is prematurely stopped (there are no more changing objectives) and ends up lower than in a run with a lower number of epochs which has continued training until the end.
>
> **This is the case of Figure 7 and is explained in its caption.**
>
> >**Q5.** How about replacing the clipping operation with a KL penalty? …
>
> **A5.** We have also performed early experiments with PPO-KL. However, the adaptive coefficient of the KL trust region makes the algorithm extremely high variance and often collapses due to the coefficient exploding or collapsing independently of feature dynamics.
>
> Nevertheless, for the runs where the coefficient was stable, **we did indeed observe the same behavior as PPO-Clip**, specifically that the features became more aliased until they collapsed, followed by a trust region collapse characterized by a blow-up of the KL divergence.
>
> >**Q6.** ...how about applying regularization in the weight space using EWMA update?
>
> **A6.** PFO doesn’t stem from the sole motivation of regularizing the representations, it also seeks to do so by extending the trust region set by the clipping in the output space to an additional regularizer in the feature space.
>
> The idea is to not impose an explicit constraint or a regularization on the weights directly to allow them to adapt to the moving targets but to regularize their dynamics through the features and output trust regions.
>
> >**Q7.** It would be beneficial to investigate Phasic Policy Gradient (PPG) [2]
>
> **A7.** This is a great suggestion. We would put PPG in the same box of algorithms that add an auxiliary representation loss to the policy network while not interfering too much with its objective, unlike sharing the actor-critic trunk, which we show can be detrimental when its rank is low to start with.
>
> In the scope of this work, we did not investigate such algorithms, as a thorough analysis would require assessing a comprehensive sample of these algorithms such as PPG, InFer (Lyle et al., 2022), and CURL (Laskin et al., 2020). We reserve this study for future work.

---

> ### Comment · Reviewer_4jUo · 2024-08-08
>
> Thank you for your responses to my questions. My answers to some of your responses are shown below.
>
> > Q1. … leads to feature collapse, but … it is uncommon to see PPO run for 6 or 8 epochs (Figure 1, 2, and 3).
>
> A1. Thank you for the detailed explanation! Increasing the number of epochs makes sense to me now.
>
> > Q2. While they attempt to explain feature collapse in the critic, the explanation is insufficient.
>
> A2. I would like you to elaborate on Lines 182-184. However, I understand that the main focus of this work is not on the plasticity of the critic network. Thanks!
>
> > Q4. … no clear correlation between the number of epochs and the plasticity loss of the critic network … it contrasts with findings from value-based plasticity studies.
>
> A4. I think I missed the caption. Thank you for letting me know.
>
> > Q5. How about replacing the clipping operation with a KL penalty? …
>
> A5. Thank you for clarifying my question! Including these results will strengthen your paper.
>
> ~All of my concerns have been well addressed, and therefore I am increasing the score from 5 to 6.~
>
> Updated: After reviewing feedback from other reviewers, I found that similar studies had been conducted prior to this work. Therefore, I have decided to maintain the original score.

---

> > ### Author Response · Authors · 2024-08-11
> >
> > Can the reviewer elaborate on their update about prior similar studies? Which studies does the reviewer refer to and to what extent are they similar to our work?
> > We clearly acknowledge building on previous work in value-based methods studying representation dynamics and reference previous works on PPO exhibiting issues with non-stationarity and performance but no connection to representation dynamics. To the best of our knowledge, our work is the first to draw connections between representation collapse, trust-region issues, and performance collapse.

---

### Official Review · Reviewer_v7XX · 2024-06-23

**Soundness:** 3
**Presentation:** 3
**Contribution:** 3
**Rating:** 7
**Confidence:** 4

**Summary:**

The paper examines the loss of plasticity and its connection representation collapse in policy networks in online reinforcement learning (as opposed to previously studied value networks in offline reinforcement learning). The paper establishes the problem in the Atari game domain including the growth in the norm of feature pre-activations of the policy network. The analysis isolates and illustrates the problem of increasing representation state collinearity and representation collapse in the trust region of a policy in a toy domain. To mitigate representation collapse the paper proposes a feature regularization loss term based on these pre-activation features of the policy network.

The evaluation compares methods to mitigate policy network representation collapse including the new loss, sharing the actor and critic feature trunk, and modifying the Adam optimizer (by resetting moments or faster decay of second moments) in the ALE and MuJoCo environments. The results show the loss has does not impact episode returns but does mitigate loss of plasticity.

**Strengths:**

# originality
Good
- Investigating plasticity loss in offline RL is established, but has not been extended to online RL.
- The analysis provides new insights on trust region optimization.

# quality
Good
- The toy domain isolates the effect of interest to strengthen the empirical results and motivate the new regularization term.
- the experiments show modest evidence for improvements and report variation in results over runs.

# clarity
Mostly good
- The exposition on trust region weaknesses and collapse could be clearer and make the main points sooner. The rest of the text was good.

# significance
Good
- Loss of plasticity is an important problem, particularly in the case of online RL.
- The paper will be of interest to researchers in continual learning and RL communities; both theoretical (for the trust region points) and practical (for the regularization term) researchers.

**Weaknesses:**

The paper would benefit from outlining the trust region insight sooner as it is a core idea that only gets explained in detail on page 7 of the paper.

The results on the regularization term are a bit confusing and mixed. The narrative would benefit from clarifying the aspects that remain unresolved or ambiguous that merit further attention. The Gravitar analysis of sparse reward environments was helpful. But I was not really sure what to make of the effects on episode returns compared to representation collapse. Some way of teasing these effects out more cleanly would help.

**Questions:**

- line 143: "do not include normalization layers"
	- Does this help plasticity preservation? (even if only to cite prior work on the effects)
- lines 151-153: "We vary the number of epochs as a way to control the effect of non-stationarity, which gives the agent a more significant number of optimization steps per rollout while not changing the optimal target it can reach due to clipping, as opposed to changing the value of ε in the trust region for example."
	- It might be interesting to contrast the effects of varying $\epsilon$ as well.
- line 280 (minor): $\pi_\theta(a_i | s)$
	- $a_i$ should be $a_1$ in this case
- The toy setting was good demonstration of the problems with trust region constraints. It may help to introduce this sooner as I was confused about the trust region claims being made until that point.
- Figure 6 (minor)
	- The caption only mentions "Top" and "Bottom", omitting the middle row of "NameThisGame".

**Limitations:**

Yes. The paper acknowledges remaining opportunities to relate the findings on representation collapse to other research on plasticity loss. And to incorporate more complex model architectures (like memory).

There is no need for additional comments on potential societal impacts.

---

> ### Author Rebuttal · Authors · 2024-08-07
>
> >**Q1.** The toy setting was good demonstration of the problems with trust region constraints. It may help to introduce this sooner … The paper would benefit from outlining the trust region insight sooner as it is a core idea that only gets explained in detail on page 7 of the paper.
>
> **A1.** We thank the reviewer for the recommendation. This is indeed a key point of interest to the readers. **We will update the introduction section to describe the insights sooner and point to the theoretical derivation for interested readers.**
>
> >**Q2.** The results on the regularization term are a bit confusing and mixed. The narrative would benefit from clarifying the aspects that remain unresolved or ambiguous that merit further attention. The Gravitar analysis of sparse reward environments was helpful. But I was not really sure what to make of the effects on episode returns compared to representation collapse. Some way of teasing these effects out more cleanly would help.
>
> **A2.** Figure 6 presents summaries of runs at the end of training aggregating runs with different numbers of epochs. It, therefore, shows the distribution of returns and representation metrics across different hyperparameters.
> With this, when a method like PFO presents an improvement of the lower tail of episodic returns and, at the same time, an improvement of the representation metrics (no extremely poor ranks or fully dead networks), we can say that **the method improves robustness to representation collapse and performance collapse, and strengthens the link we’ve drawn between the two in the earlier sections.   In this sense, the effects on episode returns and representation collapse should be appreciated in tandem.**
> It should also be noted that the strong relation between them mainly holds around the collapsing regime, as described in Figure 4 and explained in our global rebuttal answer GA2.
>
> >**Q3.** line 143: "do not include normalization layers". Does this help plasticity preservation? (even if only to cite prior work on the effects)
>
> **A3.** Lyle et al. (2023) conducted a study with non-stationary supervised learning and value-based methods and found that normalization layers do provide improvements to plasticity. In the scope of this work, we did not look at interventions that change the “degree” of non-stationarity in the input or output of the network, such as layer normalization, batch normalization, and running observation or reward normalization, as these would require a different analysis. As stated in our discussion and limitation plan to address these in future work.
>
> >**Q4.**  lines 151-153: "We vary the number of epochs as a way to control the effect of non-stationarity, which gives the agent a more significant number of optimization steps per rollout while not changing the optimal target it can reach due to clipping, as opposed to changing the value of ε in the trust region for example." It might be interesting to contrast the effects of varying epsilon as well.
>
> **A4.** As mentioned, varying epsilon would change the optimum of the surrogate objective so it’s not exactly clear if we can use this as a tool to investigate the relation between non-stationarity and feature collapse, like we did the number of epochs.
> **Nevertheless, per the reviewer’s request, we rerun Figure 1 with ALE/Phoenix with different values of epsilon** and the baseline number of epochs of 4. As expected, and as shown in Figure 36 of the additional rebuttal PDF, **we observe no apparent correlation with the time of collapse as both doubling (epsilon =0.2) and halving (epsilon =0.05) the baseline epsilon (epsilon =0.1) can yield training curves with a delayed representation and performance collapse.**
>
> >**Q5.** line 280 (minor): a_i should be a_1 in this case
>
> **A5.** Indeed. We thank the reviewer for the detailed feedback!
>
> >**Q6.** Figure 6 (minor). The caption only mentions "Top" and "Bottom", omitting the middle row of "NameThisGame".
>
> **A6.** Top and middle share the same properties. We will add this to the caption.

---

> > ### Comment · Reviewer_v7XX · 2024-08-11
> >
> > Thank you for addressing my questions.
> >
> > I do not see any substantial changes that would alter my score. After discussion with the other reviewers I am lowering my score in light of the weakness of the causal evidence for what drives collapse and the lack of strong results showing cases where PFO substantially improves over alternative methods.

---

> > > ### Author Response · Authors · 2024-08-11
> > >
> > > We thank the reviewer for the follow-up.
> > >
> > > > “in light of the weakness of the causal evidence for what drives collapse”
> > >
> > > Can the reviewer elaborate on this? Some reviewers have raised concerns about the scope of the cause evidence, which holds primarily around the collapse regime**, but it does not seem like concerns have been raised about the weakness of the evidence.**
> > >
> > > If the reviewer refers to **the scope of the collapse, then we have addressed reviewer YzGo’s concern about the underwhelming scope in our reply to them.** Quoting:
> > >
> > > “**Unfortunately, this is a fact.** We have extensively investigated the link in the early stages of the project to find strong evidence throughout training, **which to us was also the more exciting claim to prove; however, countless examples made us realize that the link did not necessarily hold throughout training but only around the collapsing regime.**
> > >
> > > Using only positive examples or a narrow view of the correlations, **to claim that such a link is present throughout training would be extremely misleading and has been an issue in offline RL.**
> > >
> > > **This does not mean that one should only be concerned about the link when the performance is starting to deteriorate.  The representations don’t collapse all of a sudden, they deteriorate throughout training until they reach collapse.  So mitigating representation degradation should happen throughout training and not only when around the collapsing regime.**”
> > >
> > >   > and the lack of strong results showing cases where PFO substantially improves over alternative methods.
> > >
> > > **"PFO substantially improving over alternative methods" has never been a claim made in this work.  This is a misunderstanding raised by reviewer qwmj, which was clarified in our reply to their review.** PFO is not meant to improve best run performance (lines 330-332), but to mitigate collapse, and to show that mitigating representation collapse mitigates performance collapse.
> > >
> > > Quoting our reply to reviewer **qwmj:**
> > >
> > > “the **significance of PFO is realized by a) successfully mitigating collapse** as observed in Figure 6 with a significantly higher median performance indicated by the black lines, and b) improving the representation metrics and obtaining a better trust region, therefore **strengthening the relation we’ve drawn between representation, trust region, and performance around collapse.**”

---

> > > > ### Comment · Reviewer_v7XX · 2024-08-13
> > > >
> > > > I thank the authors for their thorough discussion and replies here and with the AC.
> > > >
> > > > The further clarifications have been illuminating and adjusted my views overall. I've raised my score as a result of seeing:
> > > >
> > > > - the presentation was weaker than initially believed (due to the confusions being discussed and potentially resolved)
> > > > - the results are narrower in scope than I originally understood (which remains valuable but narrows the audience of interest somewhat)
> > > > - the results are empirically stronger around the circumstances of collapse and potential understanding they yield than I most recently believed

---

### Official Review · Reviewer_6zAY · 2024-07-12

**Soundness:** 3
**Presentation:** 4
**Contribution:** 3
**Rating:** 6
**Confidence:** 5

**Summary:**

The paper addresses non-stationarity in RL and its impact on deep learning networks, focusing on PPO. It identifies that networks in PPO, like those in off-policy methods, suffer from representation rank deterioration, leading to performance collapse. The authors propose Proximal Feature Optimization (PFO), a new regularization on the policy representation that regularizes preactivation changes, which can mitigate performance collapse and improve the agent’s performance.

**Strengths:**

- The work exhibits a coherent structure, and the author adeptly elucidates the underlying motivation. The author adeptly elucidates the shortcomings of PPO and offers enough reasons. This paper is well-written.
- Section 3's Q1 and Q2 effectively describe the potential problems of PPO, and I concur with the underlying motivation.
- The paper not only identifies problems but also suggests practical interventions, such as PFO and sharing the actor-critic trunk, to regularize representations and reduce non-stationarity, showing promising improvements in performance.
- The authors provide details of the experiment design, and it's clear.

**Weaknesses:**

The authors claim to open data and code; however, I could not locate them. Therefore, I apologize if I overlooked their presence. Refer to the question section for other weaknesses.

**Questions:**

- Please provide a more detailed explanation on Eq.2. I don't understand the meaning of $\phi$, and should the formula be $s_t$ instead of $S_t$?
- In section 4, the authors introduce an intervention that allows the actor and critic to share all the layers. The experiment results illustrate that this can bring some improvements but not significantly. I am more curious about whether the reason for this is the sparsity of reward or the environment itself. I think the latter is more likely, and we can verify this by setting up a control experiment with different reward functions in the same environment.

---

> ### Author Rebuttal · Authors · 2024-08-07
>
> >**Q1.** The authors claim to open data and code; however, I could not locate them. Therefore, I apologize if I overlooked their presence.
>
> **A1.** **Yes, these are available on an anonymous GitHub repo mentioned in Appendix line 578.** The GitHub repo contains further links to the Weights&Biases project with all the interactive training curves and links to download the raw logs.
>
> >**Q2.** Please provide a more detailed explanation on Eq.2. I don't understand the meaning of phi, and should the formula be s_t instead of S_t?
>
> **A2.** $\phi(S_t)$ refers to the vector of the preactivations of the penultimate layer of the network on the state $S_t$. As PFO can also take all the pre-activations in the network, $\phi(S_t)$ could also be the concatenation of the preactivations of all the layers.
> We will add these details before the equation in the camera-ready version.
> (This is analogous to $\Phi$ defined in the background (lines 100-104) as the feature matrix, consisting of the activations of the last hidden layer.)
> $S_t$ refers to the state that we capitalize as it is a random variable in the expectation over trajectories, like all other random variables in the background (lines 58 - 66) This is the notation from Sutton and Barto (2018).
>
> Therefore, the equation is essentially an L2 norm between the features of the same state at the old policy and the learned one: $\|\| \phi_\theta(S_t) - \phi_\text{old}(S_t)\|\|_2^2$.
>
> >**Q3.** In section 4, the authors introduce an intervention that allows the actor and critic to share all the layers. The experiment results illustrate that this can bring some improvements but not significantly. I am more curious about whether the reason for this is the sparsity of reward or the environment itself. I think the latter is more likely, and we can verify this by setting up a control experiment with different reward functions in the same environment.
>
> **A3.** This is a good observation! The representation of the critic plays a key role in the reasoning. When the critic, if trained separately, is subject to a degraded representation (or collapse) with a very low rank and many dead units, we observed that sharing its representation with the policy makes the policy collapse faster.
> **When the environments are similar** – we consider two ALE environments, e.g., Phoenix and Gravitar, to be similar because of the observation space, etc – **the main difference in making a critic collapse or not is sparsity of the reward** as observed by Lyle et al. (2022). **Therefore in this case, it is fair to say that the reason for the collapse is mainly due to reward sparsity.**
>
> Nevertheless, one needs to be careful when comparing environments when more than one degree of freedom changes; for example, this would make comparisons of experimental results obtained on Mujoco and ALE environments harder to attribute to sparsity only.
>
> **Per the reviewer's request we ran an experiment on ALE/Phoenix (a dense reward environment), with a reward mask** randomly masking a reward with 90% chance, and compared the effects of sharing the actor-critic trunk. The results are in the additional PDF of the general rebuttal. **As expected, while with dense rewards, sharing the trunk was beneficial in ALE/Phoenix (Figure 21 Appendix), with the sparse reward, the opposite is true: sharing the trunk is detrimental (Figure 37 Rebuttal).** This confirms our conclusion. We thank the reviewer for suggesting the experiment, we will add it to the Appendix of the paper. We hope this strengthens the reviewer's support for the paper.

---

### Official Review · Reviewer_XWGd · 2024-07-12

**Soundness:** 3
**Presentation:** 3
**Contribution:** 3
**Rating:** 6
**Confidence:** 4

**Summary:**

This work provides an empirical study of the feature rank deterioration and loss of plasticity of the Proximal Policy Optimization (PPO) algorithm on Atari and Mujoco tasks. Then links the deterioration of the performance to representation collapse and hence the break of the trust region. From there, the authors propose an auxiliary loss to PPO to maintain the representation rank.

**Strengths:**

The study is very interesting and brings some new insight into how PPO works in practice. The experiments are conducted in clear logic and the analysis and observations are all novel and interesting.

**Weaknesses:**

Figures can be plotted with better quality, don't overlap the labels (in Figure 3), and maybe set the titles to the quantity of interest.

I understand that several works are measuring the feature rank at the last policy/critic layer. But why that is sufficient evidence that the policy is losing rank? For instance, if my action space is binary, then the policy's last layer could learn to be rank-2. More information might be stored in previous layers. From another perspective, isn't it expected that the policy learns to lose rank, by compressing state information to only correlate with actions?

The proposed Proximal Feature Optimization objective is a bit confusing. First it should be a $\ell_2$ norm? What does it mean by $(\phi - \phi_\text{old})^2$. Secondly, from a high-level, isn't this just a regularization loss as in continual learning that prevents forgetting? But based on my knowledge it will exacerbate the the loss of plasticity. If you want to preserve rank, should not you add some reconstruction loss?

**Questions:**

Could the authors address my questions in the weakness section?

From Figure 6, does it mean that the proposed method does not consistently improve the loss of plasticity?

If the main goal is to prevent the preactivation norm from going larger, why not just regularize it to be smaller?

**Limitations:**

Ues

---

> ### Author Rebuttal · Authors · 2024-08-07
>
> >**Q1.a** Figures can be plotted with better quality, don't overlap the labels (in Figure 3),
>
> **A1.a** Indeed. We will correct this in the camera-ready version.
>
> >**Q1.b** and maybe set the titles to the quantity of interest.
>
> **A1b.** Can the reviewer please elaborate on this or give an example? What is meant by title and quantity of interest?
>
> >**Q2.a** isn't it expected that the policy learns to lose rank, by compressing state information to only correlate with actions?
>
> **A2.a** **Indeed, however, we distinguish between a low rank that’s beneficial and an extremely low rank that is detrimental. We have clarified this in answer GA1 in the global rebuttal.**
>
> >**Q2.b** I understand that several works are measuring the feature rank at the last policy/critic layer. But why that is sufficient evidence that the policy is losing rank? For instance, if my action space is binary, then the policy's last layer could learn to be rank-2. More information might be stored in previous layers.
>
> **A2.b** **The penultimate layer serves as a bottleneck** for decoding the actions or the value, so when this layer reaches an extremely low rank (as described in the previous answer, to distinguish for a beneficial low rank), this becomes irrecoverable. This also strongly correlates with plasticity loss, as this layer serves as a bottleneck for reconstructing information from the input state.
>
> >**Q3.a** The proposed Proximal Feature Optimization objective is a bit confusing. First it should be a l2 norm? What does it mean by (phi(s) - phi_old(s))^2.
>
> **A3.a** Yes, we thank for pointing this out. $\phi(s)$ is a vector containing the pre-activations.
> The square of the difference should be more clearly written as the squared norm of the difference between the vectors: $\|\| \phi_\theta(S_t) - \phi_\text{old}(S_t)\|\|_2^2$.
>
> >**Q3.b** Secondly, from a high-level, isn't this just a regularization loss as in continual learning that prevents forgetting? But based on my knowledge it will exacerbate the the loss of plasticity. If you want to preserve rank, should not you add some reconstruction loss?
>
> **A3.b** **PFO regularizes the change in features at every batch, unlike methods that prevent forgetting, which regularize the parameters towards a fixed optimum of a previous task.**
> In this sense, PFO is sought to be an extension of the clipping trust region applied in the output space to a trust region in the feature space. It comes from our observation that the clipping is not enough to satisfy the trust region.
> It also allows us to address the undesired symptoms we observed (high feature norm and aliased features).
> Other auxiliary losses, such as a reconstruction loss, can also be beneficial in this case but do not necessarily constrain or regularize the step size.
> Finally, to support our claim that regularizing representations would prevent trust region and performance collapse, we found that using a regularization that directly targets one of the undesired symptoms we observed would make the connection more straightforward.
>
> >**Q4.** From Figure 6, does it mean that the proposed method does not consistently improve the loss of plasticity?
>
> **A4.** **It does consistently improve the plasticity loss.** This can not show in the the figure when the non-regularized model collapses too early. This is described in the caption of Figure 18: "the tails of the plasticity loss on Phoenix with interventions can be higher than without interventions on the runs where the models collapse too early without interventions, leading to the plasticity loss of the non-collapsed models with interventions eventually becoming higher."
>
> >**Q5.** If the main goal is to prevent the preactivation norm from going larger, why not just regularize it to be smaller?
>
> **A5.** The increasing preactivation norm phenomenon can be seen as an effect of a bigger issue: the failure of the clipping trust region imposed on the output to ensure a trust region on the rest of the network. Therefore,  as mentioned in A3.b, PFO is sought to extend this trust region. Analogously to constraining ratios between the previous and current policy, it regularizes the difference in features between the previous and current policy.
> **This allows PFO to address both the feature norm and the trust region issue. This is described in lines (319-322), which we will rephrase to highlight this point more.**

---

> > ### Comment · Reviewer_XWGd · 2024-08-11
> > **Reply**
> >
> > I thank the authors for their rebuttal and clarification. By "setting the title to the quantity of interest" I just meant that you could directly set an informative title for the figures. I will maintain my scores.

---

> > > ### Author Response · Authors · 2024-08-11
> > >
> > > We thank the reviewer for their clarification. We will add this to our final version.

---

### Official Review · Reviewer_YzGo · 2024-07-13

**Soundness:** 2
**Presentation:** 3
**Contribution:** 2
**Rating:** 5
**Confidence:** 5

**Summary:**

The paper investigates the phenomenon of loss of trainability and performance collapse with PPO agents in standard RL benchmarks. The same phenonema from other settings is found to hold with PPO agents and, moreover, increasing the number of training epochs exacerbates the effect.
Further investigation finds that PPO's trust-region constraint is not maintained when the feature rank decreases and a mechanism for this finding is explained.
The authors propose a regularizer, PFO, which softly constrains the change in features is shown to mitigate trainability loss throughout the learning process.

**Strengths:**

The paper investigates the topic of trainability loss in a less-explored context, on-policy policy optimization, and focusing on a popular algorithm, PPO. Better understanding this algorithm and how it interacts with neural networks could lead to widespread impact as the research direction is developed.

A wide variety of experiments are conducted with some interesting findings. To me, the most intriguing insight is the connection between low feature ranks and increased violations of PPO's trust region. The investigation is fairly thorough, examining various metrics in tandem throughout the training process. Experiments seem to be conducted and reported in a reasonable fashion.
I appreciate the simple, clear example to demonstrate how a poor representation may lead to the clipping constraint being ineffective (line 270-...).

**Weaknesses:**

Currently, the paper feels a bit disorganized and it is a bit difficult to follow the train of thought. There are many different experiments done and placing more focus on the most important could help streamline the paper. For example, the section at line 193 feels a bit out of place since this line of investigation is not mentioned earlier. Another example is that the research questions Q1. and Q2. do not mention the trust-region although it is a large part of the paper.

Additionally, various phrases (or claims) in the paper could use more support or evidence. These are discussed more specifically in the Questions section.

I am also concerned about the link between feature rank and trust-region violation, a central point of the paper.
In Fig.4, the correlation between the prob ratios and dead neurons or prob ratios and feature rank seem weak. There is only a substantial dip in the prob ratios once the feature rank is near zero or the number of dead units is very high. In other words, only when the agent is doing very poorly can we see a strong relationship, which may limit the applicability of the finding. Ideally, one would avoid this regime since performance is poor to start with.

**Questions:**

I would be willing to revise my score based on answers to the following.

_Clarification questions_
- Line 308. Could you clarify how the average probability ratio is computed?
	As written, my interpretation is that, given some window of updates, you consider all ratios above $1+\epsilon$ and take their mean. Then, you do the same for the ratios less than $1-\epsilon$. Finally you divide the first mean ratio by the second mean ratio. Is this correct?
	If so, this seems potentially misleading because the means are computed conditionally, considering only ratios that are above 9or below) the clipping ratios. Then, we cannot tell what the mean ratio is overall, and whether there are many violations of the trust-region occurring. Moreover, ratios can be misleading since they may overemphasize small probabilities in the denominator.

	I would suggest also reporting how often the clipping criteria is violated. E.g. compute on collected states which proportion of them have a ratio outside the clipping region after a round of updates on a batch of trajectories.
	Perhaps it would also be useful to consider ratios less than 1 or greater than 1 seperately instead of computing a ratio.


- Line 164 Using the term "plasticity loss" as a metric is confusing because it also refers to the overall phenomena of neural networks being unable to learn additional tasks after training on other tasks.
Does this term refer to the capacity to fit random labels? In that case, I would use a term that is more descriptive e.g. random label trainability (RLT).
Also, how are these random labels generated in this context? In the original paper, a regression task was made since a value-based algorithm was used. With a policy-based algorithm, how is the plasticity evaluation task set up?


- Fig.1 Why does the preactivation norm plateau at 10^4?

- Which data is used to produce the box plots? The caption for Fig. 6 mentions "A boxplot includes 15 runs with the different epochs". Does this mean the box plot conains data across training times and epochs?

- In the box plots, how are outliers determined? We can see they lie outside the interquartile range indicated by the "whiskers" but what additional criteria is there?


- How is the PFO regularizer related to adding a regularizer in parameter space? Say $\ell_2$ regularization between the current parameters and the previous parameters? Basically, this seems similar in spirit to adding a target network to value-based methods.


- Line 94-99 (and elsewhere) "PPO introduces additional non-stationarity...": I do not think it would be fair to see PPO has more nonstationarity than other policy gradient methods. REINFORCE and PPO both have nonstationarity due to the changing policies. REINFORCE can also be interpreted as a trust-region-like method. Since REINFORCE is essentially gradient ascent on the expected return objective, we can view it as minimizing a linear approximation with a quadratic constraint (see paragraph containing eq. 5.5 in [4]). In this light, by doing a single optimization step before changing the surrogate objective, we could say REINFORCE has _more_ nonstationarity.


- Line 257. "The underlying assumption...approximately orthogonal..." Could you give some more support to this claim?
	As mentioned above, with state aggregation, this clipping strategy could be effective without having orthogonal features.


- Line 8-9: "...policy optimization methods which are often thought capable of training indefinitely". I would disagree with this claim. Deep RL algorithms have often be thought to be high-variance and unstable so I do not think it would be common to believe they can train indefinitely.
I suggest replacing this phrase and simply mention the phenonmenon has been underexplored in the context of on-policy policy optimization.


- The parameter norm has been linked to plasticity/trainability loss in neural networks. Is there a reason why the paper measures feature norms instead?





_Broader questions_
- The performance degradation from training more epochs could be attributed to difficulties of offline RL. As the number of epochs is increased, the more the procedure ressembles offlne RL---a setting in which standard RL algorithms are not suitable for. What do you think of this?


- I find the proposed solution (Proximal Feature Optimization) to be slightly unsatisfying, Generally, I can see how using regularization on the policy and features can mitigate the performance collapse but would this be simply delaying the problem? If we train for longer, would we still expect to see the same performance degradation?


- Interestingly, the value network is not the problem. It is the policy network that suffers from representation collapse. Do you have any thoughts about this? I think this could be an interesting avenue of investigation.


_Suggestions_

- I would suggest removing or summarizing the paragraph at line 193 into a single sentence.
	It is clear that the policy would be the same for all states if the features are constant. This would imply the policy is uniform. Currently the paragraph seems to explain this in a roundabout manner when a concise explanation suffices.

- A suggestion would be to try the Atari environemnts identified in [1] to benefit from plasticity injection. On many Atari games, little or no benefit was found, so it may be more meaningful to focus on those where there is more evidence for trainability loss.


- I would be careful about mentioning that low feature rank necessarily means that the representation is poor. Successful training of deep neural networks often involves a reduction in feature rank and the relationship between rank and performance can be complex [3].


- As a sidenote: even if there is a low dimensional representation, this does not necessarily imply the clipping criteria will be violated. For example, if we use state aggregation, then multiple states are clustered into one aggregate state. Then, the policy will be exactly the same for all these states, so if the policy exits the clipping region for one those states, there will no longer be any incentive to update the policy any further on any of them.


- A minor suggestion is to use stable rank [2] as a measure of the rank instead of effective rank since it is a continuous quantity in the singular values.




[1] "Deep Reinforcement Learning with Plasticity Injection" Nikishin et al.

[2] https://nickhar.wordpress.com/2012/02/29/lecture-15-low-rank-approximation-of-matrices/

[3] "An Empirical Study of Implicit Regularization in Deep Offline RL" Gulcehre et al.

[4] https://www.stat.cmu.edu/~ryantibs/convexopt-S15/scribes/05-grad-descent-scribed.pdf

**Limitations:**

These are discussed.

---

> ### Author Rebuttal · Authors · 2024-08-07
>
> We address the key points in the rebuttal and the remaining ones in a comment.
>
> >**Q1.** line 193 feels a bit out of place … investigation is not mentioned earlier … suggest removing or summarizing the paragraph …
>
> **A1. This information is relevant to practitioners in diagnosing the kind of collapse they face.** One common collapse or stagnation is due to entropy collapse and lack of exploration. In this work, we highlight that when collapse is associated with a high entropy, it may likely be due to collapsed representations. **We agree with the suggestion of the reviewer and will summarize this paragraph in a sentence.**
>
> > **Q2.** … research questions Q1. and Q2. do not mention the trust-region although it is a large part of the paper.
>
> **A2.** Q1 and Q2 serve as motivation for section 3.1, where they are answered. These are not the only questions addressed in the work. The trust-region exposition in section 3.2 is only motivated after answering Q1 and Q2 and observing collapse.
> We will move Q1 and Q2 inside section 3.1 to clarify their scope and reformulate section 3.2 with two questions, Q3 and Q4, to mirror 3.1 and ensure the question format covers all the main points of our work.
>
> >**Q3.** I would be careful about mentioning that low feature rank necessarily means that the representation is poor …  I am also concerned about the link between feature rank and trust-region violation …
>
> **A3.** Reviewer XWGd has also raised this concern. We have clarified this in GA1 and GA2 of our global rebuttal.
>
> > **Q4.a** how the average probability ratio is computed? …
>
> **A4.a** Yes, this is correct, and more details can be found in the Appendix lines 635-640.
>
> > **Q4.b** If so, this seems potentially misleading …
>
> **A4.b** The mean ratio without conditioning does not give information about learning or the trust region. The PPO-Clip objective increases the ratios of actions with positive advantages and decreases those with negative advantages until they reach the clip limit. Therefore, to have a signal for learning, we have to at least condition on the sign of the advantage. Still, a takeaway from the toy example is that taking the mean ratio of actions with, say, a positive advantage under bad representations would mix the ratios that suffer from interference (Figure 5 right) and would give a misleading average.
>
> Therefore, we found that the right way to quantify the trust-region violation is to condition on the violation itself (e.g., above $1+\epsilon$). We took the mean rather than the count because fully optimizing the PPO objective should push the actions until the clip limit, so it's not clear whether a high clip count would be worse than a lower one.
>
> >**Q4.c** they may overemphasize small probabilities in the denominator.
>
> **A4.c** This is true but has not been a critical issue for our analysis. In Figure 4 with ALE the smallest probability ratio is around 0.4, and in Figure 6, the largest excess ratios are between 1.5 and 2.5.
>
> >**Q4.d** I would suggest also reporting how often the clipping criteria are violated …
>
> **A4.d** Yes, we track this. As discussed above (A4.b) it’s not clear what the count of the clip fraction should be and it is not an interesting quantity for our study.
>
> >**Q4.e** consider ratios less than 1 or greater than 1 seperately
>
> **A4.e** We do separate them in Figure 4, where we look at the ratios below $1-\epsilon$ and isolate the causal relation around poor representations. For Figure 6, aggregating them into a ratio provides a better summary with larger plots.
>
> >**Q8.** How is the PFO regularizer related to adding a regularizer in parameter space? …
>
> **A8.** We can construct a spectrum ranging from regularizing the parameters to the outputs. The lower end regularizes the parameters like an L2 weight difference. The higher end regularizes the network's output like PPO clipping and almost like a target network in value-based methods. **PFO sits in the middle of this spectrum, where the regularization allows both the network's weights and final outputs to change without explicit constraints while maintaining a regularization of the feature space.**
> Also, PFO should mostly be seen combined with PPO, to extend the trust region to the feature space.
>
> >**Q12.** Is there a reason why the paper measures feature norms instead?
>
> **A12.** Other work we cite, like Lyle et al. (2024), has also looked at feature norms in addition to weight norms. We noticed that the model weights were consistently and steadily increasing regardless of collapse; however, the feature norm showed a sudden jump around collapse. **We found this to be a more apparent symptom that researchers and practitioners would want to investigate further.**
>
> >**Q13.** degradation from training more epochs could be attributed to difficulties of offline RL
>
> **A13.** Off-policiness could be a good characterization here as the algorithm has access to the action probabilities of the collected data and the policy that collected it more generally.
> In PPO the more epochs performed, the more the on-policy approximation becomes off-policy. However, the PPO-Clip objective is supposed to be robust to slight changes in the number of epochs, as it only depends on epsilon. Therefore, **we view observing a drastic collapse by moving from four to six epochs in Figure 1 not as a failure of standard RL algorithms in the off-policy setting but as a failure of the learning dynamics of PPO-Clip.**
>
> > **Q14.** I find the proposed solution PFO to be slightly unsatisfying, …
>
> **A14.** The first purpose of PFO in this work, as the reviewer mentions, is to show that explicitly regularizing the feature dynamics builds robustness to trust-region and performance collapse.
> Now as with most empirical studies and regularizations using a coefficient, there is no guarantee on how long PFO will stand the test of time, but we hope that our paper encourages further theoretical studies and only then will we have provably mitigated the problem.

---

> ### Author Response · Authors · 2024-08-07
> **Addressing the remaining points**
>
> >**Q5.** Using the term "plasticity loss" as a metric is confusing because it also refers to the overall phenomena …
>
> **A5.** Yes, we use this metric to compute the loss from fitting a random initialization's trajectories. It is defined in the background (114-122) and to fit the actor we use a KL divergence.
> In the paper, **we distinguish between the metric called plasticity loss and the phenomenon called loss of plasticity.**
> This is the value of a loss, that’s why we call it plasticity loss. However, the phenomenon has also been termed plasticity loss by Lyle et al. (2024), so we agree that reusing it for the loss can be confusing.
> We propose to state our distinction between the two forms in our background, making it less confusing. Otherwise, we can change the loss term to to random fit error.
>
> >**Q6.** Fig.1 Why does the preactivation norm plateau at 10^4?
>
> **A6.** Typically, the preactivation norm increases until all the neurons of the feature layer are dead (in the case of ReLU). After that, all the gradients for the weight matrices are 0, so the norm flattens out.
>
> >**Q7.a** Which data is used to produce the box plots? …
>
> **A7.a** Each run is summarized with an average over the last 5% of its progress. The 15 runs consist of 3 epoch values (4, 6, 8) x 5 seeds. This is presented in lines 307-308 and further detailed in the appendix lines 641 - 644, 658-659.
>
> >**Q7.b** In the box plots, how are outliers determined? …
>
> **A7.b** The whiskers are determined by the highest observed datapoint below Q3 + 1.5 IQR (similarly for the lower one) (default of matplotlib). The outliers are points outside of the whiskers.
>
> We thank the reviewer for the question and will add this information to Appendix lines 659+.
>
> >**Q9.** I do not think it would be fair to see PPO has more nonstationarity than other policy gradient methods…
>
> **A9.** We agree that, from this perspective, REINFORCE could be considered as more nonstationary than PPO. **We will remove this "more nonstationary" observation; it is not critical to the work.**
> The critical point of in work is that PPO performs multiple epochs on the same data and, in this sense, can be more prone to "overfitting" to previous experiences, which can become worse with more epochs.
>
> >**Q10.** Line 257. "The underlying assumption...approximately orthogonal..." Could you give some more support to this claim? …
>
> **A10.** The presumed orthogonality here is between the states where the policy should act differently, like with two different groups of aggregated states. Otherwise, for the states where the policy should be similar (same group of aggregated states), some alignment in the features and a reduction of dimensionality are indeed desired for generalization. However, with clipping, this alignment should be associated with the right relative feature norm; otherwise, the clipping will be violated, as observed in Figure 5, where a larger alpha would make the relative deviation of the action probabilities larger.
>
> >**Q11.** Line 8-9: "...policy optimization methods which are often thought capable of training indefinitely"...
>
> **A11.** This is indeed arguable. We believe that trust region methods aim to overcome this high variance and instability.
> However, this is not a critical claim, and we can remove it. We thank the reviewer for the suggestion.
>
> >**Q15.** Interestingly, the value network is not the problem. …
>
> **A15.** **Yes, this is one key point of our work.** Policy networks also suffer from representation collapse independently of value networks, as noted in the caption of Figure 1.
> Our intuition and the main motivation of this work is that, similarly to value networks, policy networks are also subject to the non-stationarity of their inputs and outputs (background, lines 88+), and optimizing deep neural networks under non-stationarity is known to cause issues.
>
> >**Q16.** A suggestion would be to try the Atari environemnts identified in [1] to benefit from plasticity injection…
>
> **A16.** Phoenix is one of the environments we have in common and where we see the most impactful results.
>
> For this work, we did not want to select environments with prior knowledge of existing collapse in value-based methods to demonstrate the collapse in policy optimization; we rather used the unbiased sample recommended in Atari-5 (Aitchison et al. 2023) because **we did no want to unconsciously cherry-pick environments, where certain approaches may perform better.** Moreover, we would like to highlight that, on Atari, we could not find any large-scale study of on-policy RL algorithms and PPO demonstrating the collapse of the actor’s representation from which we could pick our environments at the time of submission.
>
> >**Q17.** A minor suggestion is to use stable rank [2] as a measure of the rank …
>
> **A17.** **We have tracked 5 different measures of the rank, spanning continuous or discrete and relative or absolute metrics.** We conducted a thorough comparison of these in Appendix E.

---

> > ### Comment · Reviewer_YzGo · 2024-08-10
> >
> > I appreciate the clarifications and explanations.
> > After reading responses to other reviewers, I have to agree that certain points of Reviewer qwmj are still concerning, including the ones around the performance of the baseline. I am also still a bit underwhelmed by the fact that trust-region failures seems to only occur when the representation has already collapsed and the performance is poor.
> > I will keep my score at present.

---

> ### Author Response · Authors · 2024-08-11
>
> We are happy that our clarifications addressed all your previous concerns.
>
> > I have to agree that certain points of Reviewer qwmj are still concerning, including the ones around the performance of the baseline.
>
> **We understand the concern; however, this is a matter of clarification.** We take the thoroughness of our implementation with utmost importance and **have clarified all the necessary bits in our reply to the reviewer,** which we summarize below:
> 1.  Recall that **our results hold on Atari and we have already replicated them with the CleanRL** implementation the reviewer is using.
> 2.  Recall that **our implementation for MuJoCo is based on recent implementations** for continuous action spaces influenced by seminal work studying implementation details in PPO and that **we find this setting more relevant to our audience.**
> 3.  **Replicate our implementation in CleanRL, which is still exhibiting collapse,** so the reviewer can more easily inspect the code.
> 4.  **Adapt the default CleanRL implementation the reviewer is using with a fully state-dependent action distribution and observe collapse** in the setting they are familiar with and interested in.
>
> > I am also still a bit underwhelmed by the fact that trust-region failures seem to only occur when the representation has already collapsed and the performance is poor.
>
> **Unfortunately, this is a fact.** We extensively investigated the link in the early stages of the project to find strong evidence throughout training, **which to us was also the more exciting claim to prove; however, countless examples made us realize that the link did not necessarily hold throughout training but only around the collapsing regime.**
> Using only positive examples or a narrow view of the correlations, to **claim that such a link is present throughout training would be extremely misleading and has been an issue in offline RL.**
> The work by Gulcehre et al. (2022) that the reviewer references is an excellent example of this. They conclude that previous exciting associations between performance and rank in offline RL are misleading and do not hold when considering a large experimental scope.
>
> Our conclusion is that trust-region violations become more evident when representations are about to collapse. **This does not mean that one should only be concerned about the link when performance is starting to deteriorate.** **The representations don’t collapse all of a sudden; they deteriorate throughout training until they reach collapse. So, mitigating representation degradation should happen throughout training and not only when around the collapsing regime.**
>
> We hope these clarifications address your concerns. Thank you for your input, and we look forward to your feedback.

---

### Official Review · Reviewer_qwmj · 2024-07-15

**Soundness:** 2
**Presentation:** 3
**Contribution:** 2
**Rating:** 3
**Confidence:** 4

**Summary:**

This paper presents a series of experimental studies to diagnose the learning issues of PPO under non-stationarity in Atari-5 and MuJoCo tasks. Based on the results, this paper establishes a connection among feature rank/norm, plasticity loss and trust region violation and learning performance. To mitigate the issues in feature representation, this paper proposes Proximal Feature Optimization (PFO) and demonstrates its effectiveness in mitigating the learning issues under non-stationarity. Besides, the effects of sharing network parameters and adapting Adam are also evaluated in the same context.

**Strengths:**

- Most existing works on plasticity loss or learning under non-stationarity focus on value-based RL or off-policy AC methods. In contrast, this paper focuses on on-policy algorithms, mainly PPO, which helps the RL community gain a better understanding of this problem.
- The experimental studies contains the results with multiple metrics, i.e., feature rank, feature norm, plasticity loss, dead neuron, excess ratio. These results will be a useful reference to the audiences.
- This paper reveals the connection among factors like feature rank, plasticity loss, excess rate, and learning performance.

**Weaknesses:**

- The writing is almost clear. The empirical results and conclusions can be better organized and made prominent.
- The proposed method PFO is closely related to DR3 (Kumar et. al, 2022) and RD [1], both of which propose regularizing the feature representation. More discussions are necessary.
- Although PFO are demonstrated to mitigate the feature rank, plasticity loss, excess ratio in Section 4, it does not bring clear improvement in terms of learning performance like episode return (according to Figure 18 to Figure 27). This cripples the significance of PFO and also the potential causal connection between the learning issues investigated and the learning performance.

---

Reference:

[1] Reining Generalization in Offline Reinforcement Learning via Representation Distinction. NeurIPS 2023

**Questions:**

1. The epsiode return curves in Figure 24 and Figure 26 (i.e., for MuJoCo Hopper) look strange. Is there any explanation to the collapse? A regular implementation of PPO should work in Hopper when the epoch num is 10.
2. The authors mentioned 4 MuJoCo tasks and 5 Atari games in the Experimental Setup paragraph. However, it seems only 3 Atari games (Phoenix, NameThisGame, Gravitar) and 2 MuJoCo tasks (i.e., Humanoid, Hopper) are used for the evaluation in Section 4. Are there more results on the remaining tasks/games?

**Limitations:**

The limitations are discussed in Section 5. However, one major limitation of this work is that the results in this paper do not provide sufficient support for the point that "mitigating the learning issues in feature rank, plasticity loss of PPO can improve the learning performance in terms of episode return".

---

> ### Author Rebuttal · Authors · 2024-08-07
>
> >**Q1.** … PFO is closely related to DR3 and RD …. More discussions are necessary.
>
> **A1.** We thank the reviewer for pointing these out. Although **these regularizations emerge from value-based offline-RL challenges**, a discussion of their similarities with PFO can be valuable to our audience. We can add the following to our camera-ready version.
>
> Other interventions regularizing feature representations have been studied in value-based offline RL. Kumar et al. (2022) propose DR3, which counteracts an implicit regularization in TD learningby minimizing the dot product between the features of the estimated and target states.
> Ma et al. (2024) propose Representation Distinction (RD) which tries to avoid unwanted generalization by minimizing the dot product between the features of state-action pairs sampled from the learned policy and those sampled from the dataset or an OOD policy.
>
> **Both are related to PFO as the methods directly tackle an undesired feature learning dynamic, but there is no motivation for DR3 or RD in online RL, and PFO is conceptually different.**
> The implicit regularization that DR3 counteracts is not present in on-policy RL as shown by Kumar et al. (2022) in the SARSA experiment, and PFO differs from DR3 as it extends a trust region rather than counteracts an implicit bias. Whence the different implementations: PFO regularizes the state features between two consecutive policies, as opposed to consecutive ones of the same policy in DR3.
> Similarly, the overestimation studied by Ma et al. (2024) in the vicious backup-generalization cycle is broken by on-policy data. PFO’s motivation to bring the trust region to the feature space resembles RD’s motivation to bring the overestimation constraint to the feature space. However, they are again different in their implementation as RD regularizes state features between the learned policy and the dataset policy or an OOD policy.
> **Finally, PFO is applied to the actor, while both DR3 and RD are applied to the critic.**
>
> >**Q2.** … do not provide sufficient support for the point that "mitigating the learning issues in feature rank, plasticity loss of PPO can improve the learning performance in terms of episode return".
> >PFO … does not bring clear improvement in terms of learning performance like episode return ... This cripples the significance of PFO and also the potential causal connection …
>
> **A2.** First, we claim that **the causal connection holds around the collapse regime**, not necessarily throughout training, and that discovering and **describing such a relation only around collapse is nontrivial.** We have clarified these arguments in GA1 and GA2 in the global rebuttal.
>
> Second, we distinguish between consistently improving the best run performance (i.e. claiming to be “better” than PPO) and improving the aggregate performance across multiple runs and hyperparameters, e.g., robustness to collapse in our case. The limitation the reviewer raises is about the former, however this work argues about the latter.
>
> In this sense, the **significance of PFO is realized by a) successfully mitigating collapse** as observed in Figure 6 with a significantly higher median performance indicated by the black lines, and b) improving the representation metrics and obtaining a better trust region, therefore **strengthening the relation we’ve drawn** between representation, trust region, and performance around collapse.
>
> Ultimately, **this improvement can result in best performance improvement as well but less consistently as it requires running for long enough** to observe a collapse with the standard (tuned) hyperprameters. This is the case for NameThisGame in Figures 18 and 22, Humanoid in Figures 19 and 25, and Hopper in Figures 20 and 26, where PFO performs better than the baseline PPO. This is not a central claim of the paper, as it requires a larger computational budget to show for all environments.
> **To strengthen these points, we have added Figure 34 to the rebuttal PDF** where we show that on ALE/Phoenix with the tuned standard hyperparameters, the agent collapses when training for longer and this is mitigated with PFO, which attains the best performance in that case.
>
> >**Q3.** The epsiode return curves … for MuJoCo Hopper look strange.  Is there any explanation to the collapse? A regular implementation of PPO should work in Hopper when the epoch num is 10.
>
> **A3.** We use the same hyperparameters as the original PPO implementation, which are also the default ones in popular codebases (as noted in lines 140-143).
> Our setting differs from a “regular” implementation in the following main points, which can help explain why a collapse is not typically observed in previous work.
> 1.  We train for longer and collapse always happens after the default 1M steps. Dohare et al. (2023) also observe collapse when training for longer.
> 2.  We do not decay the learning rate (line 154) we are interested in agents that can ideally train indefinitely.
> 3.  we parameterize the action space by a TanhNormal distribution with both state-dependent mean and variance following the implementation of Haarnoja et al. (2018). (lines 614-616).
>
> >**Q4.** … 4 MuJoCo tasks and 5 Atari games in the Experimental Setup ... However … only 3 Atari games … and 2 MuJoCo tasks … used in Section 4. Are there more results on the remaining tasks/games?
>
> **A4**. Indeed, as noted in lines 305-306, while we used 4 MuJoCo tasks and 5 Atari games to demonstrate the collapse phenomenon with enough evidence, we did not use all of the environments for the evaluation.
> We selected the environments where the collapse was the most consistent to test multiple interventions while maintaining a reasonable compute budget (this is common practice like in Gulcehre et al. 2023 and Kumar et al. 2021). We believe this to be a right tradeoff between claiming that interventions are necessary to prevent collapse and providing enough insights about several types of interventions.

---

> > ### Comment · Reviewer_qwmj · 2024-08-08
> > **Response to the rebuttal**
> >
> > I appreciate the authors' careful response and the additional results.
> >
> > > More discussion about PPO implementation
> >
> > Thanks for the implementation details provided by the authors. According to Figure 8 (PPO at MuJoCo without intervention) and Figure 24 (PPO at MuJoCo-Hopper with intervention), the results show that (1) the PPO implementation used in this work exhibits collapse in HalfCheetah and Hopper with epoch=10 basically after 1M steps (Figure 8), and the collapse remains when different kinds of intervention are applied.
> >
> > I still have a concern about the validity of the PPO implementation used in this work. After a quick run with github CleanRL implementation of PPO, I did not observe collapse for HalfCheetah and Hopper after 1M steps.
> >
> > With the details provided by the authors, I realize that it should stem from the difference in implementation. On my side, the learning rate gradually decays to 0.1 * [init_learning_rate], and the variance vector is state-independent (I think this is a convention). The authors mentioned "We do not decay the learning rate (line 154) we are interested in agents that can ideally train indefinitely", which does not quite make sense to me as "train indefinitely" is meaningless after convergence or collapse in a single-task RL training process. And a proper learning rate decay (e.g., decay to 0.1 * [init_learning_rate]) should also allow the agent to learn within a long horizon.
> >
> > > More discussion about evaluation environment choice
> >
> > The authors mentioned,
> > "We selected the environments where the collapse was the most consistent to test multiple interventions while maintaining a reasonable compute budget".
> >
> > According to Figure 8, it looks like the collapse is more severe in Ant and HalfCheetah than in Humanoid. This turns out to be a bit contradictory with the response above that  Humanoid is used for evaluation but Ant and HalfCheetah are not included.
> >
> > > More discussion about the collapse
> >
> > The results in Figure 24 show that all the interventions fail to address the collapse in Hopper. And based on my personal experience, PPO with proper implementation does not collapse in MuJoCo tasks (as mentioned above). My main concern remains after the rebuttal.

---

> > > ### Author Response · Authors · 2024-08-11
> > >
> > > We sincerely thank the reviewer for promptly engaging with our rebuttal and clarifying their main concern.
> > > **We take the thoroughness of our implementation with utmost importance** and understand the concern of the reviewer. **We have been preparing a complete answer with sufficient experimental evidence to address your concerns.**
> > >
> > >
> > > We address the implementation concerns in MuJoCo below, but first, we would like to highlight that **the main results in our paper are also shown with the Atari environments** and that we have included a strong replication of our results in Atari with the **CleanRL codebase. It exhibits the same collapse we observed with our implementation.** As the reviewer, we wanted to rule out the collapse happening because of any bugs or implementation details with as much confidence as possible.
> > > In addition, note that we have referenced Dohare et al. (Overcoming Policy Collapse in Deep Reinforcement Learning, 2023), who also observed a collapse on MuJoCo.
> > >
> > > > I did not observe collapse for HalfCheetah ... I realize that it should stem from the difference in implementation.
> > >
> > > We understand the concern and have taken two actions to help the reviewer increase their trust in our results:
> > >
> > > 1. **Clarify the motivation behind our setting, replicate it on CleanRL, and observe collapse:**
> > > In addition to sharing our code with the reviewers, we replicated it in CleanRL as we did with Atari so that it is easier for a reviewer familiar with CleanRL to inspect.
> > > We would like to highlight that our setting is fully described in Appendix B.2, and its differences from **CleanRL stem from using recent implementation details for continuous action spaces since the original PPO paper** (as CleanRL is a faithful replication of it). Several works we cited have studied the implementation details of PPO and have given recommendations that shape how PPO agents are implemented these days (Andrychowicz et al., 2021; Engstrom et al., 2020).
> > > **We believe that this setting is more relevant to the community than the initial implementation of PPO in 2017.**
> > > A table with all the differences can be found below.
> > > 2.  **We have taken the ClearnRL implementation used by the reviewer and applied minimal changes that resulted in a collapse.**
> > > - Remove value loss clipping. This is an unnecessary trick that complicates the analysis (we are primarily interested in the actor). It is also not recommended by Andrychowicz et al. (2021)
> > > - Make the standard deviation output of the action space dependent on the state. Since our research focuses on the problems related to state representation, the study inherently makes more sense if the standard deviation is state-dependent. **If the standard deviation were state-independent, it would not be influenced by the representation collapse we are investigating, thus undermining the core premise of our study.**
> > >
> > > **With this, we have obtained similar collapsing curves as seen in our submission with and without an annealed learning rate.** We have communicated the implementations and training curves to the AC.
> > >
> > > > The authors mentioned "We do not decay the learning rate (line 154) we are interested in agents that can ideally train indefinitely" …
> > >
> > > Moreover, let us clarify the wording and context of our work, we acknowledge that we should have provided a more detailed explanation in our initial submission.
> > >
> > > **We are interested in an online learning setting where the agent receives the experiences continuously, and it is not straightforward to determine when the learning process should terminate.** The continual and online learning aspect of our experimental setup makes it challenging to apply standard annealing schedules, and a primary goal of our research is to maintain the plasticity of representations. Therefore, we choose not to anneal the learning rate to better study the behavior of representation collapse under these conditions.
> > >
> > > **The environments we use are single-task environments to ablate additional MDP non-stationarity, but they are complex enough for the agents to keep improving when trained for longer than our common benchmark limits.** In the tasks we have tried, the policies trained with a constant learning rate collapsed while apparently still improving and before stabilizing/converging.
> > >
> > > **We do not claim that our approach is the only way to study this phenomenon**, and we acknowledge that annealing the learning rate could be a method to delay or prevent collapse.
> > >
> > > In the additional runs we have provided, we observe collapse even when annealing the learning rate when training for long enough.
> > >
> > > > Choice of eval environments
> > >
> > > **We have communicated the results of all tasks to the AC. This should resolve this issue.**
> > >
> > > We considered the severity of the collapse regarding both Tanh and ReLU activations. The default hyperparameters of Ant don’t collapse on either, and we favored Humanoid for its large action space over HalfCheetah, which seemed redundant to us with Hopper.

---

> ### Author Response · Authors · 2024-08-11
>
> | Implementation detail          | CleanRL’s default                             | ClearnRL's default adapted | Our implementation                                  |
> | ------------------------- | --------------------------------------------- | -------------------------------------------- | ---------------------------------------------------------------------- |
> | Network output            | State-dependent mean / State-independent std | same / State-dependent std                  | same / State-dependent std                                            |
> | Transformation of the std | exponential                                   | same                                         | Softplus (recommended by Andrychowicz et al., 2021)                    |
> | Action distribution       | Normal                                        | same                                         | TahnNormal (recommended by Andrychowicz et al., 2021)                  |
> | Reward transforms         | Normalize and clip                            | same                                         | None (to keep the default non-stationarity)                            |
> | Observation transforms    | Running normalization and clip                | same                                         | Normalization at initialization (to keep the default non-stationarity) |
> | Layer initialization      | Orthogonal with custom scale                  | same                                         | Default PyTorch initialization                                         |
> | Learning rate annealing   | True                                          | True, False (collapses for both)             | False                                                                  |
> | Value loss clipping       | True                                          | False (out of scope)                         | False (recommended by Andrychowicz et al., 2021)

---

### Author Rebuttal · Authors · 2024-08-07

We thank the reviewers for their thorough and insightful reviews. We are glad that the reviewers appreciate the novelty and impact of assessing loss of plasticity in on-policy optimization and its connection to PPO’s trust region and acknowledge our thorough experimental setup.

We have **addressed all the concerns and questions of the reviewers** in the individual rebuttals.
We hope that this helps the reviewers **increase their support for the paper otherwise, kindly ask the reviewers to point out any remaining issues.**

In addition, we use this section of the rebuttal to group and address **some common questions and concerns raised by reviewers. We reference those in our individual answers to the reviewers who raised them.**

**GQ1. The distinction between low-rank and poor representations**

>**YzGO Q3.** I would be careful about mentioning that low feature rank necessarily means that the representation is poor. Successful training of deep neural networks often involves a reduction in feature rank and the relationship between rank and performance can be complex Gulcehre et al. (2022).
**XWGd Q2a.** isn't it expected that the policy learns to lose rank, by compressing state information to only correlate with actions?

**GA1.** Indeed, we reference the work of Gulcehre et al. (2022) in our work and highlight in lines (236-238) that the relation we draw between the representation dynamics, the trust region, and the performance **primarily holds around the poor representation regime which we characterize by an extremely low rank, and not necessarily throughout training.** (lines 236-238: “We observe no significant correlation in the regions where the representation is rich … but an apparent decrease of the average probability ratios below $1 − \epsilon$ is observed as the representation reaches poor values”).

It may **not be straightforward to draw a line between low-rank representations beneficial for generalization and extremely low-rank representations causing aliasing**, as also acknowledged by Gulcehre et al. (2022) at least in offline value-based RL (“Unfortunately, reasoning about what it means for the rank to be too low is hard in general”), but for environments like Atari, **our figures seem to draw the line at single-digit ranks, which can be related to the action space of dimension 8+.**

We will further clarify this distinction in the same paragraph (lines 236-238) and in the captions.

**GQ2. The scope of the connection between representations and the trust region, and the significance of this scope.**

>**YzGO Q3.** I am also concerned about the link between feature rank and trust-region violation, a central point of the paper. In Fig.4, the correlation between the prob ratios and dead neurons or prob ratios and feature rank seem weak. There is only a substantial dip in the prob ratios once the feature rank is near zero or the number of dead units is very high. In other words, only when the agent is doing very poorly can we see a strong relationship, which may limit the applicability of the finding. Ideally, one would avoid this regime since performance is poor to start with.
> **Qwmj Q2**  ... This cripples the significance of PFO and also the potential causal connection …

**GA2.** As mentioned in the previous answer we claim that the causal connection we draw between the representation dynamics, the trust region, and the performance primarily holds around the collapse regime and not necessarily throughout training.

**Yet discovering and describing such a relation only around collapse is nontrivial.** First, this gives **evidence that this regime is often attained and should be avoided.** Second, it **gives important insights into the failure mode of the popular PPO algorithm**, whose trust region is highly dependent on the representation quality, and **more generally about current trust-region methods, which only constrain the output probabilities.**
The discovery of this link can further drive research on training deep networks in non-stationary settings and influence the design of future trust-region methods (e.g., PFO forms a trust region in the representation space as well).

---

### Comment · Area_Chair_MRD7 · 2024-08-12
**Important inquiry regarding claims and positioning relative to existing works**

I appreciate the thorough experiments that reviewers are already assessing, but something has come up for which I wanted to give you an opportunity to respond.

The following two sentences of the abstract seem to be important claims/motivations for the paper.

(1)
> it has been overlooked in on-policy policy optimization methods which are often thought capable of training indefinitely

(2)
> revealing that PPO agents are also affected by feature rank deterioration and loss of plasticity.

However, there are several papers, including one well-recognized in the loss of plasticity community, that already refute (1) and establish (2):

Dohare, S., Sutton, R. S., & Mahmood, A. R. (2021). Continual backprop: Stochastic gradient descent with persistent randomness. arXiv preprint arXiv:2108.06325.

Dohare, S., Hernandez-Garcia, J. F., Rahman, P., Mahmood, A. R., & Sutton, R. S. (2023a). Maintaining plasticity in deep continual learning. arXiv preprint arXiv:2306.13812.

Dohare, S., Lan, Q., & Mahmood, A. R. (2023b). Overcoming policy collapse in deep reinforcement learning. In Sixteenth European Workshop on Reinforcement Learning.

I would like to hear from the authors about how they plan to position their claims/work within the context of this existing literature.

Furthermore, there are results given on four Mujoco tasks in the latter paper, which are highly relevant for this paper. It would be natural to build on these results. However, these existing results are not acknowledged. It is important because the acknowledgment would have allowed the reviewers to compare your results. As we look closely now, some of the results do not match. For example, Dohare et al. (2023b) show that adam-equal-betas do not collapse on Hopper, whereas in your experiment, it does. This baseline is also not used in the two new Mujoco environments that were submitted later. PPO+L2 is another important and effective baseline in the latter work that seems relevant to your work but is not mentioned or compared.

Moreover, claims regarding loss of plasticity are an important part of the paper, but references to this literature seem insufficient, except for Kumar et al. (2021), Nikishin et al. (2023), and Lyle et al. (2022, 2024). A wider coverage of existing work, e.g., the above-listed ones and the following, would help better understand your work in the context of the literature.

Sokar, G., Agarwal, R., Castro, P. S., & Evci, U. (2023). The Dormant Neuron Phenomenon in Deep Reinforcement Learning. In International Conference on Machine Learning 2023.

Kumar, S., Marklund, H., & Van Roy, B. (2023). Maintaining plasticity via regenerative regularization. arXiv preprint arXiv:2308.11958.

Abbas, Z., Zhao, R., Modayil, J., White, A., & Machado, M. C. (2023). Loss of plasticity in continual deep reinforcement learning. In Conference on Lifelong Learning Agents (pp. 620-636). PMLR.

---

> ### Author Response · Authors · 2024-08-13
> **Rebuttal and concerns about misleading claims**
>
> **We sincerely thank the AC for their commitment to a thorough review process** and for including us in the discussion of important concerns that were raised during the private discussions between the reviewers that we would not have otherwise had an opportunity to respond to. **This clarifies the recent comments we have received and allows us to discuss them constructively**. We hope to address them during the time left for the discussion phase.
>
> > several papers that already refute (1): it has been overlooked in on-policy policy optimization methods which are often thought capable of training indefinitely… and establish (2) revealing that PPO agents are also affected by feature rank deterioration and loss of plasticity.
>
> **We respectfully disagree. These works do not refute (1) nor establish (2). This is a misleading claim.**
>
>
> First, the “it” in (1) refers to “networks … exhibit a decrease in the rank of their representations, a decrease in their ability to regress to arbitrary targets, called plasticity” (line 5 abstract, and lines 25-56 introduction.) and “raises the question of how much PPO agents are impacted by the same representation degradation attributed to non-stationarity.” (lines 34-36 introduction).
>
> We agree that the “thought capable of training indefinitely” bit is controversial and that it is common to believe the opposite. We discussed this with reviewer YzGo and mentioned that it was not a critical finding of the paper and that we would remove it.
>
>
> However, **none of the works mentioned discusses the representation degradation of PPO, i.e., the “overlooked” bit of (1) and the main point of (2).**
>
> **Dohare et al. (2021)** have **a single RL experiment** with an environment **with non-stationary transition dynamics** and **only track online performance**. There is **no curve about representation degradation** (representation rank, dead units, feature norm, etc.), and the **“plasticity loss” refers to the degradation of the online performance** of this environment with non-stationary dynamics. This confuses assessing the ability to learn a sequence of optimal policies with the evolution of the learning network to fit the same arbitrary function, which we measure with the random label fitting capability.
>
> **Dohare et al. (2023a)** track dead units and effective rank but **only in the supervised learning experiments, confirming our point that representation degradation is overlooked in online RL. They also perform the same experiment as Dohare et al. (2021), using a non-stationary environment** to measure only online performance.
>
> **Dohare et al. (2023b)** have experimented in stationary online RL with PPO as we do, but again, they **only track online performance in the complex tasks (MuJoCo), with no figures of representation degradation and no figures of plasticity loss. With all due respect, all the claims made in the paper about plasticity appear to be speculations.** Representation is plotted once for a toy example, but it is one-dimensional, so already collapsed and does not show an evolution from rich to poor.
>
> These works are important in developing a better understanding of training deep neural networks under non-stationarity; however, they do not specialize in the specific topic of representation dynamics we discuss in our work. Importantly, **it is specifically tracking representations that allows us to draw a connection between representation and trust region degradation.**
>
> >  Mujoco tasks in the latter paper, which are highly relevant ... natural to build on these results. However, these existing results are not acknowledged. It is important because the acknowledgment would have allowed the reviewers to compare your results.
>
> **To the best of our knowledge, Dohare et al. 2023b do not provide code with their paper, and their setting is not fully described.** Several implementation details are missing, e.g., whether they decay the learning rate, the value of the Adam $\beta$ hyperparameters used for the plots (unlike for L2, which is well described), and the action distribution used.
> Furthermore, some of the design choices in their setting are unusual, e.g., a 4x larger minibatch size effectively dividing the number of gradient updates per batch by 4 (less overfitting) and no gradient norm clipping.
> **For these reasons and because we had a more modern setting in mind with an action space fully dependent on the representations, basing our implementation on a replication of theirs did not seem like a natural choice to us.**
>
> It is also not clear if highlighting their specific results on MuJoCo or basing our implementation on theirs would have provided a stronger reference to compare our results. **To the best of our knowledge, the paper has not been replicated, the code is not available, and the runs have not been thoroughly reviewed. In fact, reviews of the paper in EWRL16 state “reviewer points out some issues with the empirical results”.**

---

> > ### Author Response · Authors · 2024-08-13
> > **Continuation**
> >
> > > some of the results do not match ... adam-equal-betas do not collapse on Hopper, whereas in your experiment, it does.
> >
> > **This is likely due to the difference in settings. Our code and run histories are available for inspection, but it does not seem like theirs are.**
> > Moreover, it’s not clear what values of the Adam moment coefficients Dohare et al. (2023b) used. It seems that these were tuned per environment, and while their narrative suggests decaying the moments faster, i.e. decreasing the coefficients, they tuned $\beta_1$ on values strictly larger than its default.
> >
> > In our experiments, we followed their narrative and set the second-moment coefficient to match the smaller first-moment coefficient. This is fair to the other interventions, which were also not tuned.
> >
> > **Like any of the other interventions, we do not deny that further tuning the coefficients can yield better results.**
> >
> > > This baseline is also not used in the two new Mujoco environments ...
> >
> > **We sincerely apologize for this.** We ran them but used an older version of the script to generate the figures. **This is now fixed, and the updated plots can be found at the same link. We observe the same results as in the submission, that the intervention is not sufficient to prevent collapse.**
> >
> > > PPO+L2 ...
> >
> > It is important to note that the loss of plasticity subfield is rapidly growing, and several solutions are being suggested; however, no established solutions or baselines exist yet.
> >
> > If we claimed to have designed a new SOTA solution, we would have to run as many of the previously established solutions as possible to compare against them. However, this is not the purpose of this work, which is centered around building understanding. As mentioned in this discussion phase, **PFO, among the other interventions we have tried, is used to strengthen the connection we draw between representation, trust region, and performance. We found that none of the interventions suggested before directly targeted the degrading representations symptom with a connection to trust region, so we found that adding PFO to the interventions would strengthen the narrative.**
> >
> > **Our narrative does not focus on the norm of the weights**; instead, we focus on the norm of the features because the collapse happens at the feature level, which depends on both inputs and the model architecture; **therefore, weight decay did not seem like a crucial intervention to prioritize.**
> >
> > > claims regarding loss of plasticity are an important part of the paper, but references to this literature seem insufficient,
> >
> > > A wider coverage of existing work, e.g., the above-listed ones and the following, would help better understand your work in the context of the literature.
> >
> > Given an extra page, we are happy to cite and relate to all the aforementioned works to give more context to our work.
> >
> > **Kumar et al. (2023)** provide a comprehensive comparison and categorization of methods used to mitigate plasticity loss in continual supervised learning, it is not clear if solutions in this setting translate to solutions in RL; however, this setting is an important stepping stone towards maintaining plasticity.
> > **Sokar et al. (2023)** provide an alternative characterization of plasticity loss in RL using dormant neurons, however, like Lyle et al. (2022), their study only includes value-based methods.
> >
> > **Abbas et al. (2023)** study representation metrics such as feature norms like our work but in value-based methods. They observe a sparsification of the norm while we, taking the dying neurons out of the equation, find that the preactivations blow up.
> >
> > **Overall, the findings in these works and our work are complementary, and further discussing them on an additional page of related work is mutually beneficial.**
> >
> >
> > ### Conclusion
> >
> > > I would like to hear from the authors about how they plan to position their claims/work within the context of this existing literature.
> >
> > **We commit to discussing all the related work raised in this rebuttal, however, we strongly believe that this does not change the position of our work,** which provides a complementary view of representation degradation and plasticity loss in on-policy policy optimization and a novel connection to trust region methods explored both empirically and theoretically. **The plasticity community is growing with great complementary perspectives that we hope to add to. We express strong concerns against the misleading private claims raised using the line of work by Dohare et al. to undermine the contributions of our work.**

---

> > > ### Comment · Area_Chair_MRD7 · 2024-08-13
> > >
> > > Thanks for responding promptly. I am here to help you and commit to my best effort to resolve the confusion fairly. Let us continue a respectful and productive conversation to help us resolve the confusion.
> > >
> > > > This clarifies the recent comments we have received and allows us to discuss them constructively.
> > > > We express strong concerns against the misleading private claims raised using the line of work by Dohare et al. to undermine the contributions of our work.
> > >
> > > I believe the above concerns are unwarranted. The inquiries were mine and not covered by other reviewers. Sometimes, I see meta-reviews bringing new concerns that the authors cannot respond to. Hence, I am bringing these up with you for clarification as soon as I discover them. I do not see any of the recent comments by reviewers reflecting these concerns, and hence, no reason for the authors to be concerned.
> > >
> > > About the content of the response:
> > >
> > > While you claim that (1) is not refuted, you also agree to remove “thought capable of training indefinitely,” which is part of (1). Hence, claim (1) does not stand as it is. It was already shown to be not true.
> > >
> > > Moreover, the “it” of (1) in the abstract is described as the following phenomenon:
> > >
> > > > previous works have observed that networks in off-policy deep value-based methods exhibit a decrease in representation rank, often correlated with an inability to continue learning or a collapse in performance.
> > >
> > > It is the “an inability to continue learning or a collapse in performance” part that the works by Dohare et al. already show. Hence, a more nuanced and more accurate claim could be something along the lines of “although an inability to continue learning or a collapse in performance for PPO was shown in prior works, its relation with a decrease in representation rank was not studied.”
> > >
> > > Regarding the measure of loss of plasticity, there is a different, but not necessarily incompatible, one compared to yours, which considers loss of plasticity to be the phenomenon where, on a task of similar difficulty, the learner performs worse than when learning from scratch, such as in Abbas et al. (2023). I believe loss of plasticity claimed in Dohare et al. (2021, 2023a) is in a similar sense rather than merely speculative. Hence, the existence of these results, in my opinion, still weakens the claim that “revealing that PPO agents are also affected by … loss of plasticity”, which is part of (2).
> > >
> > > Regarding performance collapse, Dohare et al. (2023b) clearly show it, too, weakening the strong claim of (1) again.
> > >
> > > It is expected that newer works will be more thorough than previous ones, which I have also acknowledged for your work. However, acknowledging highly relevant prior results, such as those by Dohare et al. (2023b), helps readers assess the work more comprehensively and in the context of the literature. The weaknesses of this prior work are not a valid cause to avoid acknowledging or positioning your work relative to it. You already refer to Dohare et al. (2023) later, for which these weaknesses indeed were not a problem.
> > >
> > > I do find your reasoning for the discrepancy in results compared to Dohare et al. (2023b) acceptable. I also find your argument regarding PPO+L2 to be reasonable.
> > >
> > > > Given an extra page, we are happy to cite and relate to all the aforementioned works to give more context to our work.
> > >
> > > > Overall, the findings in these works and our work are complementary, and further discussing them on an additional page of related work is mutually beneficial.
> > >
> > > These sound reasonable to me as well.
> > >
> > > > We commit to discussing all the related work raised in this rebuttal. However, we strongly believe that this does not change the position of our work
> > >
> > > I think the position might change due to the weakening of claims (1) and (2). However, I do not think a change in the position necessarily weakens the work itself. A more accurate, fair, and comprehensive position, for example, something similar to what I suggested above regarding a revised claim, may rather strengthen the claims and the contributions.

---

> > > > ### Author Response · Authors · 2024-08-13
> > > >
> > > > We thank the AC for the prompt and detailed response and for clearing our concerns. This is very much appreciated, and we recognize the contribution of the AC, which is well beyond expectations.
> > > >
> > > > We apologize to the community if, while expressing our concerns, any of our comments came across as overly critical. Our intention is not to undermine any previous work. Instead, we want to grow the understanding of plasticity by acknowledging previous work and building on previous limitations, as future work will build on ours.
> > > >
> > > > > … you also agree to remove “thought capable of training indefinitely,” Moreover, the “it” of (1) in the abstract is described as the following phenomenon:
> > > >  … exhibit a decrease in representation rank, often correlated with an inability to continue learning or a collapse in performance.
> > > >
> > > > **The phenomenon we refer to in the quoted text is “representation collapse”,** which can be measured by tracking various metrics we discussed in our paper. **This is what we claim is overlooked. The inability to continue learning and collapse in performance are correlations to it, which can also correlate with many other undesired learning dynamics.** However, we acknowledge that when combined with the “thought capable of training indefinitely,” **readers may interpret that all parts of the correlation have been “overlooked in on-policy policy optimization.”**
> > > >
> > > > **To avoid any misinterpretation and better contextualize our work, we will, as the AC suggests, reformulate this statement** and other statements referring to previous investigations of “performance collapse” and “inability to continue learning in policy optimization” to explicitly build on Dohare et al.’s work (Dohare et al. 2021, 2023a for inability to continue learning in non-stationary environments, and Dohare et al. 2023b for collapse in stationary environments). **We will position our work as motivated to understand this previously observed collapse and inability to continue learning from the lens of representation dynamics used in value-based methods.**
> > > >
> > > > As the AC suggests, rather than weakening our work, **it will make it connect different previous works, providing a better overview of it and strengthening the “understanding” narrative of the submission.**
> > > >
> > > > > Regarding the measure of loss of plasticity ...
> > > >
> > > > We agree. The community has still to tease out the different notions of “plasticity loss,” and this has been raised by a reviewer in the discussions. We agree that picking one definition (random label fitting) and formulating strong claims of novelty with respect to it (2) can seem to undermine the claim with another but relating definition (Abbas et al. 2023) of the same name. **We suggest making our statement precisely about random label fitting and elaborate on the similarities between the two notions in our related work section, highlighting that Dohare et al. (2021, 2023a) also show a loss of plasticity in policy optimization from Abbas et al. (2023)’s perspective.** Again, this will give more context to our readers and avoid misunderstandings.
> > > >
> > > > > It is expected that newer works will be more thorough than previous ones, which I have also acknowledged for your work. … The weaknesses of this prior work are not a valid cause …
> > > >
> > > > We thank the AC for highlighting the thoroughness of our work, as stated at the beginning of this reply and in the last sentence quoted, we are happy to refer to all previous work that can better contextualize our work and grow the understanding of the plasticity community, and help the trust-region subcommunity that we bring with this work, navigate the previous works in plasticity.
> > > >
> > > > We are glad the rest of the AC concerns have been addressed. Below, we summarize the major points that we will rephrase and help to reposition our work.
> > > > -   **Adding an extra page of related work,** including **Abbas et al. (2023), Sokar et al. (2023), and Kumar et al. (2023) for broader discussions of plasticity** as discussed in our previous reply, **the line of work by Dohare et al.** as discussed in this reply, and **those from discussion with reviewers like PPG (Cobbe et al 2020), DR3 (Kumar et al. 2022), and RD (Ma et al. 2024).**
> > > > -   **Rephrasing the claims about overlooked collapse and inability to continue training in policy optimization: it is rather the connection to representation dynamics that has not been investigated.**
> > > > -   **Rephrasing the contribution of PFO refocusing its contribution on strengthening our analysis rather than emphasizing raw performance.**
> > > > -   **Clarify the scope of the connection between representation collapse and trust region collapse highlighting that although the connection mainly holds in the collapse regime, mitigating representation collapse should be throughout training.**
> > > >
> > > > We thank the AC once again for the productive feedback, and we look forward to continuing this dialogue and refining our manuscript to achieve the best outcome for the readers.

---

> > > > > ### Comment · Area_Chair_MRD7 · 2024-08-14
> > > > >
> > > > > Thank you very much for further clarification, and no worries at all. I understand very well how intense this short period of rebuttal can be. I also realize that we are perhaps coming from slightly different perspectives. I greatly appreciate the authors' effort to help these perspectives converge. This effort, if reflected in the paper, will benefit the community and strengthen the contribution by situating it well within the existing literature. This will perhaps be the most scholarly outcome of our discussion.
> > > > >
> > > > > My inquiries have been fulfilled. I do not have any further questions at this point. Thanks again for the response.

---

> > > > > > ### Author Response · Authors · 2024-08-14
> > > > > >
> > > > > > We are grateful to the AC for their understanding. We thank all the reviewers and the AC for their valuable suggestions and active engagement during the discussion phase.  We commit to reflecting all these efforts and the converged perspective in the paper.

---

### Decision · Program_Chairs · 2024-09-25

**Decision:**

Accept (poster)

**Comment:**

This paper is on rank collapse and loss of plasticity in reinforcement learning policies, which is claimed to have not been studied for on-policy policy gradient methods such as PPO. Thorough experiments are conducted to demonstrate and investigate this collapse phenomenon in PPO, revealing its connection to the degradation of the trust region, which the reviewers found to be novel, insightful, and helpful for the community.

Overall, this is a valuable addition to the recent investigative papers on the collapse and loss of plasticity phenomenon. Some reviewers showed concern about the work not leading to clear performance improvement, as mentioned by the rejecting reviewer. The authors need to address this concern by clearly describing the scope of the paper. They are also suggested to include results on the usual set of Mujoco tasks, which they produced during the rebuttal phase.

Further, the area chair discussed with the authors whether loss of plasticity and collapse had previously been shown for PPO. The area chair provided references where such evidence already exists. The authors agreed to modify their claim incorporating such existing literature, which strengthens the paper by situating it well within the literature. Due to this shared understanding and overall agreement during the discussion, this paper will be a valuable contribution to this venue.